# Learning with little mixing

**Ingvar Ziemann**
KTH Royal Institute of Technology
ziemann@kth.se

**Stephen Tu**
Robotics at Google
stephentu@google.com

## Abstract

We study square loss in a realizable time-series framework with martingale difference noise. Our main result is a fast rate excess risk bound which shows that whenever a *trajectory hypercontractivity* condition holds, the risk of the least-squares estimator on dependent data matches the iid rate order-wise after a burn-in time. In comparison, many existing results in learning from dependent data have rates where the effective sample size is deflated by a factor of the mixing-time of the underlying process, even after the burn-in time. Furthermore, our results allow the covariate process to exhibit long range correlations which are substantially weaker than geometric ergodicity. We call this phenomenon *learning with little mixing*, and present several examples for when it occurs: bounded function classes for which the $L^2$ and $L^{2+\varepsilon}$ norms are equivalent, ergodic finite state Markov chains, various parametric models, and a broad family of infinite dimensional $\ell^2(\mathbb{N})$ ellipsoids. By instantiating our main result to system identification of nonlinear dynamics with generalized linear model transitions, we obtain a nearly minimax optimal excess risk bound after only a polynomial burn-in time.

## 1  Introduction

Consider regression in the context of the time-series model:

$$Y_t = f_\star(X_t) + W_t, \qquad t = 0, 1, 2, \ldots. \tag{1}$$

Such models are ubiquitous in applications of machine learning, signal processing, econometrics, and control theory. In our setup, the learner is given access to $T \in \mathbb{N}_+$ pairs $\{(X_t, Y_t)\}_{t=0}^{T-1}$ drawn from the model (1), and is asked to output a hypothesis $\widehat{f}$ from a hypothesis class $\mathscr{F}$ which best approximates the (realizable) regression function $f_\star \in \mathscr{F}$ in terms of square loss.

In this work, we study the least-squares estimator (LSE). This procedure minimizes the empirical risk associated to the square loss over the class $\mathscr{F}$. When each pair of observations $(X_t, Y_t)$ is drawn iid from some fixed distribution, this procedure is minimax optimal over a broad set of hypothesis classes [1–4]. However, much less is known about the optimal rate of convergence for the general time-series model (1), as correlations across time in the covariates $\{X_t\}$ complicate the analysis.

With this in mind, we seek to extend our understanding of the minimax optimality of the LSE for the time-series model (1). We show that for a broad class of function spaces and covariate processes, the effects of data dependency across time enter the LSE excess risk only as a higher order term, whereas the leading term in the excess risk remains order-wise identical to that in the iid setting. Hence, after a sufficiently long, but finite *burn-in time*, the LSE's excess risk scales as if all $T$ samples are independent. This behavior applies to processes that exhibit correlations which decay slower than geometrically. We refer to this double phenomenon, where the mixing-time only enters as a burn-in time, and where the mixing requirement is mild, as *learning with little mixing*.

Our result stands in contrast to a long line of work on learning from dependent data (see e.g., [5–14] and the references within), where the blocking technique [5] is used to create independence amongst

36th Conference on Neural Information Processing Systems (NeurIPS 2022).

the dependent covariates, so that tools to analyze independent learning can be applied. While these aforementioned works differ in their specific setups, the main commonality is that the resulting dependent data rates mimic the corresponding independent rates, but with the caveat that the sample size is replaced by an "effective" sample size that is *decreased* in some way by the mixing-time, even after any necessary burn-in time. Interestingly, the results of Ziemann et al. [15] studying the LSE on the model (1) also suffer from such sample degradation, but do not rely on the blocking technique.

The model (1) captures learning dynamical systems by setting $Y_t = X_{t+1}$, so that the regression function $f_\star$ describes the dynamics of the state variable $X_t$. Recent progress in system identification shows that the lack of ergodicity does not necessarily degrade learning rates. Indeed, when the states evolve as a linear dynamical system (i.e., the function $f_\star$ is linear), learning rates are not deflated by any mixing times, and match existing rates for iid linear regression [16–19]. Kowshik et al. [20], Gao and Raskutti [21] extend results of this flavor to parameter recovery of dynamics driven by a generalized linear model. The extent to which this phenomenon—less ergodicity not impeding learning—generalizes beyond linear and generalized linear models is a key motivation for our work.

**Contributions** We consider the realizable setting, where $f_\star$ is assumed to be contained in a known function space $\mathscr{F}$. Our results rest on two assumptions regarding both the covariate process $\{X_t\}$ and the function space $\mathscr{F}$. The first assumption posits that the process $\{X_t\}$ exhibits some mild form of ergodicity (that is significantly weaker than the typical geometric ergodicity assumption). The second assumption is a hypercontractivity condition that holds uniformly in $\mathscr{F}$ along the trajectory $\{X_t\}$, extending contractivity assumptions for iid learning [3] to dependent processes.

Informally, our main result (Theorem 4.1, presented in Section 4), shows that under these two assumptions, letting $\mathrm{comp}(\mathscr{F})$ denote some (inverse) measure of complexity of $\mathscr{F}$, the LSE $\widehat{f}$ satisfies:

$$\mathbf{E}\|\widehat{f} - f_\star\|_{L^2}^2 \lesssim \left( \frac{\text{dimensional factors} \times \sigma_W^2}{T} \right)^{\mathrm{comp}(\mathscr{F})} + \text{higher order } o(1/T^{\mathrm{comp}(\mathscr{F})}) \text{ terms.} \quad (2)$$

The first term in (2) matches existing LSE risk bounds for iid learning order-wise, and most importantly, does not include any dependence on the mixing-time of the process. Indeed, all mixing-time dependencies enter only in the higher order term. Since this term scales as $o(1/T^{\mathrm{comp}(\mathscr{F})})$, it becomes negligible after a finite burn-in time. This captures the crux of our results: on a broad class of problems, given enough data, the LSE applied to time-series model (1) behaves as if all samples are independent.

Section 5 provides several examples for which the trajectory hypercontractivity assumption holds. When the covariate process $\{X_t\}$ is generated by a finite-state irreducible and aperiodic Markov chain, then any function class $\mathscr{F}$ satisfies the requisite condition. More broadly, the condition is satisfied for any bounded function classes for which the $L^2$ and $L^{2+\varepsilon}$ norms (along trajectories) are equivalent. Next, we show that many infinite dimensional function spaces based on $\ell^2(\mathbb{N})$ ellipsoids satisfy our hypercontractivity condition, demonstrating that our results are not inherently limited to finite-dimensional hypothesis classes.

To demonstrate the broad applicability of our framework, Section 6 instantiates our main result on two system identification problems that have received recent attention in the literature: linear dynamical systems (LDS), and systems with generalized linear model (GLM) transitions. For stable LDS, after a polynomial burn-in time, we recover an excess risk bound that matches the iid rate. A more general form of this result was recently established by Tu et al. [19]. For stable GLMs, also after a polynomial burn-in time, we obtain the first excess risk bound for this problem which matches the iid rate, up to logarithmic factors in various problem constants including the mixing-time. In both of these settings, our excess risk bounds also yield nearly optimal rates for parameter recovery, matching known results for LDS [16] and GLMs [20] in the stable case. In Appendix A, we show experimentally, using the stable GLM model, that the trends predicted by our theory are indeed realized in practice.

## 2 Related work

While regression is a fundamental problem studied across many disciplines, our work draws its main inspiration from Mendelson [3] and Simchowitz et al. [16]. Mendelson [3] shows that for nonparametric iid regression, only minimal assumptions are required for one-sided isometry, and

thus learning. Simchowitz et al. [16] build on this intuition and provide mixing-free rates for linear regression over trajectories generated by a linear dynamical system. We continue this trend, by leveraging one-sided isometry to show that mixing only enters as a higher order term in the rates of the nonparametric LSE. More broadly, the technical developments we follow synthesize techniques from two lines of work: nonparametric regression with iid data, and learning from dependent data.

**Nonparametric regression with iid data.** Beyond the seminal work of Mendelson [3], the works [2, 22, 23] all study iid regression with square loss under various moment equivalence conditions. In addition to moment equivalence, we build on the notion of offset Rademacher complexity defined by [24] in the context of iid regression. Indeed, we show that a martingale analogue of the offset complexity (described in [15]) characterizes the LSE rate in (1).

**Learning from dependent data.** As discussed previously, many existing results for learning from dependent data reduce the problem to independent learning via the blocking technique [5], at the expense of sample complexity deflation by the mixing-time. Nagaraj et al. [25] prove a lower bound for linear regression stating that in a worst case agnostic model, this deflation is unavoidable. Moreover, if the linear regression problem is realizable, Nagaraj et al. [25] provide upper and lower bounds showing that the mixing-time only affects the burn-in time, but not the final risk. We note that their upper bound is an algorithmic result that holds only for a specific modification of SGD. Our work can be interpreted as an upper bound in the more general nonparametric setting, where we put forth sufficient conditions to recover the iid rate after a burn-in time. Our result is algorithm agnostic and directly applies to the empirical risk minimizer. Ziemann et al. [15] also study the model (1), and provide an information-theoretic analysis of the nonparametric LSE. However, their approach fundamentally reduces to showing two-sided concentration—something our work evades— and therefore their bounds incur worst case dependency on the mixing-time. Roy et al. [13] extend the results from Mendelson [3] to the dependent data setting. While following Mendelson's argument allows their results to handle non-realizability and heavy-tailed noise, their proof ultimately still relies on two-sided concentration for both the "version space" and the "noise interaction". Hence, their rates end up degrading for slower mixing processes. We note that this is actually expected in the non-realizable setting in light of the lower bounds in Nagaraj et al. [25].

The measure of dependencies we use for the process $\{X_t\}$ is due to Samson [26]. Recently, Dagan et al. [27] use a similar measure to study learning when the covariates have no obvious sequential ordering (e.g., a graph structure or Ising model). However, our results are not directly comparable, other than noting that their risk bounds degrade as the measure of correlation increases.

Results in linear system identification show that lack of ergodicity does not degrade parameter recovery rates [16–19, 28, 29]. Beyond linear system identification, Kowshik et al. [20], Gao and Raskutti [21], Sattar and Oymak [30], Foster et al. [31] prove parameter recovery bounds for dynamical systems driven by a generalized linear model (GLM) transition. Most relevant are Kowshik et al. [20] and Gao and Raskutti [21], who again show that the lack of ergodicity does not hamper rates. Indeed, Gao and Raskutti [21] even manage to do so in a semiparametric setting with an unknown link function. As mentioned previously, our main result instantiated to these problems in the stable case matches existing excess risk and parameter recovery bounds for linear system identification, and actually provides the sharpest known excess risk bound for the GLM setting (when the link function is known). A more detailed comparison to existing LDS results is given in Appendix I.4, and to existing GLM results in Appendix J.1.

## 3   Problem formulation

The time-series (1) evolves on two subsets of Euclidean space, $\mathsf{X} \subset \mathbb{R}^{d_\mathsf{X}}$ and $\mathsf{Y} \subset \mathbb{R}^{d_\mathsf{Y}}$, with $X_t \in \mathsf{X}$ and $Y_t, W_t \in \mathsf{Y}$. Expectation (resp. probability) with respect to all the randomness of the underlying probability space is denoted by $\mathbf{E}$ (resp. $\mathbf{P}$). The Euclidean norm on $\mathbb{R}^d$ is denoted $\|\cdot\|_2$, and the unit sphere in $\mathbb{R}^d$ is denoted $\mathbb{S}^{d-1}$. For a matrix $M \in \mathbb{R}^{d_1 \times d_2}$, $\|M\|_{\mathsf{op}}$ denotes the largest singular value, $\sigma_{\min}(M)$ the smallest non-zero singular value, and $\mathrm{cond}(M) = \|M\|_{\mathsf{op}}/\sigma_{\min}(M)$ the condition number. When the matrix $M$ is symmetric, $\lambda_{\min}(M)$ will be used to denote its minimum eigenvalue.

We assume there exists a filtration $\{\mathcal{F}_t\}$ such that (a) $\{W_t\}$ is a square integrable martingale difference sequence (MDS) with respect to this filtration, and (b) $\{X_t\}$ is adapted to $\{\mathcal{F}_{t-1}\}$. Further tail conditions on this MDS will be imposed as necessary later on.

Let $\mathscr{F}$ be a hypothesis space of functions mapping $\mathbb{R}^{d_X}$ to $\mathbb{R}^{d_Y}$. We assume that the true regression function is an element of $\mathscr{F}$ (i.e., $f_\star \in \mathscr{F}$), and that $\mathscr{F}$ is known to the learner. Given two compatible function spaces $\mathscr{F}_1, \mathscr{F}_2$, let $\mathscr{F}_1 - \mathscr{F}_2 \triangleq \{f_1 - f_2 \mid f_1 \in \mathscr{F}_1, f_2 \in \mathscr{F}_2\}$. A key quantity in our analysis is the shifted function class $\mathscr{F}_\star \triangleq \mathscr{F} - \{f_\star\}$. Our results will be stated under the assumption that $\mathscr{F}_\star$ is *star-shaped*,[1] although we will see that this is not too restrictive. For any function $f : \mathsf{X} \to \mathbb{R}^{d_Y}$, we define $\|f\|_\infty \triangleq \sup_{x \in \mathsf{X}} \|f\|_2$. A function $f$ is $B$-bounded if $\|f\|_\infty \leq B$. Similarly, a hypothesis class is $B$-bounded if each of its elements is $B$-bounded. For a bounded class $\mathscr{F}$ and resolution $\varepsilon > 0$, the quantity $\mathcal{N}_\infty(\mathscr{F}, \varepsilon)$ denotes the size of the minimal $\varepsilon$-cover of $\mathscr{F}$ (contained in $\mathscr{F}$) in the $\|\cdot\|_\infty$-norm.

We fix a $T \in \mathbb{N}_+$, indicating the number of labeled observations $\{(X_t, Y_t)\}_{t=0}^{T-1}$ from the time-series (1) that are available to the learner. The joint distribution of $X_{0:T-1} \triangleq (X_0, \ldots, X_{T-1})$ is denoted $\mathsf{P}_X$. For $p \geq 1$, we endow $\mathscr{F} - \mathscr{F}$ with $L^p(\mathsf{P}_X)$ norms: $\|f - g\|_{L^p}^p \triangleq \frac{1}{T} \sum_{t=0}^{T-1} \mathbf{E}\|f(X_t) - g(X_t)\|_2^p$, where expectation is taken with respect to $\mathsf{P}_X$. We will mostly be interested in $L^2(\mathsf{P}_X)$, hereafter often just referred to as $L^2$. This is the $L^2$ space associated to the law of the uniform mixture over $X_{0:T-1}$ and thus, for iid data, coincides with the standard $L^2$ space often considered in iid regression. For a radius $r > 0$, we let $B(r)$ denote the closed ball of $\mathscr{F}_\star$ with radius $r$ in $L^2$, and we let $\partial B(r)$ denote its boundary: $B(r) \triangleq \{f \in \mathscr{F}_\star \mid \|f\|_{L^2}^2 \leq r^2\}$ and $\partial B(r) \triangleq \{f \in \mathscr{F}_\star \mid \|f\|_{L^2}^2 = r^2\}$.

The learning task is to produce an estimate $\widehat{f}$ of $f_\star$, which renders the excess risk $\|\widehat{f} - f_\star\|_{L^2}^2$ as small as possible. We emphasize that $\|\widehat{f} - f_\star\|_{L^2}^2 = \frac{1}{T} \sum_{t=0}^{T-1} \mathbf{E}_{\tilde{X}_{0:T-1}} \|\widehat{f}(\tilde{X}_t) - f_\star(\tilde{X}_t)\|_2^2$ where $\tilde{X}_{0:T-1}$ is a fresh, statistically independent, sample with the same law $\mathsf{P}_X$ as $X_{0:T-1}$. Namely, $\|\widehat{f} - f_\star\|_{L^2}^2$ is a random quantity, still depending on the internal randomness of the learner and that of the sample $X_{0:T-1}$ used to generate $\widehat{f}$. We study the performance of the least-squares estimator (LSE) defined as $\widehat{f} \in \operatorname{argmin}_{f \in \mathscr{F}} \left\{ \frac{1}{T} \sum_{t=0}^{T-1} \|Y_t - f(X_t)\|_2^2 \right\}$, and measure the excess risk $\mathbf{E}\|\widehat{f} - f_\star\|_{L^2}^2$.

## 4 Results

This section presents our main result. We first detail the definitions behind our main assumptions in Section 4.1. The main result and two corollaries are then presented in Section 4.2.

### 4.1 Hypercontractivity and the dependency matrix

**Hypercontractivity.** We first state our main trajectory hypercontractivity condition, which we will use to establish lower isometry. The following definition is heavily inspired by recent work on learning without concentration [3, 23].

**Definition 4.1** (Trajectory $(C, \alpha)$-hypercontractivity). *Fix constants $C > 0$ and $\alpha \in [1, 2]$. We say that the tuple $(\mathscr{F}, \mathsf{P}_X)$ satisfies the* trajectory $(C, \alpha)$-hypercontractivity *condition if*

$$\mathbf{E}\left[\frac{1}{T} \sum_{t=0}^{T-1} \|f(X_t)\|_2^4\right] \leq C \left(\mathbf{E}\left[\frac{1}{T} \sum_{t=0}^{T-1} \|f(X_t)\|_2^2\right]\right)^\alpha \text{ for all } f \in \mathscr{F}. \tag{3}$$

*Here, the expectation is with respect to $\mathsf{P}_X$, the joint law of $X_{0:T-1}$.*

Condition (3) interpolates between boundedness and small-ball behavior. Indeed, if the class $\mathscr{F}$ is $B$-bounded, then it satisfies trajectory $(B^2, 1)$-hypercontractivity trivially. On the other hand, for $\alpha = 2$, (3) asks that $\|f\|_{L^4} \leq C^{1/4}\|f\|_{L^2}$ for trajectory-wise $L^p$-norms; by the Paley-Zygmund inequality, this implies that a small-ball condition holds. Moreover, if for some $\varepsilon \in (0, 2)$, the trajectory-wise $L^2$ and $L^{2+\varepsilon}$ norms are equivalent on $\mathscr{F}$, then Proposition 5.2 (Section 5) shows that the condition holds for a nontrivial $\alpha = 1 + \varepsilon/2 \in (1, 2)$. More examples are given in Section 5.

Our main results assume that $(\mathscr{F}_\star, \mathsf{P}_X)$ (or a particular subset of $\mathscr{F}_\star$) satisfies the trajectory $(C, \alpha)$-hypercontractivity condition with $\alpha > 1$, which we refer to as the *hypercontractive* regime. The condition $\alpha > 1$ is required in our analysis for the lower order excess risk term to not depend on

---

[1] A function class $\mathscr{F}$ is star-shaped if for any $\alpha \in [0, 1]$, $f \in \mathscr{F}$ implies $\alpha f \in \mathscr{F}$.

the mixing-time. Our results instantiated for the $\alpha = 1$ case directly correspond to existing work by Ziemann et al. [15], and exhibit a lower order term that depends on the mixing-time.

**Ergodicity via the dependency matrix.** We now state the main definition we use to measure the stochastic dependency of a process. Recall that for two measures $\mu, \nu$ on the same measurable space with $\sigma$-algebra $\mathcal{A}$, the total-variation norm is defined as $\|\mu - \nu\|_{\mathsf{TV}} \triangleq \sup_{A \in \mathcal{A}} |\mu(A) - \nu(A)|$.

**Definition 4.2** (Dependency matrix, Samson [26, Section 2])**.** *The* dependency matrix *of a process* $\{Z_t\}_{t=0}^{T-1}$ *with distribution* $\mathsf{P}_Z$ *is the (upper-triangular) matrix* $\Gamma_{\mathsf{dep}}(\mathsf{P}_Z) = \{\Gamma_{ij}\}_{i,j=0}^{T-1} \in \mathbb{R}^{T \times T}$ *defined as follows. Let* $\mathcal{Z}_{0:i}$ *denote the* $\sigma$*-algebra generated by* $\{Z_t\}_{t=0}^{i}$*. For indices* $i < j$*, let*

$$\Gamma_{ij} = \sqrt{2 \sup_{A \in \mathcal{Z}_{0:i}} \|\mathsf{P}_{Z_{j:T-1}}(\cdot \mid A) - \mathsf{P}_{Z_{j:T-1}}\|_{\mathsf{TV}}}. \tag{4}$$

*For the remaining indices* $i \geq j$*, let* $\Gamma_{ii} = 1$ *and* $\Gamma_{ij} = 0$ *when* $i > j$ *(below the diagonal).*

Given the dependency matrix from Definition 4.2, we measure the dependency of the process $\mathsf{P}_X$ by the quantity $\|\Gamma_{\mathsf{dep}}(\mathsf{P}_X)\|_{\mathsf{op}}$. Notice that this quantity always satisfies $1 \leq \|\Gamma_{\mathsf{dep}}(\mathsf{P}_X)\|_{\mathsf{op}} \lesssim T$. The lower bound indicates that the process $\mathsf{P}_X$ is independent across time. The upper bound indicates that the process is fully dependent, e.g., $X_{t+1} = X_t$ for all $t \in \mathbb{N}$.

Our results apply to cases where $\|\Gamma_{\mathsf{dep}}(\mathsf{P}_X)\|_{\mathsf{op}}^2$ grows sub-linearly in $T$– the exact requirement depends on the specific function class $\mathscr{F}$. If the process $\{X_t\}$ is geometrically $\phi$-mixing, then $\|\Gamma_{\mathsf{dep}}(\mathsf{P}_X)\|_{\mathsf{op}}^2$ is upper bounded by a constant that depends on the mixing-time of the process, and is independent of $T$ [26, Section 2]. Other examples, such as processes satisfying Doeblin's condition [32], are given in Samson [26, Section 2]. When $\{X_t\}$ is a stationary time-homogenous Markov chain with invariant distribution $\pi$, the coefficients $\Gamma_{ij}$ simplify to $\Gamma_{ij}^2 = 2 \sup_{A \in \mathcal{X}_\infty} \|\mathsf{P}_{X_{j-i}}(\cdot \mid A) - \pi\|_{\mathsf{TV}}$ for indices $j > i$, where $\mathcal{X}_\infty$ is the $\sigma$-algebra generated by $X_\infty \sim \pi$ (cf. Proposition F.1). Hence, the requirement $\|\Gamma_{\mathsf{dep}}(\mathsf{P}_X)\|_{\mathsf{op}}^2 \lesssim T^\beta$ for $\beta \in (0, 1)$ then corresponds to $\sup_{A \in \mathcal{X}_\infty} \|\mathsf{P}_{X_t}(\cdot \mid A) - \pi\|_{\mathsf{TV}} \lesssim 1/t^{1-\beta}$ for $t \in \mathbb{N}_+$. Jarner and Roberts [33] give various examples and conditions to check polynomial convergence rates for Markov chains. We also provide further means to verify $\|\Gamma_{\mathsf{dep}}(\mathsf{P}_X)\|_{\mathsf{op}} = O(1)$ in Appendix F and Appendix G.

## 4.2 Learning with little mixing

A key quantity appearing in our bounds is a martingale variant of the notion of Gaussian complexity.

**Definition 4.3** (Martingale offset complexity, cf. Liang et al. [24], Ziemann et al. [15])**.** *For the regression problem (1), the martingale offset complexity of a function space* $\mathscr{F}$ *is given by:*

$$\mathsf{M}_T(\mathscr{F}) \triangleq \sup_{f \in \mathscr{F}} \left\{ \frac{1}{T} \sum_{t=0}^{T-1} 4\langle W_t, f(X_t) \rangle - \|f(X_t)\|_2^2 \right\}. \tag{5}$$

Recall that $\mathscr{F}_\star = \mathscr{F} - \{f_\star\}$ is the centered function class and $\partial B(r) = \{f \in \mathscr{F}_\star \mid \|f\|_{L^2} = r\}$ is the boundary of the $L^2$ ball $B(r)$. The following theorem is the main result of this paper.

**Theorem 4.1.** *Fix* $B > 0$*,* $C : (0, B] \to \mathbb{R}_+$*,* $\alpha \in [1, 2]$*, and* $r \in (0, B]$*. Suppose that* $\mathscr{F}_\star$ *is star-shaped and* $B$*-bounded. Let* $\mathscr{F}_r \subset \mathscr{F}_\star$ *be a* $r/\sqrt{8}$*-net of* $\partial B(r)$ *in the supremum norm* $\|\cdot\|_\infty$*, and suppose that* $(\mathscr{F}_r, \mathsf{P}_X)$ *satisfies the trajectory* $(C(r), \alpha)$*-hypercontractivity condition (cf. Definition 4.1). Then:*

$$\mathbf{E}\|\widehat{f} - f_\star\|_{L_2}^2 \leq 8\mathbf{E}\mathsf{M}_T(\mathscr{F}_\star) + r^2 + B^2|\mathscr{F}_r| \exp\left(\frac{-Tr^{4-2\alpha}}{8C(r)\|\Gamma_{\mathsf{dep}}(\mathsf{P}_X)\|_{\mathsf{op}}^2}\right). \tag{6}$$

The assumption that $\mathscr{F}_\star$ is star-shaped in Theorem 4.1 is not particularly restrictive. Indeed, Theorem 4.1 still holds if we replace $\mathscr{F}_\star$ by its star-hull $\mathrm{star}(\mathscr{F}_\star) \triangleq \{\gamma f \mid \gamma \in [0, 1], \ f \in \mathscr{F}_\star\}$, and $\partial B(r)$ with the boundary of the $r$-sphere of $\mathrm{star}(\mathscr{F}_\star)$. In this case, we note that (a) the metric entropy of $\mathrm{star}(\mathscr{F}_\star)$ is well controlled by the metric entropy of $\mathscr{F}_\star$,[2] and (b) the trajectory hypercontractivity conditions over a class $\mathscr{F}_\star$ and its star-hull $\mathrm{star}(\mathscr{F}_\star)$ are equivalent. Hence, at least whenever we are

---

[2]Specifically, $\log \mathcal{N}_\infty(\mathrm{star}(\mathscr{F}_\star), \varepsilon) \leq \log(2B/\varepsilon) + \log \mathcal{N}_\infty(\mathscr{F}_\star, \varepsilon/2)$ [34, Lemma 4.5].

able to verify hypercontractivity over the entire class $\mathscr{F}_\star$, little generality is lost. While most of our examples are star-shaped, we will need the observations above when we work with generalized linear model dynamics in Section 6.2.

To understand Theorem 4.1, we will proceed in a series of steps. We first need to understand the martingale complexity term $\mathbf{EM}_T(\mathscr{F}_\star)$. Since $\mathscr{F}_\star$ is $B$-bounded, if one further imposes the tail conditions that the noise process $\{W_t\}$ is a $\sigma_W^2$-sub-Gaussian MDS,[3] a chaining argument detailed in Ziemann et al. [15, Lemma 4] shows that:

$$\mathbf{EM}_T(\mathscr{F}_\star) \lesssim \inf_{\gamma > 0, \delta \in [0, \gamma]} \left\{ \frac{\sigma_W^2 \log \mathcal{N}_\infty(\mathscr{F}_\star, \gamma)}{T} + \sigma_W \sqrt{d_Y} \delta + \frac{\sigma_W}{\sqrt{T}} \int_\delta^\gamma \sqrt{\log \mathcal{N}_\infty(\mathscr{F}_\star, s)} ds + \gamma^2 \right\}. \tag{7}$$

In particular, this bound only depends on $\mathscr{F}_\star$ and is *independent* of $\|\Gamma_{\mathsf{dep}}(\mathsf{P}_X)\|_{\mathsf{op}}^2$. Furthermore, (7) coincides with the corresponding risk bound for the LSE with iid covariates [24].

Given that $\mathbf{EM}_T(\mathscr{F}_\star)$ corresponds to the rate of learning from $T$ iid covariates, the form of (6) suggests that we choose $r^2 \lesssim \mathbf{EM}_T(\mathscr{F}_\star)$, so that the dominant term in (6) is equal to $\mathbf{EM}_T(\mathscr{F}_\star)$ in scale. Given that $r$ has been set, the only remaining degree of freedom in (6) is to set $T$ large enough (the burn-in time) so that the third term is dominated by $r^2$. Thus, it is this third term in (6) that captures the interplay between the function class $\mathscr{F}_\star$ and the dependency measure $\|\Gamma_{\mathsf{dep}}(\mathsf{P}_X)\|_{\mathsf{op}}$. We will now consider specific examples to illustrate how the burn-in time can be set.

Our first example supposes that (a) $\mathscr{F}_\star$ satisfies the trajectory $(C, 2)$-hypercontractivity condition, and that (b) $\mathscr{F}_\star$ is nonparametric, but not too large:

$$\exists\, p > 0,\ q \in (0, 2)\ \text{s.t.}\ \log \mathcal{N}_\infty(\mathscr{F}_\star, \varepsilon) \le p \left( \frac{1}{\varepsilon} \right)^q\ \text{for all}\ \varepsilon \in (0, 1). \tag{8}$$

Covering numbers of the form (8) are typical for sufficiently smooth function classes, e.g. the space of $k$-times continuously differentiable functions mapping $\mathsf{X} \to \mathsf{Y}$ for any $k \ge \lceil d_X/2 \rceil$ [35]. If condition (8) holds and the noise process $\{W_t\}$ is a sub-Gaussian MDS, inequality (7) yields $\mathbf{EM}_T(\mathscr{F}_\star) \lesssim T^{-\frac{2}{2+q}}$, and hence we want to set $r^2 = o(T^{-\frac{2}{2+q}})$. Carrying out this program yields the following corollary.

**Corollary 4.1.** *Fix $B \ge 1$, $C > 0$, $p > 0$, $q \in (0, 2)$, and $\gamma \in (0, \frac{q}{2+q})$. Suppose that $\mathscr{F}_\star$ is star-shaped, $B$-bounded, satisfies (8), and $(\mathscr{F}_\star, \mathsf{P}_X)$ satisfies the trajectory $(C, 2)$-hypercontractivity condition. Suppose that $T$ satisfies:*

$$T \ge \max \left\{ \left[ 8(32p + 1)C\|\Gamma_{\mathsf{dep}}(\mathsf{P}_X)\|_{\mathsf{op}}^2 \right]^{\frac{1}{1 - \frac{q}{2}\left( \frac{2}{2+q} + \gamma \right)}}, \left[ 4 \log B \vee \frac{8}{q} \log \left( \frac{16}{q} \right) \right]^{\frac{1}{\frac{q}{2}\left( \frac{2}{2+q} + \gamma \right)}} \right\}. \tag{9}$$

*Then, we have that:*

$$\mathbf{E}\|\widehat{f} - f_\star\|_{L^2}^2 \le 8\mathbf{EM}_T(\mathscr{F}_\star) + 2T^{-\left( \frac{2}{2+q} + \gamma \right)}. \tag{10}$$

The rate (10) of Corollary 4.1 highlights the fact that the first order term of the excess risk is bounded by the martingale offset complexity $\mathbf{EM}_T(\mathscr{F}_\star)$. This behavior arises since the dependency matrix $\Gamma_{\mathsf{dep}}(\mathsf{P}_X)$ only appears as the burn-in requirement (9). Here, the value of $q$ constrains how fast $\|\Gamma_{\mathsf{dep}}(\mathsf{P}_X)\|_{\mathsf{op}}^2$ is allowed to grow. In particular, condition (9) requires that $\|\Gamma_{\mathsf{dep}}(\mathsf{P}_X)\|_{\mathsf{op}}^2 = o(T^{1 - \frac{q}{2+q}})$, otherwise the burn-in condition cannot be satisfied for any $\gamma \in (0, \frac{q}{2+q})$.

In our next example, we consider both a variable hypercontractivity parameter $C(r)$ that varies with the covering radius $r$, and also allow $\alpha \in (1, 2]$ to vary. Since our focus is on the interaction of the parameters in the hypercontractivity definition, we will consider smaller function classes with logarithmic metric entropy. This includes parametric classes but also bounded subsets of certain reproducing kernel Hilbert spaces. For such function spaces, one expects $\mathbf{EM}_T(\mathscr{F}_\star) \le \tilde{O}(T^{-1})$, and hence we set $r^2 = o(T^{-1})$.

---

[3]That is, for any $u \in \mathbb{S}^{d_Y - 1}$, $\lambda \in \mathbb{R}$, and $t \in \mathbb{N}$, we have $\mathbf{E}[\exp(\lambda \langle W_t, u \rangle) \mid \mathcal{F}_{t-1}] \le \exp(\lambda^2 \sigma_W^2 / 2)$.

**Corollary 4.2.** *Fix $B \geq 1$, $C : (0,1] \to \mathbb{R}_+$, $\alpha \in (1,2]$, $b_1 \in [0,1)$, $b_2 \in [0,2)$, $\gamma \in (0,1)$, and $p, q \geq 1$. Suppose that $\mathscr{F}_\star$ is star-shaped and $B$-bounded, and that for every $r \in (0,1)$, there exists a $r$-net $\mathscr{F}_r$ of $\partial B(r)$ in the $\|\cdot\|_\infty$-norm such that (a) $\log |\mathscr{F}_r| \leq p \log^q \left(\frac{1}{r}\right)$ and (b) $(\mathscr{F}_r, \mathsf{P}_X)$ satisfies the trajectory $(C(r), \alpha)$-hypercontractivity condition. Next, suppose the growth conditions hold:*

$$\|\Gamma_{\mathsf{dep}}(\mathsf{P}_X)\|_{\mathsf{op}}^2 \leq T^{b_1}, \ \ C(r) \leq (1/r)^{b_2} \ \forall r \in (0,1).$$

*As long as the constants $\alpha$, $b_1$, $b_2$, and $\gamma$ satisfy $\psi := 1 - b_1 - \frac{(1+\gamma)(4-2\alpha+b_2)}{2} > 0$, then for any $T \geq \mathsf{poly}_{\frac{q}{\psi}}\left(p, \log B, \frac{q}{\psi}\right)$, we have:*

$$\mathbf{E}\|\widehat{f} - f_\star\|_{L_2}^2 \leq 8\mathbf{EM}_T(\mathscr{F}_\star) + 2\left(\frac{1}{T}\right)^{1+\gamma}.$$

Here $\mathsf{poly}_{q/\psi}$ denotes a polynomial of degree $O(q/\psi)$ in its arguments– the exact expression is given in the proof. Proposition 5.4 in Section 5 gives an example of an $\ell^2(\mathbb{N})$ ellipsoid which satisfies the assumptions in Corollary 4.2. Corollary 4.2 illustrates the interplay between the function class $\mathscr{F}_\star$, the data dependence of the covariate process $\{X_t\}$, and the hypercontractivity constant $\alpha$. Let us consider a few cases. First, let us suppose that the process $\{X_t\}$ is geometrically ergodic and that $C(r)$ is a constant, so that we can set $b_1$ and $b_2$ arbitrarily close to zero (at the expense of a longer burn-in time). Then, the $\psi > 0$ condition simplifies to $\alpha > 2 - \frac{1}{1+\gamma}$. This illustrates that in the hypercontractivity regime ($\alpha > 1$), there exists a valid setting of $(b_1, b_2, \gamma)$ that satisfies $\psi > 0$. Next, let us consider the case where $C(r)$ is again a constant, but $\{X_t\}$ is not geometrically ergodic. Setting $b_2$ and $\gamma$ arbitrarily close to zero, we have $\psi > 0$ simplifies to $b_1 < \alpha - 1$. Compared to Corollary 4.1, we see that in the case when $\alpha = 2$, the parametric nature of $\mathscr{F}_\star$ allows the dependency requirement to be less strict: $o(T)$ in the parametric case versus $o(T^{1-\frac{q}{2+q}})$ in the nonparametric case.

We conclude with noting that when $\alpha = 1$, it is not possible to remove the dependence on $\|\Gamma_{\mathsf{dep}}(\mathsf{P}_X)\|_{\mathsf{op}}^2$ in the lowest order term. In this situation, our results recover existing risk bounds from Ziemann et al. [15]– see Appendix H for a discussion.

## 5 Examples of trajectory hypercontractivity

In this section, we detail a few examples of trajectory hypercontractivity. Let us begin by considering the simplest possible example: a finite hypothesis class. Let $|\mathscr{F}| < \infty$. Define for any fixed $f \in \mathscr{F}_\star$ the constant $c_f \triangleq \mathbf{E}\left[\frac{1}{T}\sum_{t=0}^{T-1}\|f(X_t)\|_2^4\right] / \left(\mathbf{E}\left[\frac{1}{T}\sum_{t=0}^{T-1}\|f(X_t)\|_2^2\right]\right)^2$, where the ratio $0/0$ is taken to be 1. Then the class $\mathscr{F}_\star$ is trajectory $(\max_{f \in \mathscr{F}_\star} c_f, 2)$-hypercontractive.

Similarly, processes evolving on a finite state space can also be verified to be hypercontractive.

**Proposition 5.1.** *Fix a $\underline{\mu} > 0$. Let $\{\mu_t\}_{t=0}^{T-1}$ denote the marginal distributions of $\mathsf{P}_X$. Suppose that the $\mu_t$'s all share a common support of a finite set of atoms $\{\psi_1, \ldots, \psi_K\} \subset \mathbb{R}^{d_X}$, and that $\min_{0 \leq t \leq T-1} \min_{1 \leq k \leq K} \mu_t(\psi_k) \geq \underline{\mu}$. For any class of functions $\mathscr{F}$ mapping $\{\psi_1, \ldots, \psi_K\} \to \mathbb{R}^{d_Y}$, we have that $\mathscr{F}$ satisfies the trajectory $(1/\underline{\mu}, 2)$-hypercontractivity condition.*

We remark that when $\mathsf{P}_X$ is an aperiodic and irreducible Markov chain over a finite state space, the condition $\underline{\mu} > 0$ is always valid even as $T \to \infty$ [36]. In this case, our findings are related to Wolfer and Kontorovich [37, Theorem 3.1], who show that in the high accuracy regime (i.e., after a burn-in time), the minimax rate of estimating the transition probabilities of such a chain is not affected by the mixing time (in their case the pseudo-spectral gap).

The examples considered thus far rely on the fact that under a certain degree of finiteness, the fourth and second moment can be made uniformly equivalent. The next proposition relaxes this assumption. Namely, if for some $\varepsilon \in (0,2]$ the $L^2$ and $L^{2+\varepsilon}$ norms are equivalent on a bounded class $\mathscr{F}$, this class then satisfies a nontrivial hypercontractivity constant, $\alpha > 1$ (cf. Mendelson [23]).

**Proposition 5.2.** *Fix $\varepsilon \in (0,2]$ and $c > 0$. Suppose that $\mathscr{F}$ is $B$-bounded and that $\|f\|_{L^{2+\varepsilon}} \leq c\|f\|_{L^2}$ for all $f \in \mathscr{F}$. Then $\mathscr{F}$ is trajectory $(B^{2-\varepsilon}c^{2+\varepsilon}, 1+\varepsilon/2)$-hypercontractive.*

Next, we show that for processes $\{X_t\}$ which converge fast enough to a stationary distribution, it suffices to verify the hypercontractivity condition only over the stationary distribution. This mimics

existing results in iid learning, where hypercontractivity is assumed over the covariate distribution [3, 24]. We first recall the definition of the $\chi^2$ divergence between two measures. Let $\mu$ and $\nu$ be two measures over the same probability space, and suppose that $\mu$ is absolutely continuous w.r.t. $\nu$. The $\chi^2(\mu, \nu)$ divergence is defined as $\chi^2(\mu, \nu) \triangleq \mathbf{E}_\nu \left[ \left( \frac{d\mu}{d\nu} - 1 \right)^2 \right]$.

**Proposition 5.3.** *Fix positive $r$, $C_{\chi^2}$, $C_{\mathsf{TV}}$, and $C_{8\to2}$. Suppose that the process $\{X_t\}$ has a stationary distribution $\pi$. Let $\{\mu_t\}$ denote the marginal distributions of $\{X_t\}$, and suppose that the marginals $\{\mu_t\}$ are absolutely continuous w.r.t. $\pi$. Assume the process is ergodic in the sense that:*

$$\sup_{t \in \mathbb{N}} \chi^2(\mu_t, \pi) \le C_{\chi^2}, \quad \frac{1}{T} \sum_{t=0}^{T-1} \|\mu_t - \pi\|_{\mathsf{TV}} \le C_{\mathsf{TV}} r^2. \tag{11}$$

*Suppose also that for all $f \in \mathscr{F}_\star$: $\mathbf{E}_\pi \|f(X)\|_2^8 \le C_{8\to2}(\mathbf{E}_\pi \|f(X)\|_2^2)^4$. Then the set $\partial B(r)$ satisfies $(C, 2)$-trajectory hypercontractivity with $C = (1 + \sqrt{C_{\chi^2}})\sqrt{C_{8\to2}}(1 + C_{\mathsf{TV}} B^2)^2$.*

We further discuss the ergodicity condition (11) in Appendix E.3.1.

**Ellipsoids in $\ell^2(\mathbb{N})$.** Given that equivalence of norms is typically a finite-dimensional phenomenon, one may wonder whether examples of hypercontractivity exist in an infinite-dimensional setting. Here we show that such examples are actually rather abundant. The key is that hypercontractivity need only be satisfied on an $\varepsilon$-cover of $\mathscr{F}_\star$. As discussed above, every finite hypothesis class (and thus every finite cover) is automatically $(C, 2)$-hypercontractive for some $C > 0$. The issue is to ensure that this constant does not grow too fast as one refines the cover. The next result shows that the growth can be controlled for $\ell^2(\mathbb{N})$ ellipsoids of orthogonal expansions. By Mercer's theorem, these ellipsoids correspond to unit balls in reproducing kernel Hilbert spaces [4, Corollary 12.26].

**Proposition 5.4.** *Fix positive constants $\beta$, $B$, $K$, and $q$. Fix a base measure $\lambda$ on $\mathsf{X}$ and suppose that $\{\phi_n\}_{n \in \mathbb{N}_+}$ is an orthonormal system in $L^2(\lambda)$ satisfying $\|\phi_n\|_\infty \le Bn^q$, $\forall n \in \mathbb{N}$. Suppose $\mu_j \le e^{-2\beta j}$ and define the ellipsoid: $\mathscr{P} \triangleq \left\{ f = \sum_{j=1}^\infty \theta_j \phi_j \,\Big|\, \sum_{j=1}^\infty \frac{\theta_j^2}{\mu_j} \le 1 \right\}$. Fix $\varepsilon > 0$, and let $m_\varepsilon$ denote the smallest positive integer solution to $m \ge \frac{2}{\beta} \left| \log \left( \frac{8B}{\beta \varepsilon} \right) \right|$ subject to $\frac{m}{\log m} \ge \frac{q}{\beta}$. Let $P \subset \mathscr{P}$ be an arbitrary subset. There exists an $\varepsilon$-cover $P_\varepsilon$ of $P$ in the $\|\cdot\|_\infty$-norm satisfying $\log |P_\varepsilon| \le m_\varepsilon \log \left( 1 + \frac{8Bm_\varepsilon^q}{\varepsilon} \right)$. Further, let $\{\mu_t\}_{t=0}^{T-1}$ be the marginal distributions of $\mathsf{P}_X$ and suppose that $\max_{0 \le t \le T-1} \max \left\{ \frac{d\mu_t}{d\lambda}, \frac{d\lambda}{d\mu_t} \right\} \le K$. Then, as long as $\varepsilon \le \inf_{f \in P} \|f\|_{L^2(\mathsf{P}_X)}$, $(P_\varepsilon, \mathsf{P}_X)$ is trajectory $(C_\varepsilon, 2)$-hypercontractive with $C_\varepsilon = (1 + 7K^3 B^4 m_\varepsilon^{4q+2})$.*

Proposition 5.4 states that when $\mathscr{F}_\star \subseteq \mathscr{P}$, then $(\partial B(r), \mathsf{P}_X)$ is $(C(r), 2)$-hypercontractive where $C(r) = C_r$ only grows *poly-logarithmically* in $1/r$ and thus verifies the assumptions of Corollary 4.2.

# 6 System identification in parametric classes

To demonstrate the sharpness of our main result, we instantiate Theorem 4.1 on two parametric system identification problems which have received recent attention in the literature: linear dynamical systems (LDS) and generalized linear model (GLM) dynamics.

## 6.1 Linear dynamical systems

Consider the setting where the process $\{X_t\}_{t \ge 0}$ is described by a linear dynamical system:

$$X_{t+1} = A_\star X_t + HV_t, \quad X_0 = HV_0, \quad V_t \in \mathbb{R}^{d_\mathsf{V}}, \quad V_t \sim N(0, I), \quad V_t \perp V_{t'} \,\forall t \ne t'. \tag{12}$$

In this setting, the system identification problem is to recover the dynamics matrix $A_\star$ from $\{X_t\}_{t=0}^{T-1}$ evolving according to (12). We derive rates for recovering $A_\star$ by first deriving an excess risk bound on the least-squares estimator via Theorem 4.1, and then converting the risk bound to a parameter error bound. Since Theorem 4.1 relies on the process being ergodic, we consider the case when $A_\star$ is stable. We start by stating a few standard definitions.

**Definition 6.1.** *Fix a $k \in \{1, \dots, d_\mathsf{X}\}$. The pair $(A, H)$ is $k$-step controllable if* $\mathrm{rank} \left( \begin{bmatrix} H & AH & A^2H & \dots & A^{k-1}H \end{bmatrix} \right) = d_\mathsf{X}$.

For $t \in \mathbb{N}$, let the $t$-step *controllability gramian* be defined as $\Gamma_t \triangleq \sum_{k=0}^{t} A^k H H^{\mathsf{T}} (A^k)^{\mathsf{T}}$. Since the noise in (12) serves as the "control" in this setting, the controllability gramian also coincides with the covariance at time $t$, i.e., $\mathbf{E}[X_t X_t^{\mathsf{T}}] = \Gamma_t$.

**Definition 6.2.** *Fix a $\tau \geq 1$ and $\rho \in (0, 1)$. A matrix $A$ is called $(\tau, \rho)$-stable if for all $k \in \mathbb{N}$ we have $\|A^k\|_{\mathsf{op}} \leq \tau \rho^k$.*

With these definitions in place, we now state our result for linear dynamical system.

**Theorem 6.1.** *Suppose that the matrix $A_\star$ in (12) is $(\tau, \rho)$-stable (cf. Definition 6.2), and that the pair $(A_\star, H)$ is $\kappa$-step controllable (cf. Definition 6.1). Suppose also that $\|A_\star\|_F \leq B$ for some $B \geq 1$. Consider the linear hypothesis class and true regression function:*

$$\mathscr{F} \triangleq \{f(x) = Ax \mid A \in \mathbb{R}^{d_{\mathsf{X}} \times d_{\mathsf{X}}}, \ \|A\|_F \leq B\}, \ f_\star(x) = A_\star x. \tag{13}$$

*Suppose that model (1) follows the process described in (12) with $Y_t = X_{t+1}$. There exists $T_0$ such that the LSE with hypothesis class $\mathscr{F}$ achieves for all $T \geq T_0$:*

$$\mathbf{E}\|\widehat{f} - f_\star\|_{L^2}^2 \leq 8\mathbf{EM}_T(\mathscr{F}_\star) + \frac{4\|H\|_{\mathsf{op}}^2 d_{\mathsf{X}}^2}{T}. \tag{14}$$

*Furthermore, $T_0$ satisfies for a universal positive constant $c_0$:*

$$T_0 = c_0 \frac{\tau^4 \|H\|_{\mathsf{op}}^4 d_{\mathsf{X}}^2}{(1-\rho)^2 \lambda_{\min}(\Gamma_{\kappa-1})^2} \left[ \kappa^2 \vee \frac{1}{(1-\rho)^2} \right] \mathrm{polylog}\left( B, d_{\mathsf{X}}, \tau, \|H\|_{\mathsf{op}}, \frac{1}{\lambda_{\min}(\Gamma_{\kappa-1})}, \frac{1}{1-\rho}, \right). \tag{15}$$

Appendix I.4 contains a more detailed discussion about the results in Theorem 6.1. There, we argue that the term $\mathbf{EM}_T(\mathscr{F}_\star)$ in (14) is proportional to $\|H\|_{\mathsf{op}}^2 d_{\mathsf{X}}^2 / T$ implying that the final rate is proportional to the minimax rate, i.e., $\mathbf{E}\|\widehat{f} - f_\star\|_{L^2}^2 \lesssim \|H\|_{\mathsf{op}}^2 d_{\mathsf{X}}^2 / T$.

## 6.2 Generalized linear models

We next consider the following non-linear dynamical system:

$$X_{t+1} = \sigma(A_\star X_t) + H V_t, \ X_0 = H V_0, \ V_t \in \mathbb{R}^{d_{\mathsf{X}}}, \ V_t \sim N(0, I), \ V_t \perp V_{t'} \ \forall t \neq t'. \tag{16}$$

Here, $A_\star \in \mathbb{R}^{d_{\mathsf{X}} \times d_{\mathsf{X}}}$ is the dynamics matrix and $\sigma : \mathbb{R}^{d_{\mathsf{X}}} \to \mathbb{R}^{d_{\mathsf{X}}}$ is a coordinate wise link function. The notation $\sigma$ will also be overloaded to refer to the individual coordinate function mapping $\mathbb{R} \to \mathbb{R}$. We study the system identification problem where the link function $\sigma$ is assumed to be known, but the dynamics matrix $A_\star$ is unknown and to be recovered from $\{X_t\}_{t=0}^{T-1}$. We will apply Theorem 4.1 to derive a nearly optimal excess risk bound for the LSE on this problem in the stable case.

We start by stating a few assumptions that are again standard in the literature [20, 31].

**Assumption 6.1.** *Suppose that $A_\star$, $H$, and $\sigma$ from the GLM process (16) satisfy:*

1. *(One-step controllability). The matrix $H \in \mathbb{R}^{d_{\mathsf{X}} \times d_{\mathsf{X}}}$ is full rank.*

2. *(Link function regularity). The link function $\sigma : \mathbb{R} \to \mathbb{R}$ is 1-Lipschitz, satisfies $\phi(0) = 0$, and there exists a $\zeta \in (0, 1]$ such that $|\sigma(x) - \sigma(y)| \geq \zeta |x - y|$ for all $x, y \in \mathbb{R}$.*

3. *(Lyapunov stability). There exists a positive definite diagonal matrix $P_\star \in \mathbb{R}^{d_{\mathsf{X}} \times d_{\mathsf{X}}}$ satisfying $P_\star \succcurlyeq I$ and a $\rho \in (0, 1)$ such that $A_\star^{\mathsf{T}} P_\star A_\star \preccurlyeq \rho P_\star$.*

With our assumptions in place, we are ready to instantiate our main result on the process (16).

**Theorem 6.2.** *Suppose the model (1) follows the process described in (16) with $Y_t = X_{t+1}$. Assume that the process (16) satisfies Assumption 6.1. Fix a $B \geq 1$, and suppose that $\|A_\star\|_F \leq B$. Consider the hypothesis class and true regression function:*

$$\mathscr{F} \triangleq \{f(x) = \sigma(Ax) \mid A \in \mathbb{R}^{d_{\mathsf{X}} \times d_{\mathsf{X}}}, \ \|A\|_F \leq B\}, \ f_\star(x) = \sigma(A_\star x). \tag{17}$$

*There exists a $T_0$ and a universal positive constant $c_0$ such that the LSE with hypothesis class $\mathscr{F}$ achieves for all $T \geq T_0$:*

$$\mathbf{E}\|\widehat{f} - f_\star\|_{L^2}^2 \leq c_0 \frac{\|H\|_{\mathsf{op}}^2 d_{\mathsf{X}}^2}{T} \log\left( \max\left\{ T, B, d_{\mathsf{X}}, \|P_\star\|_{\mathsf{op}}, \|H\|_{\mathsf{op}}, \frac{1}{1-\rho} \right\} \right). \tag{18}$$

*Furthermore, for a universal constant $c_1 > 0$, we may choose $T_0$ that satisfies:*

$$T_0 = c_1 \max \left( \frac{\|P_\star\|_{\mathsf{op}}^2 \mathrm{cond}(H)^4 d_{\mathsf{X}}^4}{\zeta^4 (1-\rho)^6}, \frac{1}{\|H\|_{\mathsf{op}}^{1/3}} \right) \mathrm{polylog} \left( B, d_{\mathsf{X}}, \|P_\star\|_{\mathsf{op}}, \mathrm{cond}(H), \frac{1}{\zeta}, \frac{1}{1-\rho} \right).$$

(19)

Further discussion regarding Assumption 6.1 and Theorem 6.2, including a more detailed comparison with existing results, can be found in Appendix J.3.

## 7 Conclusion

We developed a framework for showing when the mixing-time of the covariates plays a relatively small role in the rate of convergence of the least-squares estimator. In many situations, after a finite burn-in time, this learning procedure exhibits an excess risk that scales as if all the samples were independent (Theorem 4.1). As a byproduct of our framework, by instantiating our results to system identification for dynamics with generalized linear model transitions (Section 6.2), we derived the sharpest known excess risk rate for this problem; our rates are nearly minimax optimal after only a polynomial burn-in time.

To arrive at Theorem 4.1, we leveraged insights from Mendelson [3] via a one-sided concentration inequality (Theorem B.2). As mentioned in Section 4.1, hypercontractivity is closely related to the small-ball condition [3]. Such conditions can be understood as quantitative identifiability conditions by providing control of the "version space" (cf. Mendelson [3]). Given that identifiability conditions also play a key role in linear system identification—a setting in which a similar phenomenon as studied here had already been reported—this suggests an interesting direction for future work: are such conditions actually necessary for learning with little mixing?

## Acknowledgements

We thank Dheeraj Nagaraj for helpful discussions regarding the results in Nagaraj et al. [25], and Abhishek Roy for clarifying the results in Roy et al. [13]. We also thank Mahdi Soltanolkotabi for an informative discussion about the computational aspects of empirical risk minimization for generalized linear models under the square loss.

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
