# Contents

# A Numerical experiments

We conduct a simple numerical simulation to illustrate the phenomenon of learning with little mixing empirically. We consider system identification of the GLM dynamics described in Section 6.2.

We first describe how the covariate process $\{X_t\}$ is generated. We set $d_X = 25$. The true dynamic matrix $A_\star$ is randomly sampled from the distribution described in Section 7 of Kowshik et al. [20]. Specifically, $A_\star = U\Sigma U^\mathsf{T}$, where $U$ is uniform from the Haar measure on the space of orthonormal $d_X \times d_X$ matrices, and $\Sigma = \mathrm{diag}(\underbrace{\rho, \ldots, \rho}_{\lfloor d_X/2 \rfloor \text{ times}}, \rho/3, \ldots, \rho/3)$. We vary $\rho \in \{0.9, 0.99\}$ for this experiment. Next, we set the activation function $\sigma$ to be the LeakyReLU with slope 0.5, i.e., $\sigma(x) = 0.5x\mathbf{1}\{x < 0\} + x\mathbf{1}\{x \geq 0\}$. Observe that these dynamics satisfy Assumption 6.1 with $\zeta = 0.5$, and where the Lyapunov matrix $P$ can be taken to be identity, since $\|A_\star\|_{\mathsf{op}} = \rho < 1$. Next, we generate $X_0 \sim N(0, I_{d_X})$, and $X_{t+1} = \sigma(A_\star X_t) + W_t$ with $W_t \sim N(0, 0.01 I_{d_X})$ and $W_t \perp W_{t'}$ for $t \neq t'$. From this trajectory, the labelled dataset is $\{(X_t, Y_t)\}_{t=0}^{T-1}$ with $Y_t = X_{t+1}$.

To study the effects of the correlation from a single trajectory $\{X_t\}$ for learning, we consider the following *independent baseline* motivated by the Ind-Seq-LS baseline described in Tu et al. [19]. Let $\mu_t$ denote the marginal distribution of $X_t$. We sample $\bar{X}_t \sim \mu_t$ independently across time, and sample $\bar{Y}_t \mid \bar{X}_t$ from the conditional distribution $N(\sigma(A_\star \bar{X}_t), 0.01 I_{d_X})$; the labelled dataset is $\{(\bar{X}_t, \bar{Y}_t)\}_{t=0}^{T-1}$. This ensures that the $L^2$ risk of a fixed hypothesis $f(x) = \sigma(Ax)$ is the same under both the independent baseline and the single trajectory distribution, so that our experiment singles out the effect of learning from correlated data. In practice, each $\bar{X}_t$ is sampled from a new independent rollout up to time $t$.

Given a dataset $\{(X_t, Y_t)\}_{t=0}^{T-1}$, we search for the empirical risk minimizer (ERM) of the loss

$$\hat{A} = \mathrm{argmin}_{A \in \mathbb{R}^{d_X \times d_X}} \left\{ \frac{1}{T} \sum_{t=0}^{T-1} \|\sigma(AX_t) - Y_t\|_2^2 \right\} \tag{20}$$

by running `scipy.optimize.minimize` with the `L-BFGS-B` method, using the default linesearch and termination criteria options. To calculate the $L^2$ excess risk $\frac{1}{T}\sum_{t=0}^{T-1} \mathbf{E}\|\sigma(\hat{A}X_t) - \sigma(A_\star X_t)\|_2^2$ of a hypothesis $\hat{A}$, we draw 1000 new trajectories and average the excess risk over these trajectories. The experimental code is implemented with `jax` [38], and run using the CPU backend with `float64` precision on a single machine.[4]

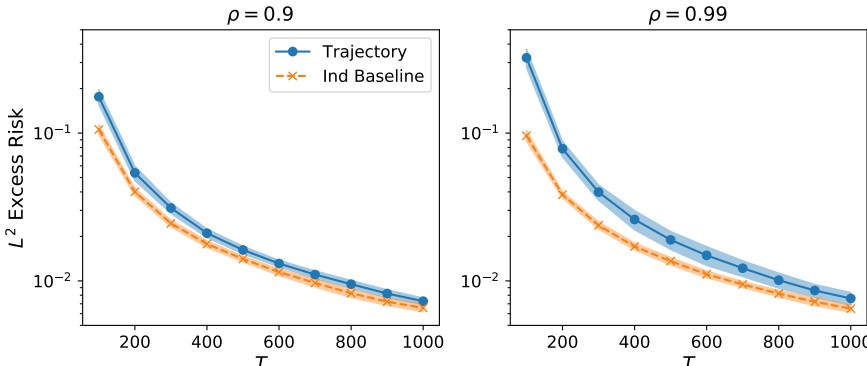

Figure 1: $L^2$ excess risk as a function of dataset length $T$ of the empirical risk minimizer on the single trajectory (Trajectory) dataset versus the independent baseline (Ind Baseline) dataset.

The results of this experiment are shown in Figure 1 and Figure 2. In Figure 1, we plot the $L^2$ excess risk of the ERM $\hat{A}$ from (20) on both the trajectory dataset $\{(X_t, Y_t)\}$ and the independent baseline dataset $\{(\bar{X}_t, \bar{Y}_t)\}$, varying $\rho \in \{0.9, 0.99\}$. The shaded region indicates $\pm$ one standard deviation from the mean over 20 training datasets. In Figure 2, we plot the $L^2$ excess risk *ratio* of the estimator

---

[4]Code available at: https://github.com/google-research/google-research/tree/master/learning_with_little_mixing

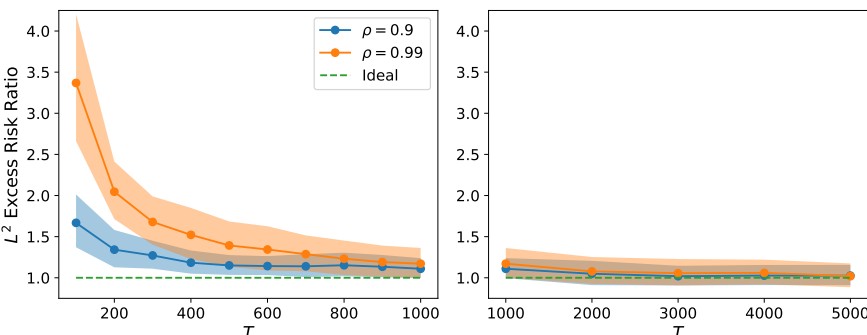

Figure 2: Ratio of the $L^2$ excess risk as a function of dataset length $T$ of the empirical risk minimizer (ERM) on the single trajectory dataset over the ERM on the independent baseline dataset. The dashed green curve (Ideal) marks a ratio of exactly one.

$\hat{A}$ from the single trajectory dataset over the estimator $\hat{A}$ from the independent baseline trajectory, again varying $\rho \in \{0.9, 0.99\}$. Here, the shaded region is constructed using $\pm$ one standard deviation of the numerator and denominator taken over 20 training datasets.

Figure 1 and Figure 2 illustrate two different trends, which are both predicted by our theory. First, for a fixed $\rho$, as $T$ increases, the $L^2$ excess risk of the ERM on the trajectory dataset approaches that of the ERM on the independent dataset. This illustrates the learning with little mixing phenomenon, where despite correlations in the covariates $\{X_t\}$ of the trajectory dataset across time, the statistical behavior of the ERM approaches that of the ERM on the independent dataset where the covariates $\{\bar{X}_t\}$ are independent across time. Next, for a fixed $T$, the burn-in time increases as $\rho$ approaches one. That is, systems that mix slower have longer burn-in times.

## B   Proof techniques

In this section, we highlight the main ideas behind the proof of Theorem 4.1. The full details can be found in the appendix. We start with a key insight from Mendelson [3]: establishing a one-sided inequality between the empirical versus true risk is substantially easier than the corresponding two-sided inequality. Recall that the closed ball of radius $r$ is $B(r) = \left\{ f \in \mathscr{F}_\star \mid \frac{1}{T} \sum_{t=0}^{T-1} \mathbf{E}\|f(X_t)\|_2^2 \leq r^2 \right\}$. We identify conditions which depend mildly on $\|\Gamma_{\mathsf{dep}}(\mathsf{P}_X)\|_{\mathsf{op}}$, so that with high probability we have:

$$\forall f \in \mathscr{F}_\star \setminus B(r), \ \ \mathbf{E}\left[ \frac{1}{T} \sum_{t=0}^{T-1} \|f(X_t)\|_2^2 \right] \lesssim \frac{1}{T} \sum_{t=0}^{T-1} \|f(X_t)\|_2^2. \tag{21}$$

Once this *lower isometry* condition (21) holds, we bound the empirical excess risk, the RHS of inequality (21), by a version of the basic inequality of least squares [15, 24]. This leads to an upper bound of the empirical excess risk by the martingale complexity term (Definition 4.3). As our innovation mainly lies in establishing the lower isometry condition (21), we focus on this component for the remainder of the proof outline.

**Lower isometry**   The key tool we use is the following exponential inequality, which controls the lower tail of sums of non-negative dependent random variables via the dependency matrix $\Gamma_{\mathsf{dep}}(\mathsf{P}_X)$.

**Theorem B.1** (Samson [26, Theorem 2]). *Let $g : \mathsf{X} \to \mathbb{R}$ be non-negative. Then for any $\lambda > 0$:*

$$\mathbf{E} \exp\left( -\lambda \sum_{t=0}^{T-1} g(X_t) \right) \leq \exp\left( -\lambda \sum_{t=0}^{T-1} \mathbf{E}g(X_t) + \frac{\lambda^2 \|\Gamma_{\mathsf{dep}}(\mathsf{P}_X)\|_{\mathsf{op}}^2 \sum_{t=0}^{T-1} \mathbf{E}g^2(X_t)}{2} \right). \tag{22}$$

We note that Samson [26, Theorem 2] actually proves a much stronger statement than Theorem B.1 (a Talagrand-style uniform concentration inequality), from which Theorem B.1 is a byproduct. With Theorem B.1 in hand, the following proposition allows us to relate hypercontractivity to lower isometry.

**Proposition B.1.** *Fix $C > 0$ and $\alpha \in [1, 2]$. Let $g : \mathsf{X} \to \mathbb{R}$ be a non-negative function satisfying*

$$\mathbf{E}\left[\frac{1}{T}\sum_{t=0}^{T-1} g^2(X_t)\right] \le C\left(\mathbf{E}\left[\frac{1}{T}\sum_{t=0}^{T-1} g(X_t)\right]\right)^{\alpha}. \tag{23}$$

*Then we have:*

$$\mathbf{P}\left(\sum_{t=0}^{T-1} g(X_t) \le \frac{1}{2}\sum_{t=0}^{T-1}\mathbf{E}g(X_t)\right) \le \exp\left(-\frac{T}{8C\|\Gamma_{\mathsf{dep}}(\mathsf{P}_X)\|_{\mathsf{op}}^2}\left(\frac{1}{T}\sum_{t=0}^{T-1}\mathbf{E}g(X_t)\right)^{2-\alpha}\right).$$

Now fix an $f \in \mathscr{F}_\star \setminus B(r)$, and put $g(x) = \|f(x)\|_2^2$. Substituting $g$ into (23) yields the trajectory hypercontractivity condition (3) in Definition 4.1. Thus, Proposition B.1 establishes the lower isometry condition (21) for any fixed function. Hence, it remains to take a union bound over a supremum norm cover of $\mathscr{F}_\star \setminus B(r)$ at resolution $r$. It turns out that it suffices to instead cover the boundary $\partial B(r)$ since $\mathscr{F}_\star$ is star-shaped. Carrying out these details leads to the main lower isometry result.

**Theorem B.2.** *Fix constants $\alpha \in [1, 2]$ and $C, r > 0$. Let $\mathscr{F}_\star$ be star-shaped, and suppose that there exists a $r/\sqrt{8}$-net $\mathscr{F}_r$ of $\partial B(r)$ in the $\|\cdot\|_\infty$-norm such that $(\mathscr{F}_r, \mathsf{P}_X)$ satisfies the trajectory $(C, \alpha)$-hypercontractivity condition. Then the following lower isometry holds:*

$$\mathbf{P}\left(\sup_{f \in \mathscr{F}_\star \setminus B(r)}\left\{\frac{1}{T}\sum_{t=0}^{T-1}\|f(X_t)\|_2^2 - \mathbf{E}\frac{1}{8T}\sum_{t=0}^{T-1}\|f(X_t)\|_2^2\right\} \le 0\right) \le |\mathscr{F}_r|\exp\left(\frac{-Tr^{4-2\alpha}}{8C\|\Gamma_{\mathsf{dep}}(\mathsf{P}_X)\|_{\mathsf{op}}^2}\right).$$

### B.1 Handling unbounded trajectories

Our main result Theorem 4.1 requires boundedness of both the hypothesis class $\mathscr{F}$ and the covariate process $\{X_t\}$ to hold. However, when this does not hold, Theorem 4.1 can often still be applied via a careful truncation argument. In this section, we outline the key ideas of this argument, with the full details given in Appendix G.

For concreteness, let us consider a Markovian process driven by Gaussian noise. Let $\{W_t\}_{t \ge 0}$ and $\{W_t'\}_{t \ge 0}$ be sequences of iid $N(0, I)$ vectors in $\mathbb{R}^{d_\mathsf{x}}$. Fix a dynamics function $f : \mathbb{R}^{d_\mathsf{x}} \to \mathbb{R}^{d_\mathsf{x}}$ and truncation radius $R > 0$. Define the truncated noise process $\{\bar{W}_t\}_{t \ge 0}$ as $\bar{W}_t \triangleq W_t'\mathbf{1}\{\|W_t'\|_2 \le R\}$, and denote the original process and its truncated process by:

$$X_{t+1} = f(X_t) + HW_t, \ X_0 = HW_0, \qquad \text{(original process)} \tag{24a}$$
$$\bar{X}_{t+1} = f(\bar{X}_t) + H\bar{W}_t, \ \bar{X}_0 = H\bar{W}_0. \qquad \text{(truncated process)} \tag{24b}$$

Setting $R$ appropriately, it is clear that the original process (24a) coincides with the truncated process (24b) with high probability by standard Gaussian concentration inequalities. Furthermore, the truncated noise process $\{H\bar{W}_t\}$ remains a martingale difference sequence due to the symmetry of the truncation. Additionally, since $\{H\bar{W}_t\}$ is bounded, if $f$ is appropriately Lypaunov stable then the process $\{\bar{X}_t\}$ becomes bounded. In turn any class $\mathscr{F}$ containing continuous functions is bounded as well on (24b). Hence, the LSE $\hat{f}$ on (24a) can be controlled by the LSE $\bar{f}$ on (24b).

So far, this is a straightforward reduction. However, a subtle point arises in applying Theorem 4.1 to the LSE $\bar{f}$ on (24b): the dependency matrix $\Gamma_{\mathsf{dep}}$ now involves the truncated process (24b) instead of the original process (24a). This is actually *necessary* for this strategy to work, as the supremum in the dependency matrix coefficients (4) is now over the truncated process $\{\bar{X}_t\}$, instead of the original process $\{X_t\}$ which is unbounded. However, there is a trade-off, as bounding the coefficients for $\{\bar{X}_t\}$ is generally more complex than for $\{X_t\}$.[5] Nevertheless, a coupling argument allows us to switch back to bounding the dependency matrix coefficients for $\{X_t\}$, but crucially keep the supremum over the truncated $\{\bar{X}_t\}$. This reduction substantially broadens the scope of Theorem 4.1 without any modification to the proof.

---

[5]The clearest example of this is when the dynamics function $f$ is linear: in this case, $\{X_t\}$ is jointly Gaussian (and hence (4) can be bounded by closed-form expressions), whereas $\{\bar{X}_t\}$ is not due to the truncation operator.

## C  Proof of Theorem 4.1

### C.1  Proof of Lemma B.1

Let us abbreviate $\Gamma = \Gamma_{\mathsf{dep}}(\mathsf{P}_X)$. A Chernoff argument yields

$$\mathbf{P}\left(\sum_{t=0}^{T-1} g(X_t) \leq \frac{1}{2}\sum_{t=0}^{T-1}\mathbf{E}g(X_t)\right)$$

$$\leq \inf_{\lambda \geq 0}\mathbf{E}\exp\left(\frac{\lambda}{2}\sum_{t=0}^{T-1}\mathbf{E}g(X_t) - \lambda\sum_{t=0}^{T-1}g(X_t)\right) \qquad \text{(Chernoff)}$$

$$\leq \inf_{\lambda \geq 0}\exp\left(-\frac{\lambda}{2}\sum_{t=0}^{T-1}\mathbf{E}g(X_t) + \frac{\lambda^2\|\Gamma\|_{\mathsf{op}}^2\sum_{t=0}^{T-1}\mathbf{E}g^2(X_t)}{2}\right) \qquad \text{(Proposition B.1)}$$

$$= \exp\left(-\frac{\left(\sum_{t=0}^{T-1}\mathbf{E}g(X_t)\right)^2}{8\|\Gamma\|_{\mathsf{op}}^2\sum_{t=0}^{T-1}\mathbf{E}g^2(X_t)}\right) \qquad \left(\lambda = \frac{\sum_{t=0}^{T-1}\mathbf{E}g(X_t)}{2\|\Gamma\|_{\mathsf{op}}^2\sum_{t=0}^{T-1}\mathbf{E}g^2(X_t)}\right)$$

$$\leq \exp\left(-\frac{T}{8C\|\Gamma\|_{\mathsf{op}}^2} \times \left(\frac{1}{T}\sum_{t=0}^{T-1}\mathbf{E}g(X_t)\right)^{2-\alpha}\right), \qquad \text{(Using (23))}$$

as per requirement. ∎

### C.2  Proof of Theorem B.2

The hypothesis that $\mathscr{F}_\star$ is star-shaped allows us to rescale, so it suffices to prove the result for $f \in \partial B(r)$. Namely, if $f \in \mathscr{F}_\star \setminus B(r)$ then $\frac{r}{\|f\|_{L^2}} < 1$ and so $rf/\|f\|_{L^2} \in \partial B(r)$ by the star-shaped hypothesis. Recall that $\mathscr{F}_r \subset \partial B(r)$ is a $r/\sqrt{8}$-net of $\partial B(r)$ in the supremum norm. Hence, by construction and parallellogram, for every $f \in \partial B(r)$, there exists $f_i \in \mathscr{F}_r$ such that:

$$\frac{1}{T}\sum_{t=0}^{T-1}\|f(X_t)\|_2^2 \geq \frac{1}{2T}\sum_{t=0}^{T-1}\|f_i(X_t)\|_2^2 - \frac{r^2}{8}. \tag{25}$$

Define the event:

$$\mathcal{E} \triangleq \bigcup_{f \in \mathscr{F}_r}\left\{\frac{1}{T}\sum_{t=0}^{T-1}\|f(X_t)\|_2^2 \leq \mathbf{E}\frac{1}{2T}\sum_{t=0}^{T-1}\|f(X_t)\|_2^2\right\}.$$

Invoking Lemma B.1 with $g(x) = \|f(x)\|_2^2$ for $f \in \mathscr{F}_r$, by a union bound it clear that

$$\mathbf{P}(\mathcal{E}) \leq |\mathscr{F}_r|\exp\left(\frac{-Tr^{4-2\alpha}}{8C\|\Gamma_{\mathsf{dep}}(\mathsf{P}_X)\|_{\mathsf{op}}^2}\right). \tag{26}$$

Fix now arbitrary $f \in \partial B(r)$. On the complement $\mathcal{E}^c$ it is true that

$$\frac{1}{T}\sum_{t=0}^{T-1}\|f(X_t)\|_2^2 \geq \frac{1}{2T}\sum_{t=0}^{T-1}\|f_i(x_t)\|_2^2 - \frac{r^2}{8} \qquad \text{(we may find such an } f_i \text{ by observation (25))}$$

$$\geq \mathbf{E}\frac{1}{4T}\sum_{t=0}^{T-1}\|f_i(X_t)\|_2^2 - \frac{r^2}{8} \qquad \text{(by definition of } \mathcal{E})$$

$$= \frac{r^2}{4} - \frac{r^2}{8} \qquad (f_i \in \partial B(r))$$

$$\geq \frac{r^2}{8}.$$

Since $f \in \partial B(r)$ was arbitrary, by virtue of the estimate (26) we have that:

$$\mathbf{P}\left(\sup_{f \in \partial B(r)}\left\{\frac{1}{T}\sum_{t=0}^{T-1}\|f(X_t)\|_2^2 - \frac{r^2}{8}\right\} \leq 0\right) \leq |\mathscr{F}_r|\exp\left(\frac{-Tr^{4-2\alpha}}{8C\|\Gamma_{\mathsf{dep}}(\mathsf{P}_X)\|_{\mathsf{op}}^2}\right).$$

The result follows by rescaling. ■

### C.3 Proof of Theorem 4.1

Define the event:

$$\mathcal{B}_r \triangleq \left(\sup_{f \in \mathscr{F}_\star \setminus B(r)}\left\{\frac{1}{T}\sum_{t=0}^{T-1}\|f(X_t)\|_2^2 - \mathbf{E}\frac{1}{8T}\sum_{t=0}^{T-1}\|f(X_t)\|_2^2\right\} \leq 0\right).$$

By definition, on the complement of $\mathcal{B}_r$ we have that:

$$\|\widehat{f} - f_\star\|_{L^2}^2 \leq r^2 \vee \frac{8}{T}\sum_{t=0}^{T-1}\|\widehat{f}(X_t) - f_\star(X_t)\|_2^2 \leq r^2 + \frac{8}{T}\sum_{t=0}^{T-1}\|\widehat{f}(X_t) - f_\star(X_t)\|_2^2. \tag{27}$$

Therefore, we can decompose $\mathbf{E}\|\widehat{f} - f_\star\|_{L^2}^2$ as follows:

$$\mathbf{E}\|\widehat{f} - f_\star\|_{L^2}^2 = \mathbf{E}\mathbf{1}_{\mathcal{B}_r}\|\widehat{f} - f_\star\|_{L^2}^2 + \mathbf{E}\mathbf{1}_{\mathcal{B}_r^c}\|\widehat{f} - f_\star\|_{L^2}^2$$

$$\leq B^2\mathbf{P}(\mathcal{B}_r) + r^2 + 8\mathbf{E}\left[\frac{1}{T}\sum_{t=0}^{T-1}\|\widehat{f}(X_t) - f_\star(X_t)\|_2^2\right]. \quad (B\text{-bdd \& ineq. (27)})$$

$$\tag{28}$$

Theorem B.2 informs us that:

$$\mathbf{P}(\mathcal{B}_r) \leq |\mathscr{F}_r|\exp\left(\frac{-Tr^{4-2\alpha}}{8C\|\Gamma_{\mathsf{dep}}(\mathsf{P}_X)\|_{\mathsf{op}}^2}\right). \tag{29}$$

On the other hand, we have by the basic inequality (as in [24]):

$$\frac{1}{T}\sum_{t=0}^{T-1}\|\widehat{f}(X_t) - f_\star(X_t)\|_2^2 \leq \frac{1}{T}\sup_{f \in \mathscr{F}_\star}\sum_{t=0}^{T-1}4\langle W_t, f(X_t)\rangle - \|f(X_t)\|_2^2 \tag{30}$$

Combining inequalities (28), (29) and (30) we conclude:

$$\mathbf{E}\|\widehat{f} - f_\star\|_{L^2}^2 \leq 8\mathbf{E}\left[\sup_{f \in \mathscr{F}_\star}\frac{1}{T}\sum_{t=0}^{T-1}4\langle W_t, f(X_t)\rangle - \|f(X_t)\|_2^2\right]$$

$$+ r^2 + B^2|\mathscr{F}_r|\exp\left(\frac{-Tr^{4-2\alpha}}{8C\|\Gamma_{\mathsf{dep}}(\mathsf{P}_X)\|_{\mathsf{op}}^2}\right),$$

as per requirement. ■

## D Proofs for corollaries in Section 4

### D.1 Proof of Corollary 4.1

We set $r^2 = \frac{1}{T^{\frac{2}{2+q}+\gamma}}$. We first use Vershynin [39, Exercise 4.2.10] followed by (8) to bound:

$$\log\mathcal{N}_\infty(\partial B(r), r/\sqrt{8}) \leq \log\mathcal{N}_\infty(\mathscr{F}_\star, r/(2\sqrt{8})) \leq p\left(\frac{2\sqrt{8}}{r}\right)^q.$$

Therefore:

$$B^2\mathcal{N}_\infty(\partial B(r), r/\sqrt{8})\exp\left(\frac{-T}{8C\|\Gamma_{\mathsf{dep}}(\mathsf{P}_X)\|_{\mathsf{op}}^2}\right) \leq B^2\exp\left(32pT^{\frac{q}{2+q}+\frac{q\gamma}{2}} - \frac{T}{8C\|\Gamma_{\mathsf{dep}}(\mathsf{P}_X)\|_{\mathsf{op}}^2}\right).$$

We want to solve for $T$ such that:

$$B^2 \exp\left(32pT^{\frac{q}{2+q}+\frac{q\gamma}{2}} - \frac{T}{8C\|\Gamma_{\mathsf{dep}}(\mathsf{P}_X)\|_{\mathsf{op}}^2}\right) \leq \frac{1}{T^{\frac{2}{2+q}+\gamma}}.$$

To do this, we first require that:

$$32pT^{\frac{q}{2+q}+\frac{q\gamma}{2}} - \frac{T}{8C\|\Gamma_{\mathsf{dep}}(\mathsf{P}_X)\|_{\mathsf{op}}^2} \leq -T^{\frac{q}{2+q}+\frac{q\gamma}{2}} \iff T^{1-\left(\frac{q}{2+q}+\frac{q\gamma}{2}\right)} \geq 8(32p+1)C\|\Gamma_{\mathsf{dep}}(\mathsf{P}_X)\|_{\mathsf{op}}^2$$

$$\iff T \geq \left[8(32p+1)C\|\Gamma_{\mathsf{dep}}(\mathsf{P}_X)\|_{\mathsf{op}}^2\right]^{\frac{1}{1-\frac{q}{2}\left(\frac{2}{2+q}+\gamma\right)}}.$$

Now, with this requirement, we are left with the sufficient condition:

$$B^2 \exp\left(-T^{\frac{q}{2+q}+\frac{q\gamma}{2}}\right) \leq \frac{1}{T^{\frac{2}{2+q}+\gamma}} \iff T^{\frac{q}{2+q}+\frac{q\gamma}{2}} \geq \log(B^2 T^{\frac{2}{2+q}+\gamma}).$$

It suffices to require that:

$$T^{\frac{q}{2+q}+\frac{q\gamma}{2}} \geq 4\log B \iff T \geq (4\log B)^{\frac{1}{\frac{q}{2}\left(\frac{2}{2+q}+\gamma\right)}},$$

$$T^{\frac{q}{2+q}+\frac{q\gamma}{2}} \geq 2\log(T^{\frac{2}{2+q}+\gamma}) = \frac{4}{q}\log(T^{\frac{q}{2+q}+\frac{q\gamma}{2}}).$$

By Simchowitz et al. [16, Lemma A.4], the bottom inequality holds when:

$$T^{\frac{q}{2+q}+\frac{q\gamma}{2}} \geq \frac{8}{q}\log\left(\frac{16}{q}\right).$$

The claim now follows from Theorem 4.1. ∎

## D.2   Proof of Corollary 4.2

We set $r^2 = 1/T^{1+\gamma}$. By the given assumptions, we can construct a $r/\sqrt{8}$-net $\mathscr{F}_r$ of $\partial B(r)$ in the $\|\cdot\|_\infty$-norm that (a) satisfies

$$\log|\mathscr{F}_r| \leq p\log^q\left(\frac{\sqrt{8}}{r}\right),$$

and (b) satisfies the trajectory $(C(r/\sqrt{8}), \alpha)$-hypercontractivity condition. Recalling the bounds $\|\Gamma_{\mathsf{dep}}(\mathsf{P}_X)\|_{\mathsf{op}}^2 \leq T^{b_1}$ and $C(r) \leq (1/r)^{b_2}$, we have:

$$B^2|\mathscr{F}_r|\exp\left(\frac{-Tr^{4-2\alpha}}{8C(r/\sqrt{8})\|\Gamma_{\mathsf{dep}}(\mathsf{P}_X)\|_{\mathsf{op}}^2}\right) \leq B^2\exp\left\{p\log^q\left(\frac{\sqrt{8}}{r}\right) - \frac{T^{1-b_1}r^{4-2\alpha+b_2}}{8^{1+b_2/2}}\right\}$$

$$= B^2\exp\left\{p\log^q\left(\sqrt{8}T^{\frac{1+\gamma}{2}}\right) - \frac{T^{1-b_1-\frac{(1+\gamma)(4-2\alpha+b_2)}{2}}}{8^{1+b_2/2}}\right\}$$

$$= B^2\exp\left\{p\log^q\left(\sqrt{8}T^{\frac{1+\gamma}{2}}\right) - \frac{T^\psi}{8^{1+b_2/2}}\right\}$$

$$\leq B^2\exp\left\{p\log^q\left(\sqrt{8}T^{\frac{1+\gamma}{2}}\right) - \frac{T^\psi}{64}\right\}.$$

Above, the last inequality holds since $b_2 < 2$. Now, we choose $T$ large enough so that:

$$p\log^q\left(\sqrt{8}T^{\frac{1+\gamma}{2}}\right) - \frac{T^\psi}{64} \leq -\frac{T^\psi}{128} \iff T^\psi \geq 128p\log^q\left(\sqrt{8}T^{\frac{1+\gamma}{2}}\right)$$

$$\iff T^{\psi/q} \geq (128p)^{1/q}\left(\log(\sqrt{8}) + \frac{1+\gamma}{2\psi/q}\log(T^{\psi/q})\right).$$

Thus, it suffices to require that:

$$T^{\psi/q} \geq (128p)^{1/q}\log 8, \quad T^{\psi/q} \geq (128p)^{1/q}\frac{1+\gamma}{\psi/q}\log(T^{\psi/q}).$$

By Simchowitz et al. [16, Lemma A.4], the right hand side inequality holds when:

$$T^{\psi/q} \geq 2(128p)^{1/q}\frac{1+\gamma}{\psi/q}\log\left(4(128p)^{1/q}\frac{1+\gamma}{\psi/q}\right).$$

We finish the proof by finding $T$ such that:

$$B^2\exp\left(\frac{-T^\psi}{128}\right) \leq \frac{1}{T^{1+\gamma}} \iff T^\psi \geq 128\log(B^2 T^{1+\gamma})$$

$$\iff T^\psi \geq 256\log B + 128\frac{(1+\gamma)}{\psi}\log(T^\psi).$$

Thus, it suffices to require that:

$$T^\psi \geq 512\log B, \ \ T^\psi \geq 256\frac{(1+\gamma)}{\psi}\log(T^\psi).$$

Another application of Simchowitz et al. [16, Lemma A.4] yields that the latter inequality holds if:

$$T^\psi \geq 512\frac{1+\gamma}{\psi}\log\left(1024\frac{1+\gamma}{\psi}\right).$$

Combining all our requirements on $T$, we require that $T \geq \max\{T_1, T_2\}$, with:

$$T_1 \triangleq \max\left\{(128p)^{1/\psi}(\log 8)^{q/\psi}, (128p)^{1/\psi}\left[\frac{4q}{\psi}\log\left((128p)^{1/q}\frac{8q}{\psi}\right)\right]^{q/\psi}\right\},$$

$$T_2 \triangleq \max\left\{(512\log B)^{1/\psi}, \left[\frac{1024}{\psi}\log\left(\frac{2056}{\psi}\right)\right]^{1/\psi}\right\}.$$

∎

# E  Proofs for Section 5

## E.1  Proof of Proposition 5.1

For notational brevity, we make the identification of the atoms $\{\psi_1, \ldots, \psi_K\}$ with the integers $\{1, \ldots, K\}$. Fix a function $f : \{1, \ldots, K\} \to \mathbb{R}^{d_Y}$. For any time indices $t_1, t_2 \in \{0, \ldots, T-1\}$:

$$\mathbf{E}\|f(X_{t_1})\|_2^2 \mathbf{E}\|f(X_{t_2})\|_2^2 = \left(\sum_{k=1}^K \|f(k)\|_2^2\mu_{t_1}(k)\right)\left(\sum_{k=1}^K \|f(k)\|_2^2\mu_{t_2}(k)\right)$$

$$= \sum_{k_1=1}^K\sum_{k_2=1}^K \|f(k_1)\|_2^2\|f(k_2)\|_2^2\mu_{t_1}(k_1)\mu_{t_2}(k_2) \geq \sum_{k_1=1}^K \|f(k_1)\|_2^4\mu_{t_1}(k_1)\mu_{t_2}(k_1)$$

$$\geq \underline{\mu}\sum_{k_1=1}^K \|f(k_1)\|_2^4\mu_{t_1}(k_1) = \underline{\mu}\mathbf{E}\|f(X_{t_1})\|_2^4.$$

Therefore:

$$\left(\frac{1}{T}\sum_{t=0}^{T-1}\mathbf{E}\|f(X_t)\|_2^2\right)^2 = \frac{1}{T^2}\sum_{t_1=0}^{T-1}\sum_{t_2=0}^{T-1}\mathbf{E}\|f(X_{t_1})\|^2\mathbf{E}\|f(X_{t_2})\|_2^2$$

$$\geq \frac{\underline{\mu}}{T^2}\sum_{t_1=0}^{T-1}\sum_{t_2=0}^{T-1}\mathbf{E}\|f(X_{t_1})\|_2^4 = \frac{\underline{\mu}}{T}\sum_{t_1=0}^{T-1}\mathbf{E}\|f(X_{t_1})\|_2^4.$$

The claim now follows since we assume $\underline{\mu} > 0$. ∎

## E.2   Proof of Proposition 5.2

Recall that $\mathbf{E}\left[\frac{1}{T}\sum_{t=0}^{T-1}\|f(X_t)\|_2^p\right] = \|f\|_{L^p}^p$ for $p \geq 1$. We estimate the left hand side of inequality (3) as follows:

$$
\begin{aligned}
\mathbf{E}\left[\frac{1}{T}\sum_{t=0}^{T-1}\|f(X_t)\|_2^4\right] &= \mathbf{E}\left[\frac{1}{T}\sum_{t=0}^{T-1}\|f(X_t)\|_2^{2-\varepsilon}\|f(X_t)\|_2^{2+\varepsilon}\right] \\
&\leq B^{2-\varepsilon}\mathbf{E}\left[\frac{1}{T}\sum_{t=0}^{T-1}\|f(X_t)\|_2^{2+\varepsilon}\right] &&(B\text{-bounded}) \\
&= B^{2-\varepsilon}\|f\|_{L^{2+\varepsilon}}^{2+\varepsilon} \\
&\leq B^{2-\varepsilon}(c\|f\|_{L^2})^{2+\varepsilon} &&(L^2 - L^{2+\varepsilon}\text{-equivalence}) \\
&= B^{2-\varepsilon}c^{2+\varepsilon}\|f\|_{L^2}^{2+\varepsilon} \\
&= B^{2-\varepsilon}c^{2+\varepsilon}\left(\mathbf{E}\left[\frac{1}{T}\sum_{t=0}^{T-1}\|f(X_t)\|_2^2\right]\right)^{1+\varepsilon/2}.
\end{aligned}
$$

The result now follows. ∎

## E.3   Proof of Proposition 5.3

We first state an auxiliary proposition.

**Proposition E.1.** *Let $\mu$ and $\nu$ be distributions satisfying $\mu \ll \nu$. Let $g$ be any measurable function such that $\mathbf{E}_\nu g^2 < \infty$. We have:*

$$
\mathbf{E}_\mu g - \mathbf{E}_\nu g \leq \sqrt{\mathbf{E}_\nu g^2}\sqrt{\chi^2(\mu, \nu)}.
$$

*Proof.* By Cauchy-Schwarz:

$$
\mathbf{E}_\mu g - \mathbf{E}_\nu g = \int g\left(\frac{d\mu}{d\nu} - 1\right)d\nu \leq \sqrt{\int g^2 d\nu}\sqrt{\int\left(\frac{d\mu}{d\nu} - 1\right)^2 d\nu} = \sqrt{\mathbf{E}_\nu g^2}\sqrt{\chi^2(\mu, \nu)}.
$$

∎

We can now complete the proof of Proposition 5.3. Fix any $f \in \mathscr{F}_\star$. First, we note that the the condition (5.3) implies:

$$
\mathbf{E}_\pi\|f\|_2^4 \leq (\mathbf{E}_\pi\|f\|_2^8)^{1/2} \leq (C_{8\to 2}(\mathbf{E}_\pi\|f\|_2^2)^4)^{1/2} = \sqrt{C_{8\to 2}}(\mathbf{E}_\pi\|f\|_2^2)^2. \tag{31}
$$

Therefore, for any $f \in \mathscr{F}_\star$ and any $t \in \mathbb{N}$:

$$
\begin{aligned}
\mathbf{E}\|f(X_t)\|_2^4 &\leq \mathbf{E}_\pi\|f\|_2^4 + \sqrt{\mathbf{E}_\pi\|f\|_2^8}\sqrt{\chi^2(\mu_t, \pi)} &&\text{using Proposition E.1} \\
&\leq (1 + \sqrt{C_{\chi^2}})\sqrt{C_{8\to 2}}(\mathbf{E}_\pi\|f\|_2^2)^2 &&\text{using (11), (5.3), and (31).} \tag{32}
\end{aligned}
$$

Now let $f \in \partial B(r)$. By Kuznetsov and Mohri [12, Lemma 1], since $\|f(x)\|_2^2 \in [0, B^2]$, we have:

$$
\mathbf{E}_\pi\|f\|_2^2 - \mathbf{E}\|f(X_t)\|_2^2 \leq B^2\|\mu_t - \pi\|_{\mathsf{TV}}. \tag{33}
$$

Therefore:

$$
\begin{aligned}
\mathbf{E}_\pi\|f\|_2^2 &= \frac{1}{T}\sum_{t=0}^{T-1}(\mathbf{E}_\pi\|f\|_2^2 - \mathbf{E}\|f(X_t)\|_2^2) + r^2 &&\text{since } f \in \partial B(r) \\
&\leq \frac{B^2}{T}\sum_{t=0}^{T-1}\|\mu_t - \pi\|_{\mathsf{TV}} + r^2 &&\text{using (33)} \\
&\leq (1 + C_{\mathsf{TV}}B^2)r^2 &&\text{using (11).} \tag{34}
\end{aligned}
$$

Combining these inequalities:

$$\frac{1}{T}\sum_{t=0}^{T-1}\mathbf{E}\|f(X_t)\|_2^4 \le (1+\sqrt{C_{\chi^2}})\sqrt{C_{8\to 2}}(\mathbf{E}_\pi\|f\|_2^2)^2 \qquad\qquad \text{using (32)}$$

$$\le (1+\sqrt{C_{\chi^2}})\sqrt{C_{8\to 2}}(1+C_{\mathsf{TV}}B^2)^2 r^4 \qquad\qquad \text{using (34)}$$

$$= (1+\sqrt{C_{\chi^2}})\sqrt{C_{8\to 2}}(1+C_{\mathsf{TV}}B^2)^2\left(\frac{1}{T}\sum_{t=0}^{T-1}\mathbf{E}\|f(X_t)\|_2^2\right)^2 \quad \text{since } f\in\partial B(r).$$

The claim now follows. ∎

### E.3.1   Further discussion related to Proposition 5.3

Let us discuss the ergodicity conditions in Proposition 5.3. The condition $\sup_{t\in\mathbb{N}}\chi^2(\mu_t,\pi)<\infty$ from (11) is quite mild. To illustrate this point, suppose that $\{X_t\}$ are regularly spaced samples in time from the Itô stochastic differential equation:

$$dZ_t = f(Z_t)\,dt + \sqrt{2}\,dB_t,$$

where $(B_t)$ is standard Brownian motion in $\mathbb{R}^{d_\mathsf{x}}$. Assume the process $(Z_t)$ admits a stationary distribution $\pi$, and let $\rho_t$ denote the measure of $Z_t$ at time $t$. A standard calculation [40, Theorem 4.2.5] shows that $\frac{d}{dt}\chi^2(\rho_t,\pi) = -2\mathbf{E}_\pi\left\|\nabla\left(\frac{\rho_t}{\pi}\right)\right\|_2^2 \le 0$, and hence $\sup_{t\ge 0}\chi^2(\rho_t,\pi)\le\chi^2(\rho_0,\pi)$. Thus, as long as the initial measure $\rho_0$ has finite divergence with $\pi$, then this condition holds. One caveat is that $\chi^2(\rho_0,\pi)$ can scale as $e^{d_\mathsf{x}}$, resulting in a hypercontractivity constant that scales exponentially in dimension. This however only affects the burn-in time and not the final rate.

The second condition in (11) is $\frac{1}{T}\sum_{t=0}^{T-1}\|\mu_t-\pi\|_{\mathsf{TV}}\lesssim r^2$. A typical setting is $r^2\asymp 1/T^\beta$ for some $\beta\in(0,1]$, where $\beta$ is dictated by the function class $\mathscr{F}$. Hence, this requirement reads:

$$\frac{1}{T}\sum_{t=0}^{T-1}\|\mu_t-\pi\|_{\mathsf{TV}}\lesssim\frac{1}{T^\beta} \iff \sum_{t=0}^{T-1}\|\mu_t-\pi\|_{\mathsf{TV}}\lesssim T^{1-\beta}.$$

Therefore, the setting of $\beta$ determines the level of ergodicty required. For example, if $\beta=1$ (which corresponds to the parametric function case), then this condition necessitates geometric ergodicity, since it requires that $\sum_{t=0}^{T-1}\|\mu_t-\pi\|_{\mathsf{TV}}=O(1)$. On the other hand, suppose that $\beta\in(0,1)$. Then this condition is satisfied if $\|\mu_t-\pi\|_{\mathsf{TV}}\lesssim 1/t^\beta$, allowing for slower mixing rates.

### E.4   Proof of Proposition 5.4

**Covering:**   We first approximate $\mathscr{P}$ by a finite-dimensional ellipsoid at resolution $\varepsilon/4$. To this end, fix an integer $m\in\mathbb{N}_+$ and define:

$$\mathscr{P}_m = \left\{f=\sum_{j=1}^m\theta_j\phi_j \,\middle|\, \sum_{j=1}^\infty\frac{\theta_j^2}{\mu_j}\le 1\right\}.$$

Fix now an element $f \in \mathscr{P}$ with coordinates $\theta$. Let $f'$ be the orthogonal projection onto the subspace of the first $m$-many coordinates ($f' \in \mathscr{P}_m$). Then:

$$
\begin{aligned}
\|f - f'\|_\infty = \left\| \sum_{j=m+1}^{\infty} \theta_j \phi_j \right\|_\infty &\leq \left\| \underbrace{\sqrt{\sum_{j=m+1}^{\infty} \frac{\theta_j^2}{\mu_j}}}_{\leq 1} \sqrt{\sum_{j=m+1}^{\infty} \mu_j \|\phi_j\|_2^2} \right\|_\infty && \text{(Cauchy-Schwarz)} \\
&\leq B \sqrt{\sum_{j=m+1}^{\infty} j^{2q} e^{-2\beta j}} && (\|\phi_j\|_\infty \leq B j^q, \mu_j \leq e^{-2\beta j}) \\
&\leq B \sqrt{\sum_{j=m+1}^{\infty} e^{-\beta j}} && \left( \text{if } \frac{m}{\log m} \geq \frac{q}{\beta} \right) \\
&= B \frac{e^{-\beta m/2}}{\sqrt{e^\beta - 1}} \leq 2B \frac{e^{-\beta m/2}}{\beta}. && (\sqrt{e^{2x} - 1} \geq e^x - 1 \geq x, x \geq 0)
\end{aligned}
$$

(35)

Hence, we can take $m_\varepsilon$ to be the smallest integer solution to $m \geq \frac{2}{\beta} \left| \log \left( \frac{8B}{\beta \varepsilon} \right) \right|$ to guarantee that for every $f \in \mathscr{P}$ there exists $f' \in \mathscr{P}_m$ at most $\varepsilon/4$ removed from $f$, i.e., $\|f - f'\|_\infty \leq \varepsilon/4$.

Next, we construct an $\varepsilon/4$-covering of the set $\mathscr{P}_m$. Observe now that the set of parameters of $\Theta_m$ defining $\mathscr{P}_m$ satisfies:

$$
\Theta_m \triangleq \left\{ \theta \in \mathbb{R}^m \ \middle| \ \sum_{j=1}^{m} \frac{\theta_j^2}{\mu_j} \leq 1 \right\}.
$$

Using this, we obtain a covering of $(\mathscr{P}_m, \|\cdot\|_\infty)$ by regarding it as a subset of $\mathbb{R}^m$. More precisely, $\Theta_m$ is the unit ball in the norm $\|\theta\|_\mu \triangleq \sqrt{\sum_{i=1}^m \theta_i^2 / \mu_i}, \theta \in \mathbb{R}^m$. Hence, by a standard volumetric argument, we need no more than $(1 + 2/\delta)^m$ points to cover $\Theta_m$ at resolution $\delta$ in $\|\cdot\|_\mu$. Let now $\delta > 0$ and choose $N \in \mathbb{N}_+$ so that $\{\theta^1, \ldots, \theta^N\}$ is an optimal $\delta$-covering of $\Theta_m$. We thus obtain the cover $\mathscr{P}_m^N \triangleq \{(\theta^1)^\top \phi(\cdot), \ldots, (\theta^N)^\top \phi(\cdot)\} \subset \mathscr{P}_m$ where $\phi(\cdot) = (\phi_1(\cdot), \ldots, \phi_m(\cdot))$. Let $f' = (\theta')^\top \phi \in \mathscr{P}_m$ be arbitrary. It remains to verify the resolution of $\mathscr{P}_m^N$:

$$
\begin{aligned}
\min_{n \in [N]} \|f' - (\theta^n)^\top \phi\|_\infty = \min_{n \in [N]} \left\| \sum_{j=1}^{m} (\theta'_j - \theta_j^n) \phi_j \right\|_\infty & \\
&\leq \min_{n \in [N]} \left\| \sqrt{\sum_{j=1}^{m} \frac{(\theta'_j - \theta_j^n)^2}{\mu_j}} \sqrt{\sum_{j=1}^{m} \mu_j \|\phi_j\|_2^2} \right\|_\infty && \text{(Cauchy-Schwarz)} \\
&\leq \delta B m^q && (\|\phi_j\|_\infty \leq B m^q \text{ if } j \leq m).
\end{aligned}
$$

Hence, if we take $N$ large enough so that $\delta \leq \frac{\varepsilon}{4Bm^q}$, $\mathscr{P}_m^N$ is a cover of $\mathscr{P}_m$ at resolution $\varepsilon/4$. Hence, since we may take $m \leq m_\varepsilon$:

$$
N \leq \left( 1 + \frac{8Bm_\varepsilon^q}{\varepsilon} \right)^{m_\varepsilon}.
$$

Now, we can immediately convert the covering $\mathscr{P}_m^N$ into an *exterior cover*[6] of the set $P$. For every $f \in P$, by the approximation property of $\mathscr{P}_m$, there exists an $f' \in \mathscr{P}_m$ such that $\|f - f'\|_\infty \leq \varepsilon/4$. But since $f' \in \mathscr{P}_m$, there exists an $f'' \in \mathscr{P}_m^N$ such that $\|f' - f''\|_\infty \leq \varepsilon/4$. By triangle inequality, $\|f - f''\|_\infty \leq \varepsilon/2$. Thus, $\mathscr{P}_m^N$ forms an exterior cover of $P$ at resolution $\varepsilon/2$. By Vershynin [39, Exercise 4.2.9], this means that there exists a (proper) cover of $P$ at resolution $\varepsilon$ with cardinality bounded by $N$.

---

[6] An exterior cover of a set $T$ is a cover where the elements are not restricted to $T$.

**Hypercontractivity:** We first show that every $f \in \mathscr{P}_m$ is hypercontractive. First, observe by orthogonality that the second moment takes the form:

$$\int \left\|\sum_{i=1}^{m_\varepsilon} \theta_i \phi_i\right\|_2^2 d\lambda = \sum_{i=1}^{m_\varepsilon} \theta_i^2 \|\phi_i\|_{L_2(\lambda)}^2 = \|\theta\|_2^2.$$

On the other hand by the eigenfunction growth condition:

$$\int \left\|\sum_{i=1}^{m_\varepsilon} \theta_i \phi_i\right\|_2^4 d\lambda \leq \int \left(\sum_{i=1}^{m_\varepsilon} |\theta_i| \|\phi_i\|_2\right)^4 d\lambda$$

$$\leq B^4 m_\varepsilon^{4q} \left(\sum_{i=1}^{m_e} |\theta_i|\right)^4$$

$$\leq B^4 m_\varepsilon^{4q+2} \|\theta\|_2^4$$

$$= B^4 m_\varepsilon^{4q+2} \left(\int \left\|\sum_{i=1}^{m_\varepsilon} \theta_i \phi_i\right\|_2^2 d\lambda\right)^2.$$

Now, for any $t \in \mathbb{N}$, by a change of measure, with $f = \sum_{i=1}^{m_\varepsilon} \theta_i \phi_i$,

$$\mathbf{E}_{\mu_t} \|f\|_2^4 = \int \|f\|_2^4 \frac{d\mu_t}{d\lambda} d\lambda \leq K \int \|f\|_2^4 d\lambda \leq K B^4 m_\varepsilon^{4q+2} \left(\int \|f\|_2^2 d\lambda\right)^2.$$

Hence, applying the previous inequality and another change of measure:

$$\frac{1}{T} \sum_{t=0}^{T-1} \mathbf{E}_{\mu_t} \|f\|_2^4 \leq K B^4 m_\varepsilon^{4q+2} \left(\int \|f\|_2^2 d\lambda\right)^2$$

$$= K B^4 m_\varepsilon^{4q+2} \left(\frac{1}{T} \int \sum_{t=0}^{T-1} \|f\|_2^2 \frac{d\lambda}{d\mu_t} d\mu_t\right)^2$$

$$\leq K^3 B^4 m_\varepsilon^{4q+2} \left(\frac{1}{T} \sum_{t=0}^{T-1} \mathbf{E}_{\mu_t} \|f\|_2^2\right)^2. \tag{36}$$

Next, fix a $f \in P$. We will show that $f$ is hypercontractive. First, recall that $f'$ is the element in $\mathscr{P}_m$ satisfying $\|f - f'\|_\infty \leq \varepsilon/4$. Hence, we have for every $x$:

$$\|f(x)\|_2^4 \leq 8(\|f(x) - f'(x)\|_2^4 + \|f'(x)\|_2^4) \leq \frac{\varepsilon^4}{32} + 8\|f'(x)\|_2^4, \tag{37}$$

$$\|f'(x)\|_2^2 \leq 2(\|f(x) - f'(x)\|_2^2 + \|f(x)\|_2^2) \leq \frac{\varepsilon^2}{2} + 2\|f(x)\|_2^2. \tag{38}$$

We now bound:

$$\frac{1}{T}\sum_{t=0}^{T-1} \mathbf{E}_{\mu_t}\|f\|_2^4 \overset{(a)}{\leq} \frac{\varepsilon^4}{32} + \frac{1}{T}\sum_{t=0}^{T-1}\mathbf{E}_{\mu_t}\|f'\|_2^4 \overset{(b)}{\leq} \frac{\varepsilon^4}{32} + K^3 B^4 m_\varepsilon^{4q+2}\left(\frac{1}{T}\sum_{t=0}^{T-1}\mathbf{E}_{\mu_t}\|f'\|_2^2\right)^2$$

$$\overset{(c)}{\leq} \frac{\varepsilon^4}{32} + K^3 B^4 m_\varepsilon^{4q+2}\left(\frac{\varepsilon^2}{2} + \frac{2}{T}\sum_{t=0}^{T-1}\mathbf{E}_{\mu_t}\|f\|_2^2\right)^2$$

$$\overset{(d)}{\leq} \left(\frac{1}{32} + \frac{25}{4}K^3 B^4 m_\varepsilon^{4q+2}\right)\left(\frac{1}{T}\sum_{t=0}^{T-1}\mathbf{E}_{\mu_t}\|f\|_2^2\right)^2.$$

Above, (a) uses the inequality (37), (b) uses the fact that $f' \in \mathscr{P}_m$ and (36), (c) uses (38), and (d) uses the assumption that $\varepsilon \leq \inf_{f \in P}\|f\|_{L^2(\mathsf{P}_X)}$, which implies that $\varepsilon^2 \leq \frac{1}{T}\sum_{t=0}^{T-1}\mathbf{E}_{\mu_t}\|f\|_2^2$ and $\varepsilon^4 \leq \left(\frac{1}{T}\sum_{t=0}^{T-1}\mathbf{E}_{\mu_t}\|f\|_2^2\right)^2$. Since $f \in P$ is arbitrary, the claim follows. $\blacksquare$

# F Basic tools for analyzing the dependency matrix

In this section, we outline some basic tools used to analyze the dependency matrix $\Gamma_{\mathsf{dep}}(\mathsf{P}_X)$. We will introduce the following shorthand. Given a process $\{Z_t\}_{t \geq 0}$ and indices $0 \leq i \leq j \leq k$, we will write $\mathsf{P}_{Z_{j:k}}(\cdot \mid Z_{0:i} = z_{0:i})$ as shorthand for $\mathsf{P}_{Z_{j:k}}(\cdot \mid A)$ for $A \in \mathcal{Z}_{0:i}$, where we recall that $\mathcal{Z}_{0:i}$ denotes the $\sigma$-algebra generated by $Z_{0:i}$. We will also write $\operatorname{ess\,sup}_{z_{0:i} \in \mathsf{Z}_{0:i}}$ as shorthand for $\sup_{A \in \mathcal{Z}_{0:i}}$.

Before we proceed, we recall the coupling representation of the total-variation norm:

$$\|\mu - \nu\|_{\mathsf{TV}} = \inf\{\mathbf{P}(X \neq Y) \mid (X, Y) \text{ is a coupling of } (\mu, \nu)\}. \tag{39}$$

**Proposition F.1.** *Suppose that $\{Z_t\}_{t \geq 0}$ is a Markov chain. For any integers $0 \leq i \leq j \leq k$:*

$$\|\mathsf{P}_{Z_{j:k}}(\cdot \mid Z_i = z) - \mathsf{P}_{Z_{j:k}}\|_{\mathsf{TV}} = \|\mathsf{P}_{Z_j}(\cdot \mid Z_i = z) - \mathsf{P}_{Z_j}\|_{\mathsf{TV}}.$$

*Proof.* Let us first prove the upper bound. Let $(Z_j, Z_j')$ be a coupling of $(\mathsf{P}_{Z_j}(\cdot \mid Z_i = z), \mathsf{P}_{Z_j})$. We can construct a coupling $(\bar{Z}_{j:k}, \bar{Z}_{j:k}')$ of $(\mathsf{P}_{Z_{j:k}}(\cdot \mid Z_i = z), \mathsf{P}_{Z_{j:k}})$ by first setting $\bar{Z}_j = Z_j$, $\bar{Z}_j' = Z_j'$, and then evolving the chains onward via the following process. If $\bar{Z}_j = \bar{Z}_j'$, we evolve $\bar{Z}_{j+1:k}$ onwards according to the dynamics, and copy $\bar{Z}_{j+1:k}' = \bar{Z}_{j+1:k}$. Otherwise if $\bar{Z}_j \neq \bar{Z}_j'$, then we evolve both chains separately. Observe that $\bar{Z}_{j:k} \neq \bar{Z}_{j:k}'$ iff $Z_j \neq Z_j'$. Hence $\|\mathsf{P}_{Z_{j:k}}(\cdot \mid Z_i = x) - \mathsf{P}_{Z_{j:k}}\|_{\mathsf{TV}} \leq \mathbf{P}(Z_j \neq Z_j')$. Since the coupling $(Z_j, Z_j')$ is arbitrary, taking the infimum over all couplings of $(\mathsf{P}_{Z_j}(\cdot \mid Z_i = z), \mathsf{P}_{Z_j})$ yields the upper bound via (39).

We now turn to the lower bound. Let $(Z_{j:k}, Z_{j:k}')$ be a coupling of $(\mathsf{P}_{Z_{j:k}}(\cdot \mid Z_i = z), \mathsf{P}_{Z_{j:k}})$. Since projection $(Z_j, Z_j')$ is a coupling for $(\mathsf{P}_{Z_j}(\cdot \mid Z_i = z), \mathsf{P}_{Z_j})$, and $Z_j \neq Z_j'$ implies $Z_{j:k} \neq Z_{j:k}'$, we have again by (39):

$$\|\mathsf{P}_{Z_j}(\cdot \mid Z_i = x) - \mathsf{P}_{Z_j}\|_{\mathsf{TV}} \leq \mathbf{P}(Z_j \neq Z_j') \leq \mathbf{P}(Z_{j:k} \neq Z_{j:k}').$$

Taking the infimum over all couplings of $(\mathsf{P}_{Z_{j:k}}(\cdot \mid Z_i = z), \mathsf{P}_{Z_{j:k}})$ yields the lower bound. ∎

**Proposition F.2.** *Let $M, N$ be two size conforming matrices with all non-negative entries. Suppose that $M \leq N$, where the inequality holds elementwise. Then, $\|M\|_{\mathsf{op}} \leq \|N\|_{\mathsf{op}}$.*

*Proof.* Let $Q$ be a matrix with non-negative entries, and let $q_i$ denote the rows of $Q$. The variational form of the operator norm states that $\|Q\|_{\mathsf{op}} = \sup_{\|v\|_2 \leq 1} \|Qv\|_2 = \sup_{\|v\|_2 \leq 1} \sqrt{\sum_i \langle q_i, v \rangle^2}$. Since each $q_i$ only has non-negative entries. The supremum must be attained by a vector $v$ with non-negative entries, otherwise flipping the sign of the negative entries in $v$ would only possibly increase the value of $\|Qv\|_2$, and never decrease the value.

Now let $m_i, n_i$ denote the rows of $M, N$, and let $v$ be a vector with non-negative entries. Since $0 \leq m_i \leq n_i$ (elementwise), it is clear that $\langle m_i, v \rangle^2 \leq \langle n_i, v \rangle^2$. Hence the claim follows. ∎

**Proposition F.3.** *Let $a_1, \ldots, a_n \in \mathbb{R}$, and let $M \in \mathbb{R}^{n \times n}$ be the upper triangular Toeplitz matrix:*

$$M = \begin{bmatrix} a_1 & a_2 & a_3 & a_4 & \cdots & a_n \\ 0 & a_1 & a_2 & a_3 & \cdots & a_{n-1} \\ 0 & 0 & a_1 & a_2 & \cdots & a_{n-2} \\ \vdots & \vdots & \vdots & \vdots & \ddots & \vdots \\ 0 & 0 & 0 & 0 & \vdots & a_1 \end{bmatrix}.$$

*We have that:*

$$\|M\|_{\mathsf{op}} \leq \sum_{i=1}^n |a_i|.$$

*Proof.* Let $E_i$, for $i = 1, \ldots, n$, denote the shift matrix where $E_i$ has ones along the $(i-1)$-th super diagonal and is zero everywhere else (the zero-th diagonal refers to the main diagonal). It is not hard to see that $\|E_i\|_{\mathsf{op}} \leq 1$ for all $i$, since it simply selects (and shifts) a subset of the coordinates of the input. With this notation, $M = \sum_{i=1}^n a_i E_i$. The claim now follows by the triangle inequality. ∎

**Proposition F.4.** *Let $\{Z_t\}_{t\geq 0}$ be a Markov process, and let $\mathsf{P}_Z$ denote the joint distribution of $\{Z_t\}_{t=0}^{T-1}$. We have that:*

$$\|\Gamma_{\mathsf{dep}}(\mathsf{P}_Z)\|_{\mathsf{op}} \leq 1 + \sqrt{2}\sum_{k=1}^{T-1} \max_{t=0,\ldots,T-1-k} \operatorname{ess\,sup}_{z\in\mathsf{Z}_t} \sqrt{\|\mathsf{P}_{Z_{t+k}}(\cdot \mid Z_t = z) - \mathsf{P}_{Z_{t+k}}\|_{\mathsf{TV}}}.$$

*Proof.* For any indices $0 \leq i < j$, by the Markov property and Proposition F.1:

$$\operatorname{ess\,sup}_{z_{0:i}\in\mathsf{Z}_{0:i}} \|\mathsf{P}_{Z_{j:T-1}}(\cdot \mid Z_{0:i} = z_{0:i}) - \mathsf{P}_{Z_{j:T-1}}\|_{\mathsf{TV}} = \operatorname{ess\,sup}_{z\in\mathsf{Z}_i} \|\mathsf{P}_{Z_{j:T-1}}(\cdot \mid Z_i = z) - \mathsf{P}_{Z_{j:T-1}}\|_{\mathsf{TV}}$$

$$= \operatorname{ess\,sup}_{z\in\mathsf{Z}_i} \|\mathsf{P}_{Z_j}(\cdot \mid Z_i = z) - \mathsf{P}_{Z_j}\|_{\mathsf{TV}}.$$

Therefore:

$$\Gamma_{\mathsf{dep}}(\mathsf{P}_Z)_{ij} = \sqrt{2}\operatorname{ess\,sup}_{z\in\mathsf{Z}_i} \sqrt{\|\mathsf{P}_{Z_j}(\cdot \mid Z_i = z) - \mathsf{P}_{Z_j}\|_{\mathsf{TV}}}$$

$$\leq \sqrt{2}\max_{t=0,\ldots,T-1-(j-i)} \operatorname{ess\,sup}_{z\in\mathsf{Z}_t} \sqrt{\|\mathsf{P}_{Z_{t+j-i}}(\cdot \mid Z_t = z) - \mathsf{P}_{Z_{t+j-i}}\|_{\mathsf{TV}}} \triangleq a_{j-i}.$$

Thus, we can construct a matrix $\Gamma'$ such that for all indices $0 \leq i < j$, we have $\Gamma'_{ij} = a_{j-i}$, and the other entries are identical to $\Gamma_{\mathsf{dep}}(\mathsf{P}_Z)$. This gives us the entry-wise bound $\Gamma_{\mathsf{dep}}(\mathsf{P}_Z) \leq \Gamma'$. Applying Proposition F.2 and Proposition F.3, we conclude $\|\Gamma_{\mathsf{dep}}(\mathsf{P}_Z)\|_{\mathsf{op}} \leq \|\Gamma'\|_{\mathsf{op}} \leq 1 + \sum_{k=1}^{T-1} a_k$. ∎

## G  Mixing properties of truncated Gaussian processes

We first recall the notation from Appendix B.1. Let $\{W_t\}_{t\geq 0}, \{W'_t\}_{t\geq 0}$ be sequences of iid $N(0,I)$ vectors in $\mathbb{R}^{d_\mathsf{x}}$. Fix a dynamics function $f : \mathbb{R}^{d_\mathsf{x}} \to \mathbb{R}^{d_\mathsf{x}}$ and radius $R > 0$. Define the truncated Gaussian noise process $\{\bar{W}_t\}_{t\geq 0}$ as $\bar{W}_t \triangleq W'_t\mathbf{1}\{\|W'_t\|_2 \leq R\}$. Now, consider the two processes:

$$X_{t+1} = f(X_t) + HW_t, \quad X_0 = HW_0, \tag{40a}$$

$$\bar{X}_{t+1} = f(\bar{X}_t) + H\bar{W}_t, \quad \bar{X}_0 = H\bar{W}_0. \tag{40b}$$

We develop the necessary arguments in this section to transfer mixing properties of the original process (40a) to the truncated process (40b). This will let us apply our results in Section 4 to unbounded processes of the form (40a), by studying their truncated counterparts (40b).

The main tool to do this is the following coupling argument.

**Proposition G.1.** *Fix a $\delta \in (0,1)$. Let $k \in \{1,\ldots,T-1\}$ and $t \in \{0,\ldots,T-1-k\}$. Consider the processes $\{X_t\}_{t\geq 0}$ and $\{\bar{X}_t\}_{t\geq 0}$ described in (40a) and (40b) with $R$ satisfying the inequality $R \geq \sqrt{d_\mathsf{X}} + \sqrt{2\log(T/\delta)}$. The following bound hold for any $x \in \mathbb{R}^{d_\mathsf{x}}$:*

$$\|\mathsf{P}_{X_{t+k}}(\cdot \mid X_t = x) - \mathsf{P}_{\bar{X}_{t+k}}(\cdot \mid \bar{X}_t = x)\|_{\mathsf{TV}} \leq \delta.$$

*The following bound also holds for any $t \in \{0,\ldots,T-1\}$:*

$$\|\mathsf{P}_{X_t} - \mathsf{P}_{\bar{X}_t}\|_{\mathsf{TV}} \leq \delta.$$

*Proof.* Let $(Z_{t+k}, Z'_{t+k})$ be a coupling of $(\mathsf{P}_{X_{t+k}}(\cdot \mid X_t = x), \mathsf{P}_{\bar{X}_{t+k}}(\cdot \mid \bar{X}_t = x))$ defined as follows. We initialize both $X_t = \bar{X}_t = x$. We let $\{W_s\}_{s=t}^{t+k-1}$ be iid draws from $N(0,I)$, we set $\bar{W}_s = W_s\mathbf{1}\{\|W_s\|_2 \leq R\}$, and we evolve $X_t, \bar{X}_t$ forward to $X_{t+k}, \bar{X}_{t+k}$ according to their laws (40a) and (40b), respectively. Let $\mathcal{E}$ denote the event $\mathcal{E} = \{\max_{s=t,\ldots,t+k-1}\|W_s\|_2 \leq R\}$. A standard Gaussian concentration plus union bound yields $\mathbf{P}(\mathcal{E}^c) \leq \delta$, since $t + k - 1 \leq T - 2$. By the coupling representation (39) of the total-variation norm:

$$\|\mathsf{P}_{X_{t+k}}(\cdot \mid X_t = x) - \mathsf{P}_{\bar{X}_{t+k}}(\cdot \mid \bar{X}_t = x)\|_{\mathsf{TV}} \leq \mathbf{P}\{Z_{t+k} \neq Z'_{t+k}\}$$

$$= \mathbf{P}(\{Z_{t+k} \neq Z'_{t+k}\} \cap \mathcal{E}) + \mathbf{P}(\{Z_{t+k} \neq Z'_{t+k}\} \cap \mathcal{E}^c)$$

$$\leq \mathbf{P}(\mathcal{E}^c) \leq \delta.$$

The second inequality holds since on $\mathcal{E}$, $Z_{t+k} = Z'_{t+k}$ because the truncation is inactive the entire duration of the process. This establishes the first inequality.

The second inequality holds by a nearly identical coupling argument, where we set $\{W_s\}_{s=0}^{t-1}$ to be iid draws from $N(0, I)$, we set $\bar{W}_s = W_s \mathbf{1}\{\|W_s\|_2 \leq R\}$, and we initialize the processes at $X_0 = HW_0$ and $\bar{X}_0 = H\bar{W}_0$. ∎

The next result states that as long as we set the failure probability $\delta$ in $R$ as $1/T^2$, then we can bound the dependency matrix appropriately.

**Proposition G.2.** *Let* $\mathsf{P}_X$ *denote the joint distribution of* $\{X_t\}_{t=0}^{T-1}$ *from* (40a), *and let* $\mathsf{P}_{\bar{X}}$ *denote the joint distribution of* $\{\bar{X}_t\}_{t=0}^{T-1}$ *from* (40b), *with* $R \geq \sqrt{d_X} + \sqrt{6 \log T}$. *We have that:*

$$\|\Gamma_{\mathsf{dep}}(\mathsf{P}_{\bar{X}})\|_{\mathsf{op}} \leq 3 + \sqrt{2} \sum_{k=1}^{T-1} \max_{t=0,\ldots,T-1-k} \operatorname*{ess\,sup}_{x \in \bar{X}_t} \sqrt{\|\mathsf{P}_{X_{t+k}}(\cdot \mid X_t = x) - \mathsf{P}_{X_{t+k}}\|_{\mathsf{TV}}}. \quad (41)$$

*Proof.* First, we invoke Proposition F.4 to obtain:

$$\|\Gamma_{\mathsf{dep}}(\mathsf{P}_{\bar{X}})\|_{\mathsf{op}} \leq 1 + \sqrt{2} \sum_{k=1}^{T-1} \max_{t=0,\ldots,T-1-k} \operatorname*{ess\,sup}_{x \in \bar{X}_t} \sqrt{\|\mathsf{P}_{\bar{X}_{t+k}}(\cdot \mid \bar{X}_t = x) - \mathsf{P}_{\bar{X}_{t+k}}\|_{\mathsf{TV}}}.$$

Now fix $k \in \{1, \ldots, T-1\}$, $t \in \{0, \ldots, T-1-k\}$, and $x \in \bar{X}_t$. By triangle inequality:

$$\|\mathsf{P}_{\bar{X}_{t+k}}(\cdot \mid \bar{X}_t = x) - \mathsf{P}_{\bar{X}_{t+k}}\|_{\mathsf{TV}} \leq \|\mathsf{P}_{X_{t+k}}(\cdot \mid X_t = x) - \mathsf{P}_{X_{t+k}}\|_{\mathsf{TV}}$$
$$+ \|\mathsf{P}_{\bar{X}_{t+k}}(\cdot \mid \bar{X}_t = x) - \mathsf{P}_{X_{t+k}}(\cdot \mid X_t = x)\|_{\mathsf{TV}}$$
$$+ \|\mathsf{P}_{\bar{X}_{t+k}} - \mathsf{P}_{X_{t+k}}\|_{\mathsf{TV}}.$$

By setting $\delta = 1/T^2$ in Proposition G.1, the last two terms are bounded by $1/T^2$. Hence:

$$\|\mathsf{P}_{\bar{X}_{t+k}}(\cdot \mid \bar{X}_t = x) - \mathsf{P}_{\bar{X}_{t+k}}\|_{\mathsf{TV}} \leq \|\mathsf{P}_{X_{t+k}}(\cdot \mid X_t = x) - \mathsf{P}_{X_{t+k}}\|_{\mathsf{TV}} + \frac{2}{T^2}.$$

The claim now follows. ∎

Crucially, the essential supremum in (41) is over $\bar{X}_t$ and *not* $X_t$, of which the latter is unbounded.

The next condition that we need to check for the truncated process (40b) is that the noise process $\{H\bar{W}_t\}_{t \geq 0}$ is still a zero-mean sub-Gaussian martingale difference sequence. By symmetry of the truncation, it is clear that the noise process remains zero-mean. To check sub-Gaussianity, we use the following result.

**Proposition G.3.** *Let* $A \subseteq \mathbb{R}^{d_X}$ *be any set that is symmetric about the origin. Let* $W \sim N(0, I)$, *and let* $\bar{W} := W \mathbf{1}\{W \in A\}$. *We have that* $\bar{W}$ *is* 4-*sub-Gaussian. Hence for any* $H$, $H\bar{W}$ *is* $4\|H\|_{\mathsf{op}}^2$-*sub-Gaussian.*

*Proof.* Since $A$ is symmetric about the origin, $\bar{W}$ inherits the symmetry of $W$, i.e., $\mathbf{E}[\bar{W}] = 0$. Now fix a unit vector $u \in \mathbb{R}^{d_X}$, and $\lambda \in \mathbb{R}$. First, let us assume that $\lambda^2 \leq 1/2$. Let $\varepsilon$ denote a Rademacher random variable[7] that is independent of $\bar{W}$. Since $\bar{W}$ is a symmetric zero-mean distribution, we have that $\langle u, \bar{W} \rangle$ has the same distribution as $\varepsilon \langle u, \bar{W} \rangle$. Therefore:

$$\begin{aligned}
\mathbf{E} \exp(\lambda \langle u, \bar{W} \rangle) &= \mathbf{E}_{\bar{W}} \mathbf{E}_\varepsilon \exp(\lambda \varepsilon \langle u, \bar{W} \rangle) \\
&\leq \mathbf{E}_{\bar{W}} \exp(\lambda^2 \langle u, \bar{W} \rangle^2 / 2) && \cosh(x) \leq \exp(x^2/2) \, \forall x \in \mathbb{R} \\
&\leq \mathbf{E}_{\bar{W}} \exp(\lambda^2 \langle u, W \rangle^2 / 2) \\
&= \frac{1}{(1 - \lambda^2)^{1/2}} && \text{since } \langle u, W \rangle \sim N(0, 1) \text{ and } \lambda^2 < 1 \\
&\leq \exp(\lambda^2) && \frac{1}{1 - x} \leq \exp(2x) \, \forall x \in [0, 1/2].
\end{aligned}$$

---

[7] That is, $\mathbf{P}(\varepsilon = 1) = \mathbf{P}(\varepsilon = -1) = 1/2$.

Now, let us assume $\lambda^2 > 1/2$. We have:

$$
\begin{aligned}
\mathbf{E}\exp(\lambda\langle u, \bar{W}\rangle) &= \mathbf{E}\exp(\lambda\langle u, W\rangle)\mathbf{1}\{W \in A\} + \mathbf{P}(W \notin A) \\
&\leq \mathbf{E}\exp(\lambda\langle u, W\rangle) + 1 \\
&= \exp(\lambda^2/2) + 1 && \text{since } \langle u, W\rangle \sim N(0,1) \\
&\leq \exp(\log 2 + \lambda^2/2) && \text{since } 1 \leq \exp(\lambda^2/2) \\
&\leq \exp((2\log 2 + 1/2)\lambda^2) && \text{since } \lambda^2 > 1/2 \\
&\leq \exp(2\lambda^2).
\end{aligned}
$$

The claim now follows. ∎

The following result will be useful later on. It states that the truncation does not affect the isotropic nature of the noise, as long as the truncation probability is a sufficiently small constant.

**Proposition G.4.** *Let $A \subseteq \mathbb{R}^{d_\times}$ be any set. Let $W \sim N(0,I)$ and $\bar{W} = W\mathbf{1}\{W \in A\}$, and suppose that $\mathbf{P}(W \notin A) \leq 1/12$. We have that:*

$$
\frac{1}{2}I \preccurlyeq \mathbf{E}[\bar{W}\bar{W}^\mathsf{T}] \preccurlyeq I.
$$

*Proof.* The upper bound is immediate. For the lower bound, fix a $v \in \mathbb{S}^{d_\times - 1}$. We have:

$$
\begin{aligned}
\mathbf{E}[\langle v, \bar{W}\rangle^2] &= \mathbf{E}[\langle v, \bar{W}\rangle^2\mathbf{1}\{W \in A\}] + \mathbf{E}[\langle v, \bar{W}\rangle^2\mathbf{1}\{W \notin A\}] \\
&= \mathbf{E}[\langle v, \bar{W}\rangle^2\mathbf{1}\{W \in A\}] && \text{since } \bar{W} = W\mathbf{1}\{W \in A\} \\
&= \mathbf{E}[\langle v, W\rangle^2] - \mathbf{E}[\langle v, W\rangle^2\mathbf{1}\{W \notin A\}] \\
&\geq 1 - \sqrt{\mathbf{E}[\langle v, W\rangle^4]\mathbf{P}(W \notin A)} && \text{since } \langle v, W\rangle \sim N(0,1) \text{ and Cauchy-Schwarz} \\
&\geq 1 - \sqrt{3\delta} \\
&\geq 1/2 && \text{since } \mathbf{P}(W \notin A) \leq 1/12.
\end{aligned}
$$

Since $v \in \mathbb{S}^{d_\times - 1}$ is arbitrary, the claim follows. ∎

**Proposition G.5.** *Let $w \sim N(0,I)$ and let $M$ be positive semidefinite. We have:*

$$
\mathbf{E}[(w^\mathsf{T}Mw)^2] \leq 3(\mathbf{E}[w^\mathsf{T}Mw])^2.
$$

*Proof.* This is a standard calculation [see e.g. 41, Lemma 6.2]. ∎

We will also need the following result which states that the square of quadratic forms under $\bar{W}$ can be upper bounded by the square of the same quadratic form under the original noise $W$.

**Proposition G.6.** *Let $A \subseteq \mathbb{R}^{d_\times}$ be any set. Let $W \sim N(0,I)$ and $\bar{W} = W\mathbf{1}\{W \in A\}$. Fix a $k \geq 1$. Let $M \in \mathbb{R}^{d_\times k \times d_\times k}$ be a positive semidefinite matrix, and let $\{W_i\}_{i=1}^k$ and $\{\bar{W}_i\}_{i=1}^k$ be iid copies of $W$ and $\bar{W}$, respectively. Let $W_{1:k} \in \mathbb{R}^{d_\times k}$ denote the stacked column vector of $\{W_i\}_{i=1}^k$ and similarly for $\bar{W}_{1:k} \in \mathbb{R}^{d_\times k}$. We have that:*

$$
\mathbf{E}[(\bar{W}_{1:k}^\mathsf{T}M\bar{W}_{1:k})^2] \leq \mathbf{E}[(W_{1:k}^\mathsf{T}MW_{1:k})^2].
$$

*Proof.* Let $\{M_{ij}\}_{i,j=1}^k \subset \mathbb{R}^{d_\times \times d_\times}$ denote the blocks of $M$. We have:

$$
\mathbf{E}[(\bar{W}_{1:k}^\mathsf{T}M\bar{W}_{1:k})^2] = \sum_{a,b,c,d} \mathbf{E}[(\bar{W}_a^\mathsf{T}M_{ab}\bar{W}_b)(\bar{W}_c^\mathsf{T}M_{cd}\bar{W}_d)].
$$

Since $\bar{W}$ is zero-mean, the only terms that are non-zero in the summation have the following form $\mathbf{E}[(\bar{W}_a^\mathsf{T}M_{aa}\bar{W}_a)(\bar{W}_b^\mathsf{T}M_{bb}\bar{W}_b)]$. Hence:

$$
\begin{aligned}
\mathbf{E}[(\bar{W}_{1:k}^\mathsf{T}M\bar{W}_{1:k})^2] &= \sum_a \mathbf{E}[(\bar{W}_a^\mathsf{T}M_{aa}\bar{W}_a)^2] + \sum_{a\neq b} \mathbf{E}[(\bar{W}_a^\mathsf{T}M_{aa}\bar{W}_a)(\bar{W}_b^\mathsf{T}M_{bb}\bar{W}_b)] \\
&\overset{(a)}{\leq} \sum_a \mathbf{E}[(W_a^\mathsf{T}M_{aa}W_a)^2] + \sum_{a\neq b} \mathbf{E}[(W_a^\mathsf{T}M_{aa}W_a)(W_b^\mathsf{T}M_{bb}W_b)] \\
&= \mathbf{E}[(W_{1:k}^\mathsf{T}MW_{1:k})^2].
\end{aligned}
$$

Above, (a) holds since the matrix $M$ is positive semidefinite and therefore so are its diagonal sub-blocks $M_{aa}$. This ensures that each of the quadratic forms are non-negative, and hence we can upper bound the first expression by removing the indicators. ∎

We conclude this section with a result that will be useful for analyzing the mixing properties of the Gaussian process (40a), when the dynamics function $f$ is nonlinear. First, recall the definition of the 1-Wasserstein distance:

$$W_1(\mu, \nu) \triangleq \inf\{\mathbf{E}\|X - Y\|_2 \mid (X, Y) \text{ is a coupling of } (\mu, \nu)\}. \tag{42}$$

The following result uses the smoothness of the Gaussian transition kernel to upper bound the TV norm via the 1-Wasserstein distance. This result is inspired by the work of Chae and Walker [42].

**Lemma G.1.** *Let $X_0, Y_0$ be random vectors in $\mathbb{R}^p$, and let $f : \mathbb{R}^p \to \mathbb{R}^n$ be an L-Lipschitz function. Suppose that $X_0, Y_0$ are both absolutely continuous w.r.t. the Lebesgue measure on $\mathbb{R}^p$. Let $\Sigma \in \mathbb{R}^{n \times n}$ be positive definite, and let $X_1, Y_1$ be random vectors in $\mathbb{R}^n$ defined conditionally: $X_1 \mid X_0 = N(f(X_0), \Sigma)$ and $Y_1 \mid Y_0 = N(f(Y_0), \Sigma)$. Then:*

$$\|\mathsf{P}_{X_1} - \mathsf{P}_{Y_1}\|_{\mathsf{TV}} \leq \frac{L\sqrt{\mathrm{tr}(\Sigma^{-1})}}{2} W_1(\mathsf{P}_{X_0}, \mathsf{P}_{Y_0}).$$

*Proof.* Since $X_0, Y_0$ are absolutely continuous, the Radon-Nikodym theorem ensures that there exists densities $p_0, q_0$ for $X_0, Y_0$, respectively. Let $\phi$ denote the density of the $N(0, \Sigma)$ distribution. Let $p_1, q_1$ denote the densities of $X_1, Y_1$, respectively. We have the following convolution expressions:

$$p_1(x) = \int \phi(x - f(x_0))p_0(x_0)dx_0 = \int \phi(x - f(X_0))dX_0,$$

$$q_1(x) = \int \phi(x - f(x_0))q_0(x_0)dx_0 = \int \phi(x - f(Y_0))dY_0.$$

Now, let $\pi$ be a coupling of $(X_0, Y_0)$. We can equivalently write $p_1, q_1$ as a convolution over $\pi$:

$$p_1(x) = \int \phi(x - f(X_0))d\pi(X_0, Y_0),$$

$$q_1(x) = \int \phi(x - f(Y_0))d\pi(X_0, Y_0).$$

Hence:

$$(p_1 - q_1)(x) = \int [\phi(x - f(X_0)) - \phi(x - f(Y_0))]d\pi(X_0, Y_0)$$
$$= \mathbf{E}_{\pi(X_0, Y_0)}[\phi(x - f(X_0)) - \phi(x - f(Y_0))].$$

Now by the $L^1$ representation of total-variation norm [see e.g. 1, Lemma 2.1]:

$$\|\mathsf{P}_{X_1} - \mathsf{P}_{Y_1}\|_{\mathsf{TV}} = \frac{1}{2} \int |p_1(x) - q_1(x)|dx$$
$$= \frac{1}{2} \int |\mathbf{E}_{\pi(X_0, Y_0)}[\phi(x - f(X_0)) - \phi(x - f(Y_0))]|dx$$
$$\overset{(a)}{\leq} \frac{1}{2} \int \mathbf{E}_{\pi(X_0, Y_0)}[|\phi(x - f(X_0)) - \phi(x - f(Y_0))|]dx$$
$$\overset{(b)}{=} \frac{1}{2}\mathbf{E}_{\pi(X_0, Y_0)}\left[\int |\phi(x - f(X_0)) - \phi(x - f(Y_0))|dx\right].$$

The inequality (a) is Jensen's inequality, and the equality (b) is Tonelli's theorem since the integrand is non-negative. We now focus on the inner integral inside the expectation over $\pi$. By the mean-value theorem, since $\phi$ is continuously differentiable, fixing $x, X_0, Y_0$:

$$|\phi(x - f(X_0)) - \phi(x - f(Y_0))| = \left|\int_0^1 \nabla\phi((1 - s)(x - f(Y_0)) + s(x - f(X_0)))^\mathsf{T}(f(Y_0) - f(X_0))ds\right|$$
$$\leq \|f(Y_0) - f(X_0)\|_2 \int_0^1 \|\nabla\phi(x - (sf(X_0) - (1 - s)f(Y_0)))\|_2 ds.$$

Hence, by another application of Tonelli's theorem:

$$
\int |\phi(x - f(X_0)) - \phi(x - f(Y_0))|dx \le \|f(X_0) - f(Y_0)\|_2 \int \int_0^1 \|\nabla\phi(x - (sf(X_0) - (1-s)f(Y_0)))\|_2 \, ds \, dx
$$

$$
= \|f(X_0) - f(Y_0)\|_2 \int_0^1 \int \|\nabla\phi(x - (sf(X_0) - (1-s)f(Y_0)))\|_2 \, dx \, ds
$$

$$
\le \|f(X_0) - f(Y_0)\|_2 \sqrt{\mathrm{tr}(\Sigma^{-1})}.
$$

The last inequality follows from the following computation. Observe that $\nabla\phi(x) = -\Sigma^{-1} x \phi(x)$ and define $\mu = sf(X_0) - (1-s)f(Y_0)$. Since $\mu$ does not depend on $x$, by the translation invariance of the Lebesgue integral:

$$
\int \|\nabla\phi(x - (sf(X_0) - (1-s)f(Y_0)))\|_2 dx = \int \|\Sigma^{-1}(x - \mu)\|_2 \phi(x - \mu)dx
$$

$$
= \int \|\Sigma^{-1} x\|_2 \phi(x)dx
$$

$$
= \mathbf{E}_{x \sim N(0, \Sigma^{-1})}[\|x\|_2]
$$

$$
\le \sqrt{\mathrm{tr}(\Sigma^{-1})}.
$$

The last inequality above is another application of Jensen's inequality. Therefore, combining the inequalities thus far, and using the $L$-Lipschitz property of $f$:

$$
\|\mathsf{P}_{X_1} - \mathsf{P}_{Y_1}\|_{\mathsf{TV}} \le \frac{\sqrt{\mathrm{tr}(\Sigma^{-1})}}{2} \mathbf{E}_{\pi(X_0, Y_0)}\left[\|f(X_0) - f(Y_0)\|_2\right]
$$

$$
\le \frac{L\sqrt{\mathrm{tr}(\Sigma^{-1})}}{2} \mathbf{E}_{\pi(X_0, Y_0)}[\|X_0 - Y_0\|_2].
$$

Since the coupling $\pi$ of $(X_0, Y_0)$ was arbitrary, the result now follows by taking the infimum of the right hand side over all valid couplings. ∎

## H   Recovering Ziemann et al. [15] via boundedness

Here we show how to recover the results for mixing systems from Ziemann et al. [15, Theorem 1 combined with Proposition 2], corresponding to $\alpha = 1$. This rests on the observation that $(B^2, 1)$-hypercontractivity is automatic by $B$-boundedness.

**Corollary H.1.** *Suppose that $\mathscr{F}_\star$ is star-shaped and $B$-bounded. Fix also $p \in \mathbb{R}_+$ and $q \in (0, 2)$ and suppose further that $\mathscr{F}_\star$ satisfies condition ([8]). Then we have that:*

$$
\mathbf{E}\|\widehat{f} - f_\star\|_{L^2}^2 \le 8\mathsf{EM}_T(\mathscr{F}_\star) + \frac{1}{\sqrt{8}}\left(16B^2 p\|\Gamma_{\mathsf{dep}}(\mathsf{P}_X)\|_{\mathsf{op}}^2\right)^{2/q} T^{-2/(2+q)}
$$

$$
+ \exp\left(\frac{-T^{q/(2+q)}}{16B^2\|\Gamma_{\mathsf{dep}}(\mathsf{P}_X)\|_{\mathsf{op}}^2}\right). \quad (43)
$$

The first two terms in inequality (43) are both of order $T^{-2/(2+q)}$ if $\|\Gamma_{\mathsf{dep}}\|_{\mathsf{op}}^2 = O(1)$. Note that, without further control of the moments of $f \in \mathscr{F}_\star$, the bound in Theorem 4.1 thus degrades by a factor of the dependency matrix through the second term.

**Proof of Corollary H.1**   Fix $c > 0$ to be determined later and choose $r = cT^{-1/(2+q)}$. We find:

$$
B^2 \mathcal{N}_\infty(\mathscr{F}_\star, r/\sqrt{8}) \exp\left(\frac{-Tr^2}{8B^2\|\Gamma_{\mathsf{dep}}(\mathsf{P}_X)\|_{\mathsf{op}}^2}\right)
$$

$$
\le \exp\left(p\left(\frac{\sqrt{8}}{r}\right)^q - \frac{Tr^2}{8B^2\|\Gamma_{\mathsf{dep}}(\mathsf{P}_X)\|_{\mathsf{op}}^2}\right) \qquad \text{(Condition (8))}
$$

$$
= \exp\left(\left[p\left(\frac{\sqrt{8}}{c}\right)^q - \frac{1}{8B^2\|\Gamma_{\mathsf{dep}}(\mathsf{P}_X)\|_{\mathsf{op}}^2}\right]T^{q/(2+q)}\right). \quad (r = cT^{-1/(2+q)})
$$

Hence we may solve for $c = \frac{1}{\sqrt{8}}\left(16B^2 p\|\Gamma_{\mathsf{dep}}(\mathsf{P}_X)\|_{\mathsf{op}}^2\right)^{1/q}$ in

$$p\left(\frac{\sqrt{8}}{c}\right)^q = \frac{1}{16B^2\|\Gamma_{\mathsf{dep}}(\mathsf{P}_X)\|_{\mathsf{op}}^2}$$

to arrive at the desired conclusion. ∎

# I   Linear dynamical systems

We define the truncated linear dynamics:

$$\bar{X}_{t+1} = A_\star \bar{X}_t + H\bar{V}_t, \quad \bar{X}_0 = H\bar{V}_0, \quad \bar{V}_t = V_t\mathbf{1}\{\|V_t\|_2 \le R\}. \tag{44}$$

We set $R = \sqrt{d_\mathsf{X}} + \sqrt{2(1+\beta)\log T}$ where $\beta > 4$ is a free parameter. Define the event $\mathcal{E}$ as:

$$\mathcal{E} := \left\{\max_{0 \le t \le T-1}\|V_t\|_2 \le R\right\}. \tag{45}$$

Note that by the setting of $R$, we have $\mathbf{P}(\mathcal{E}^c) \le 1/T^\beta$ using standard Gaussian concentration results plus a union bound. Furthermore on $\mathcal{E}$, the original GLM process driven by Gaussian noise (12) coincides with the truncated process (44). Let $\widehat{f}$ denote the LSE on the original process (44), and let $\bar{f}$ denote the LSE on the truncated process (44). Hence:

$$\mathbf{E}\|\widehat{f} - f_\star\|_{L^2}^2 = \mathbf{E}\|\widehat{f} - f_\star\|_{L^2}^2\mathbf{1}\{\mathcal{E}\} + \mathbf{E}\|\widehat{f} - f_\star\|_{L^2}^2\mathbf{1}\{\mathcal{E}^c\}$$
$$\le \mathbf{E}\|\bar{f} - f_\star\|_{L^2}^2 + \mathbf{E}\|\widehat{f} - f_\star\|_{L^2}^2\mathbf{1}\{\mathcal{E}^c\}.$$

Let us now control the error term $\mathbf{E}\|\widehat{f} - f_\star\|_{L^2}^2\mathbf{1}\{\mathcal{E}^c\}$. Since $X_t$ is a linear function of the Gaussian noise $\{W_t\}$ process, by Proposition G.5 we have $\mathbf{E}\|X_t\|_2^4 \le 3(\mathbf{E}\|X_t\|_2^2)^2$. Write $\widehat{f}(x) = \widehat{A}x$, and put $\widehat{\Delta} = \widehat{A} - A_\star$. We have:

$$\mathbf{E}\|\widehat{f} - f_\star\|_{L^2}^2\mathbf{1}\{\mathcal{E}^c\} = \frac{1}{T}\sum_{t=0}^{T-1}\mathbf{E}\|\widehat{\Delta}X_t\|_2^2\mathbf{1}\{\mathcal{E}^c\} \overset{(a)}{\le} \frac{4B^2}{T}\sum_{t=0}^{T-1}\mathbf{E}\|X_t\|_2^2\mathbf{1}\{\mathcal{E}^c\}$$

$$\overset{(b)}{\le} \frac{4B^2}{T^{1+\beta/2}}\sum_{t=0}^{T-1}\sqrt{\mathbf{E}\|X_t\|_2^4} \overset{(c)}{\le} \frac{4\sqrt{3}B^2}{T^{1+\beta/2}}\sum_{t=0}^{T-1}\mathbf{E}\|X_t\|_2^2$$

$$= \frac{4\sqrt{3}B^2}{T^{1+\beta/2}}\sum_{t=0}^{T-1}\mathrm{tr}(\Gamma_t) \overset{(d)}{\le} \frac{4\sqrt{3}B^2\,\mathrm{tr}(\Gamma_{T-1})}{T^{\beta/2}} \overset{(e)}{\le} \frac{4\sqrt{3}B^2\|H\|_{\mathsf{op}}^2\tau^2 d_\mathsf{X}}{(1-\rho)T^{\beta/2}}. \tag{46}$$

Above, (a) follows from the definition of $\mathscr{F}$, (b) follows from Cauchy-Schwarz, (c) uses the hypercontractivity bound $\mathbf{E}\|X_t\|_2^4 \le 3(\mathbf{E}\|X_t\|_2^2)^2$, (d) uses the fact that $\Gamma_t$ is monotonically increasing in the Loewner order, and (e) uses the following bound on $\mathrm{tr}(\Gamma_{T-1})$ using the $(\tau, \rho)$-stability of $A_\star$:

$$\mathrm{tr}(\Gamma_{T-1}) \le \frac{\|H\|_{\mathsf{op}}^2\tau^2 d_\mathsf{X}}{1-\rho^2} \le \frac{\|H\|_{\mathsf{op}}^2\tau^2 d_\mathsf{X}}{1-\rho}.$$

The remainder of the proof is to bound the error of the LSE $\bar{f}$ using Theorem 4.1. This involves two main steps: showing the trajectory hypercontractivity condition Definition 4.1 holds, and bounding the dependency matrix $\|\Gamma_{\mathsf{dep}}(\mathsf{P}_{\bar{X}})\|_{\mathsf{op}}$ (cf. Definition 4.2), where $\mathsf{P}_{\bar{X}}$ denotes the joint distribution of the process $\{\bar{X}_t\}_{t=0}^{T-1}$. Before we proceed, we define some reoccurring constants:

$$\mu \triangleq \lambda_{\min}(\Gamma_{\kappa-1}), \quad B_{\bar{X}} \triangleq \frac{\|H\|_{\mathsf{op}}\tau(\sqrt{d_\mathsf{X}} + \sqrt{2(1+\beta)\log T})}{1-\rho}. \tag{47}$$

## I.1   Trajectory hypercontractivity for truncated LDS

**Proposition I.1.** *Suppose that $T \ge \max\{6, 2\kappa\}$. The pair $(\mathscr{F}_\star, \mathsf{P}_{\bar{X}})$ with $\mathscr{F}$ given in (13) and $\mathsf{P}_{\bar{X}}$ as the joint distribution of $\{\bar{X}_t\}_{t=0}^{T-1}$ from (44) satisfies the $(C_{\mathsf{LDS}}, 2)$-trajectory hypercontractivity condition with $C_{\mathsf{LDS}} = \frac{108\tau^4\|H\|_{\mathsf{op}}^4}{(1-\rho)^2\mu^2}$.*

*Proof.* Fix any size-conforming matrix $M$. Let the noise process $\{\bar{V}_t\}_{t=0}^{T-1}$ be stacked into a noise vector $\bar{V}_{0:T-1} \in \mathbb{R}^{d_X T}$. Observe that we can write $MX_t = MT_t\bar{V}_{0:T-1}$ for some matrix $T_t$. We invoke the comparison inequality in Proposition G.6 followed by the Gaussian fourth moment identity in Proposition G.5 to conclude that:

$$\mathbf{E}\|M\bar{X}_t\|_2^4 = \mathbf{E}\|MT_t\bar{V}_{0:T-1}\|_2^4 \leq \mathbf{E}\|MX_t\|_2^4 \leq 3(\mathbf{E}\|MX_t\|_2^2)^2 = 3\operatorname{tr}(M^\mathsf{T}M\Gamma_t)^2.$$

By monotonicity of $\Gamma_t$ and the assumption $T \geq 6$:

$$\frac{1}{T}\sum_{t=0}^{T-1}\Gamma_t \succcurlyeq \frac{1}{T}\sum_{t=\lfloor T/2\rfloor}^{T-1}\Gamma_t \succcurlyeq \frac{T-\lfloor T/2\rfloor}{T}\Gamma_{\lfloor T/2\rfloor} \succcurlyeq \frac{1}{3}\Gamma_{\lfloor T/2\rfloor}. \tag{48}$$

Since $T \geq 2\kappa$, the inequality $\Gamma_{\lfloor T/2\rfloor} \succcurlyeq \Gamma_{\kappa-1}$ holds, and therefore $\Gamma_{\lfloor T/2\rfloor}$ is invertible since $(A_\star, H)$ is $\kappa$-step controllable. Therefore:

$$\frac{1}{T}\sum_{t=0}^{T-1}\mathbf{E}\|M\bar{X}_t\|_2^4 \leq 3\operatorname{tr}(M^\mathsf{T}M\Gamma_{T-1})^2$$

$$= 3\operatorname{tr}(M\Gamma_{\lfloor T/2\rfloor}^{1/2}\Gamma_{\lfloor T/2\rfloor}^{-1/2}\Gamma_{T-1}\Gamma_{\lfloor T/2\rfloor}^{-1/2}\Gamma_{\lfloor T/2\rfloor}^{1/2}M^\mathsf{T})^2$$

$$\leq 3\|\Gamma_{\lfloor T/2\rfloor}^{-1}\Gamma_{T-1}\|_{\mathsf{op}}^2 \operatorname{tr}(M^\mathsf{T}M\Gamma_{\lfloor T/2\rfloor})^2$$

$$\leq 27\|\Gamma_{\lfloor T/2\rfloor}^{-1}\Gamma_{T-1}\|_{\mathsf{op}}^2 \operatorname{tr}\left(M^\mathsf{T}M \cdot \frac{1}{T}\sum_{t=0}^{T-1}\Gamma_t\right)^2 \quad \text{using (48)}$$

$$= 27\|\Gamma_{\lfloor T/2\rfloor}^{-1}\Gamma_{T-1}\|_{\mathsf{op}}^2 \left(\frac{1}{T}\sum_{t=0}^{T-1}\mathbf{E}\|MX_t\|_2^2\right)^2$$

$$\leq 108\|\Gamma_{\lfloor T/2\rfloor}^{-1}\Gamma_{T-1}\|_{\mathsf{op}}^2 \left(\frac{1}{T}\sum_{t=0}^{T-1}\mathbf{E}\|M\bar{X}_t\|_2^2\right)^2 \quad \text{using Proposition G.4.}$$

Since the matrix $M$ is arbitrary, the claim follows using the following bound for $\|\Gamma_{\lfloor T/2\rfloor}^{-1}\Gamma_{T-1}\|_{\mathsf{op}}^2$:

$$\|\Gamma_{\lfloor T/2\rfloor}^{-1}\Gamma_{T-1}\|_{\mathsf{op}}^2 \leq \frac{\tau^4\|H\|_{\mathsf{op}}^4}{(1-\rho^2)^2\mu^2} \leq \frac{\tau^4\|H\|_{\mathsf{op}}^4}{(1-\rho)^2\mu^2}.$$

∎

### I.2 Bounding the dependency matrix for truncated LDS

We control $\|\Gamma_{\mathsf{dep}}(\mathsf{P}_{\bar{X}})\|_{\mathsf{op}}$ by a direct computation of the mixing properties of the original Gaussian process (12).

**Proposition I.2.** *Consider the process $\{\bar{X}_t\}_{t\geq 0}$ from (44), and let $\mathsf{P}_{\bar{X}}$ denote the joint distribution of $\{\bar{X}_t\}_{t=0}^{T-1}$. We have that:*

$$\|\Gamma_{\mathsf{dep}}(\mathsf{P}_{\bar{X}})\|_{\mathsf{op}} \leq 5\kappa + \frac{22}{1-\rho}\log\left(\frac{\tau^2}{4\mu}\left[B_{\bar{X}}^2 + \frac{d_X\|H\|_{\mathsf{op}}^2}{1-\rho}\right]\right).$$

*Proof.* We first construct an almost sure bound on the process $\{\bar{X}_t\}_{t\geq 0}$. Indeed, for any $t \geq 0$, using the $(\tau, \rho)$-stability of $A_\star$:

$$\|\bar{X}_t\|_2 \leq \frac{\|H\|_{\mathsf{op}}\tau R}{1-\rho} = \frac{\|H\|_{\mathsf{op}}\tau(\sqrt{d_X} + \sqrt{2(1+\beta)\log T})}{1-\rho} = B_{\bar{X}}.$$

Also, by $(\tau, \rho)$-stability, we have for any indices $s \leq t$:

$$\|\Gamma_s - \Gamma_t\|_{\mathsf{op}} = \left\|\sum_{k=s+1}^{t}A^k HH^\mathsf{T}(A^k)^\mathsf{T}\right\|_{\mathsf{op}} \leq \|H\|_{\mathsf{op}}^2\tau^2\sum_{k=s+1}^{t}\rho^{2k} \leq \frac{\|H\|_{\mathsf{op}}^2\tau^2}{1-\rho^2}\rho^{2(s+1)}. \tag{49}$$

The marginal and conditional distributions of $\{X_t\}_{t\geq 0}$ are easily characterized. We have that $X_t \sim N(0, \Gamma_t)$. Furthermore, $X_t \mid X_0 = x$ for $t \geq 1$ is distributed as $N(A^t x, \Gamma_{t-1})$. So now for any $t \geq 0$ and $k \geq 1$:

$$\mathsf{P}_{X_{t+k}}(\cdot \mid X_t = x) = N(A^k x, \Gamma_{k-1}), \ \ \mathsf{P}_{X_{t+k}} = N(0, \Gamma_{t+k}).$$

Now suppose $k \geq \kappa$. The matrices $\Gamma_{k-1}$ and $\Gamma_{t+k}$ will both be invertible, so the two distributions are mutually absolutely continuous. We can then use the closed-form expression for the KL-divergence between two multivariate Gaussians:

$\mathsf{KL}(N(A^k x, \Gamma_{k-1}), N(0, \Gamma_{t+k}))$

$$= \frac{1}{2} \left[ \mathrm{tr}(\Gamma_{t+k}^{-1}\Gamma_{k-1}) + x^\mathsf{T}(A^k)^\mathsf{T}\Gamma_{t+k}^{-1}A^k x - d_\mathsf{X} + \log\det(\Gamma_{t+k}\Gamma_{k-1}^{-1}) \right]$$

$$\leq \frac{1}{2} x^\mathsf{T}(A^k)^\mathsf{T}\Gamma_{t+k}^{-1}A^k x + \frac{d_\mathsf{X}}{2} \log\|\Gamma_{t+k}\Gamma_{k-1}^{-1}\|_{\mathsf{op}} \qquad\qquad \text{since } \Gamma_{k-1} \preccurlyeq \Gamma_{t+k}$$

$$\leq \frac{\tau^2\rho^{2k}\|x\|_2^2}{2\mu} + \frac{d_\mathsf{X}}{2}\log\left(1 + \frac{\|\Gamma_{t+k} - \Gamma_{k-1}\|_{\mathsf{op}}}{\mu}\right) \qquad\qquad \text{using } (\tau,\rho)\text{-stability}$$

$$\leq \frac{\tau^2\rho^{2k}\|x\|_2^2}{2\mu} + \frac{d_\mathsf{X}}{2}\log\left(1 + \frac{\|H\|_{\mathsf{op}}^2\tau^2}{(1-\rho^2)\mu}\rho^{2k}\right) \qquad\qquad \text{using (49)}$$

$$\leq \left[\frac{\tau^2\|x\|_2^2}{2\mu} + \frac{d_\mathsf{X}\|H\|_{\mathsf{op}}^2\tau^2}{2(1-\rho^2)\mu}\right]\rho^{2k} \qquad\qquad \log(1+x) \leq x\,\forall x \geq 0.$$

Hence by Pinsker's inequality [see e.g. 1, Lemma 2.5], whenever $k \geq \kappa$:

$$\|\mathsf{P}_{X_{t+k}}(\cdot \mid X_t = x) - \mathsf{P}_{X_{t+k}}\|_{\mathsf{TV}} \leq \sqrt{\mathsf{KL}(N(A^k x, \Gamma_{k-1}), N(0, \Gamma_{t+k}))/2}$$

$$\leq \sqrt{\frac{\tau^2\|x\|_2^2}{4\mu} + \frac{d_\mathsf{X}\|H\|_{\mathsf{op}}^2\tau^2}{4(1-\rho^2)\mu}}\,\rho^k.$$

By Proposition G.2 (which we can invoke since we constrained $\beta \geq 2$), for any $\ell \in \mathbb{N}$:

$$\|\Gamma_{\mathsf{dep}}(\mathsf{P}_{\bar{X}})\|_{\mathsf{op}} \leq 3 + \sqrt{2}\sum_{k=1}^{T-1} \max_{t=0,\ldots,T-1-k} \operatorname*{ess\,sup}_{x\in\bar{\mathsf{X}}_t} \sqrt{\|\mathsf{P}_{X_{t+k}}(\cdot \mid X_t = x) - \mathsf{P}_{X_{t+k}}\|_{\mathsf{TV}}}$$

$$\leq 3 + \sqrt{2}(\kappa - 1 + \ell) + \sum_{k=\kappa+\ell}^{T-1}\left[\frac{\tau^2 B_{\bar{X}}^2}{4\mu} + \frac{d_\mathsf{X}\|H\|_{\mathsf{op}}^2\tau^2}{4(1-\rho^2)\mu}\right]^{1/4}\rho^{k/2}$$

$$\leq 5(\kappa + \ell) + \left[\frac{\tau^2 B_{\bar{X}}^2}{4\mu} + \frac{d_\mathsf{X}\|H\|_{\mathsf{op}}^2\tau^2}{4(1-\rho^2)\mu}\right]^{1/4}\frac{\rho^{(\kappa+\ell)/2}}{1-\rho^{1/2}}.$$

Now, define $\psi \triangleq \frac{\tau^2 B_{\bar{X}}^2}{4\mu} + \frac{d_\mathsf{X}\|H\|_{\mathsf{op}}^2\tau^2}{4(1-\rho^2)\mu}$. We choose $\ell = \max\left\{\left\lceil\frac{\log(\psi^{1/4})}{1-\rho^{1/2}}\right\rceil - \kappa, 0\right\}$, so $\rho^{(\kappa+\ell)/2} \leq 1/\psi^{1/4}$. With this choice of $\ell$ and the observation that $\inf_{x\in[0,1]}\frac{1-\sqrt{x}}{1-x} = \frac{1}{2}$,

$$\|\Gamma_{\mathsf{dep}}(\mathsf{P}_{\bar{X}})\|_{\mathsf{op}} \leq 5\kappa + \frac{11\log\psi}{4(1-\rho^{1/2})} \leq 5\kappa + \frac{22\log\psi}{1-\rho}.$$

The claim now follows. ∎

## I.3 Finishing the proof of Theorem 6.1

For what follows, $c_i$ will denote universal positive constants whose values remain unspecified.

For any $\varepsilon > 0$ and $r > 0$, we now construct an $\varepsilon$-covering of $\partial B(r)$ with $\mathscr{F}_\star$ the offset class of $\mathscr{F}$ from (13). To this end, we let $\{A_1, \ldots, A_N\}$ be a $\delta$-cover of $\mathscr{A} \triangleq \{A \in \mathbb{R}^{d_\mathsf{X} \times d_\mathsf{X}} \mid \|A\|_F \leq B\}$ for $\delta$ to be specified. By a volumetric argument we may choose $\{A_1, \ldots, A_N\}$ such that $N \leq \left(1 + \frac{2B}{\delta}\right)^{d_\mathsf{X}^2}$. Now, any realization of $\{\bar{X}_t\}$ will have norm less than $B_{\bar{X}}$, where $B_{\bar{X}}$ is given by (47) and satisfies

$$B_{\bar{X}} \leq c_0\frac{\|H\|_{\mathsf{op}}\tau(\sqrt{d_\mathsf{X}} + \sqrt{(1+\beta)\log T})}{1-\rho}.$$

Let $A \in \mathscr{A}$, and let $A_i$ denote an element in the covering satisfying $\|A - A_i\|_F \leq \delta$. For any $x$ satisfying $\|x\|_2 \leq B_{\bar{X}}$:

$$\|(A_i x - A_\star x) - (A x - A_\star x)\|_F = \|(A_i - A) x\|_2 \leq \|A_i - A\|_F \|x\|_2 \leq \delta B_{\bar{X}}.$$

Thus, it suffices to take $\delta = \varepsilon / B_{\bar{X}}$ to construct an $\varepsilon$-covering of $\mathscr{F}_\star$ over $\{\bar{X}_t\}$, which shows that $\mathcal{N}_\infty(\mathscr{F}_\star, \varepsilon) \leq \left(1 + \frac{2BB_{\bar{X}}}{\varepsilon}\right)^{d_X^2}$. Since $\partial B(r) \subset \mathscr{F}_\star$, we have the following inequality [see e.g. 39, Exercise 4.2.10]:

$$\mathcal{N}_\infty(\partial B(r), \varepsilon) \leq \mathcal{N}_\infty(\mathscr{F}_\star, \varepsilon/2) \leq \left(1 + \frac{4BB_{\bar{X}}}{\varepsilon}\right)^{d_X^2}.$$

By Proposition I.1, $(\mathscr{F}_\star, \mathsf{P}_{\bar{X}})$ is $(C_{\mathsf{LDS}}, 2)$-hypercontractive for all $T \geq \max\{6, 2\kappa\}$, with

$$C_{\mathsf{LDS}} = \frac{108\tau^4 \|H\|_{\mathsf{op}}^4}{(1-\rho)^2 \mu^2}.$$

Also by Proposition I.2,

$$\|\Gamma_{\mathsf{dep}}(\mathsf{P}_{\bar{X}})\|_{\mathsf{op}}^2 \leq c_1 \kappa^2 + \frac{c_2}{(1-\rho)^2} \log^2\left(\frac{\tau^2}{4\mu}\left[B_{\bar{X}}^2 + \frac{d_X \|H\|_{\mathsf{op}}^2}{1-\rho}\right]\right) \triangleq \gamma^2.$$

Since $\mathscr{F}_\star$ is convex and contains the zero function, it is also star-shaped. Furthermore, on the truncated process (44), the class $\mathscr{F}_\star$ is $2BB_{\bar{X}}$-bounded. Invoking Theorem 4.1, we thus have for every $r > 0$ that

$$\mathbf{E}\|\bar{f} - f_\star\|_{L^2}^2 \leq 8\mathbf{E}\bar{\mathsf{M}}_T(\mathscr{F}_\star) + r^2 + 4B^2 B_{\bar{X}}^2 \left(1 + \frac{4\sqrt{8}BB_{\bar{X}}}{r}\right)^{d_X^2} \exp\left(\frac{-T}{8C_{\mathsf{LDS}}\gamma^2}\right). \quad (50)$$

Here, the notation $\mathbf{E}\bar{\mathsf{M}}_T(\mathscr{F}_\star)$ is meant to emphasize that the offset complexity is with respect to the truncated process $\mathsf{P}_{\bar{X}}$ and *not* the original process $\mathsf{P}_X$. We now set $r^2 = \|H\|_{\mathsf{op}}^2 d_X^2 / T$, and compute a $T_0$ such that the third term in (50) is also bounded by $\|H\|_{\mathsf{op}}^2 d_X^2 / T$. To do this, it suffices to compute $T_0$ such that for all $T \geq T_0$:

$$T \geq c_3 C_{\mathsf{LDS}} \gamma^2 d_X^2 \log\left(\frac{TBB_{\bar{X}}}{\|H\|_{\mathsf{op}}\sqrt{d_X}}\right).$$

Thus it suffices to set $T_0$ as (provided that $\beta$ is at most polylogarithmic in the problem constants—we later make such a choice):

$$T_0 = c_4 \frac{\tau^4 \|H\|_{\mathsf{op}}^4 d_X^2}{(1-\rho)^2 \mu^2}\left[\kappa^2 + \frac{1}{(1-\rho)^2}\right] \mathrm{polylog}\left(B, d_X, \tau, \|H\|_{\mathsf{op}}, \frac{1}{\mu}, \frac{1}{1-\rho},\right). \quad (51)$$

We do not attempt to compute the exact power of the polylog term; it can in principle be done via Du et al. [43, Lemma F.2].

Next, by (46), $\mathbf{E}\|\widehat{f} - f_\star\|_{L^2}^2 \mathbf{1}\{\mathcal{E}^c\} \leq \frac{4\sqrt{3}B^2\|H\|_{\mathsf{op}}^2\tau^2 d_X}{(1-\rho)T^{\beta/2}}$. Thus we also need to set $T_0$ large enough so that this term is bounded by $\|H\|_{\mathsf{op}}^2 d_X^2 / T$. To do this, it suffices to constrain $\beta > 2$ and set $T_0 \geq c_5 \left[\frac{B^2\tau^2}{1-\rho}\right]^{\frac{1}{\beta/2-1}}$. Hence, setting $\beta = \max\{4, c_6 \log B\}$ implies that (51) suffices.

Let us now upper bound $\mathbf{E}\bar{\mathsf{M}}_T(\mathscr{F}_\star)$ by $\mathbf{E}\mathsf{M}_T(\mathscr{F}_\star)$ plus $\|H\|_{\mathsf{op}}^2 d_X^2 / T$. Recall the definition of $\mathcal{E}$ from (45). We first write:

$$\begin{aligned}
\mathbf{E}\bar{\mathsf{M}}_T(\mathscr{F}_\star) &= \mathbf{E}\bar{\mathsf{M}}_T(\mathscr{F}_\star)\mathbf{1}\{\mathcal{E}\} + \mathbf{E}\bar{\mathsf{M}}_T(\mathscr{F}_\star)\mathbf{1}\{\mathcal{E}^c\} \\
&= \mathbf{E}\mathsf{M}_T(\mathscr{F}_\star)\mathbf{1}\{\mathcal{E}\} + \mathbf{E}\bar{\mathsf{M}}_T(\mathscr{F}_\star)\mathbf{1}\{\mathcal{E}^c\} \\
&\leq \mathbf{E}\mathsf{M}_T(\mathscr{F}_\star) + \mathbf{E}\bar{\mathsf{M}}_T(\mathscr{F}_\star)\mathbf{1}\{\mathcal{E}^c\}.
\end{aligned}$$

The last inequality holds since it can be checked that $\mathsf{M}_T(\mathscr{F}_\star) \geq 0$ (i.e., by lower bounding the supremum with the zero function which is in $\mathscr{F}_\star$ since $\mathscr{F}$ contains $f_\star$). Furthermore, an elementary linear algebra calculation yields that we can upper bound $\bar{\mathsf{M}}_T(\mathscr{F}_\star)$ deterministically by:

$$\bar{\mathsf{M}}_T(\mathscr{F}_\star) \leq \frac{4}{T} \left\| \left( \left( \sum_{t=0}^{T-1} \bar{X}_t \bar{X}_t^\mathsf{T} \right)^\dagger \right)^{1/2} \sum_{t=0}^{T-1} \bar{X}_t \bar{V}_t^\mathsf{T} H^\mathsf{T} \right\|_F^2 \leq \frac{4}{T} \sum_{t=0}^{T-1} \| H \bar{V}_t \|_2^2$$

Here, the $\dagger$ notation refers to the Moore-Penrose pseudo-inverse. Therefore taking expectations:

$$\mathbf{E}\bar{M}_T(\mathscr{F}_\star)\mathbf{1}\{\mathcal{E}^c\} \leq \frac{4}{T} \sum_{t=0}^{T-1} \mathbf{E}\|H\bar{V}_t\|_2^2 \mathbf{1}\{\mathcal{E}^c\} \leq \frac{4}{T} \sum_{t=0}^{T-1} \mathbf{E}\|HV_t\|_2^2 \mathbf{1}\{\mathcal{E}^c\} \overset{(a)}{\leq} \frac{4}{T^{1+\beta/2}} \sum_{t=0}^{T-1} \sqrt{\mathbf{E}\|HV_t\|_2^4}$$

$$\overset{(b)}{\leq} \frac{4\sqrt{3}}{T^{1+\beta/2}} \sum_{t=0}^{T-1} \mathbf{E}\|HV_t\|_2^2 = \frac{4\sqrt{3}\|H\|_F^2}{T^{\beta/2}} \leq \frac{4\sqrt{3}d_\mathsf{X}\|H\|_{\mathsf{op}}^2}{T^{\beta/2}}.$$

Here, (a) is Cauchy-Schwarz, and (b) follows from Proposition G.5. This last term will be bounded by $\|H\|_{\mathsf{op}}^2 d_\mathsf{X}^2/T$ as soon as $T \geq 4\sqrt{3}$, since we set $\beta \geq 4$. The claim now follows. ∎

### I.4 Further discussion related to Theorem 6.1

We first discuss the rate (14) prescribed by Theorem 6.1. A simple computation shows that the martingale complexity $\mathbf{E}\mathsf{M}_T(\hat{\mathscr{F}}_\star)$ can be upper bounded by $1/T$ times the self-normalized martingale term which typically appears in the analysis of least-squares [44]. Specifically, when the empirical covariance matrix $\sum_{t=0}^{T-1} X_t X_t^\mathsf{T}$ is invertible:

$$\mathbf{E}\mathsf{M}_T(\mathscr{F}_\star) \leq \frac{4}{T}\mathbf{E}\left\| \left( \sum_{t=0}^{T-1} X_t X_t^\mathsf{T} \right)^{-1/2} \sum_{t=0}^{T-1} X_t V_t^\mathsf{T} H^\mathsf{T} \right\|_F^2 .$$

A sharp analysis of this self-normalized martingale term [19, Lemma 4.1] shows that $\mathbf{E}\mathsf{M}_T(\mathscr{F}_\star) \lesssim \frac{\|H\|_{\mathsf{op}}^2 d_\mathsf{X}^2}{T}$, and hence (14) yields the minimax optimal rate up to constant factors after a polynomial burn-in time.[8] This is unlike the chaining bound (7) which yields extra logarithmic factors [see e.g. 15, Lemma 4]. Note that the burn-in time of $\tilde{O}(d_\mathsf{X}^2)$ given by our result is sub-optimal by a factor of $d_\mathsf{X}$. This extra factor comes from the union bound over a Frobenius norm ball of $d_\mathsf{X} \times d_\mathsf{X}$ matrices in Theorem 4.1.

To convert (14) into a parameter recovery bound, we simply lower bound the excess risk:

$$\mathbf{E}\|\hat{f} - f_\star\|_{L^2}^2 \geq \mathbf{E}\|\hat{A} - A_\star\|_F^2 \lambda_{\min}(\bar{\Gamma}_T) \implies \mathbf{E}\|\hat{A} - A_\star\|_F^2 \lesssim \frac{\|H\|_{\mathsf{op}}^2 d_\mathsf{X}^2}{T \lambda_{\min}(\bar{\Gamma}_T)}, \qquad (52)$$

where $\bar{\Gamma}_T \triangleq \frac{1}{T} \sum_{t=0}^{T-1} \mathbf{E}[X_t X_t^\mathsf{T}]$ is the average covariance matrix. The rate (52) recovers, after the polynomial burn-in time, existing results [16, 18, 19, 29] for stable systems, with a few caveats. First, most of the existing results are given in operator instead of Frobenius norm. We ignore this issue, since the only difference is the extra unavoidable factor of $d_\mathsf{X}$ in the rate for the Frobenius norm compared to the operator norm rate. Second, since Theorem 4.1 ultimately relies on some degree of ergodicity for the covariate process $\{X_t\}_{t\geq0}$, we cannot handle the marginally stable case (where $A_\star$ is allowed to have spectral radius equal to one) as in Simchowitz et al. [16], Sarkar and Rakhlin [18], Tu et al. [19], nor the unstable case as in Faradonbeh et al. [17], Sarkar and Rakhlin [18].

We conclude with a short discussion on the proof of Theorem 6.1. As the LDS process (12) is unbounded, we use the truncation argument outlined in Appendix B.1 so that Theorem 4.1 still applies. Furthermore, since the process (12) is jointly Gaussian, the dependency matrix coefficients are simple to bound, resulting in polynomial rates (15) for the burn-in time. A much wider variety of non-Gaussian noise distributions can be handled via ergodic theory for Markov chains [see e.g. 32, Chapter 15]. While these results typically do not offer explicit expressions for the mixing coefficients, both Douc et al. [46] and Hairer and Mattingly [47] provide a path forward for deriving explicit bounds. We however omit these calculations in the interest of simplicity.

---

[8] While the burn-in time is polynomial in the problem constants listed in (15), Tsiamis and Pappas [45] show that these constants (specifically $1/\lambda_{\min}(\Gamma_{\kappa-1})$) can scale exponentially in $\kappa$, the controllability index of the system.

# J   General linearized model dynamics

## J.1   Comparison to existing results

We first compare the results of Theorem 6.2 to the existing bounds from Kowshik et al. [20], Sattar and Oymak [30], Foster et al. [31]. Before doing so, we note that these existing results bound the loss of specific gradient based algorithms. On the other hand, Theorem 6.2 directly applies to the empirical risk minimizer of the square loss. In general, the LSE optimization problem specialized to this setting is non-convex due to the composition of the square loss with the link function $\sigma$. However, we believe it should be possible to show that the quasi-Newton method described in Kowshik et al. [20, Algorithm 1] can be used to optimize the empirical risk to precision of order $\|H\|_{\mathsf{op}}^2 d_{\mathsf{X}}^2/T$, in which case a simple modification of Theorem 4.1 combined with the current analysis in Theorem 6.2 would apply to bound the excess risk of the final iterate of this algorithm. This is left to future work.

For our comparison, we will ignore all logarithmic factors, and assume any necessary burn-in times, remarking that the existing results all prescribe sharper burn-in times than Theorem 6.2. First, we compare with Sattar and Oymak [30, Corollary 6.2]. In doing so, we will assume that $H = (1 + \sigma)I$ for some $\sigma > 0$, since this is the setting they study. When $H$ is diagonal, (66) is actually invariant to the noise scale $\sigma$ which is the correct behavior: $\mathbf{E}\|\widehat{A} - A_\star\|_F^2 \leq \tilde{O}(1)\frac{d_{\mathsf{X}}^2}{\zeta^2 T}$. On the other hand, Sattar and Oymak [30, Corollary 6.2] gives a high probability bound of $\|\widehat{A} - A_\star\|_F^2 \leq \tilde{O}(1)\frac{\sigma^2 d_{\mathsf{X}}^2}{\zeta^4(1-\rho)^3 T}$. Thus, (66) improves on this rate by not only a factor of $1/\zeta^2$, but also in moving the $1/(1-\rho)$ dependence into the log. We note that their result seems to improve as $\sigma \to 0$, but the probability of success also tends to 0 as $\sigma \to 0$.

Next, we turn our attention to Foster et al. [31, Theorem 2, fast rate]. This result actually gives both in-sample excess risk and parameter recovery bounds. For simplicity, we only compare to the parameter recovery bounds, as this is their sharper result. Their result yields a high probability bound that $\|\widehat{A} - A_\star\|_F^2 \leq \tilde{O}(1)\frac{\|H\|_{\mathsf{op}}^2\|P_\star\|_{\mathsf{op}} d_{\mathsf{X}}^2}{\zeta^4(1-\rho)T}$. Again, we see (66) improve this rate by a factor of $1/\zeta^2$, and moves the dependence on $\|P_\star\|_{\mathsf{op}}$ and $1/(1-\rho)$ into the logarithm. We note again, this rate seems to improve as $\|H\|_{\mathsf{op}} \to 0$, but the number of iterations $m$ of GLMtron needed tends to $\infty$ as $\|H\|_{\mathsf{op}} \to 0$. We conclude by noting that the rate of Foster et al. [31] does not have any burn-in times.

Finally, we compare to Kowshik et al. [20, Theorem 1]. We will assume that $H = \sigma I$ again, as this is the setting of their work. Their parameter recovery bound states that with high probability, $\|\widehat{A} - A_\star\|_F^2 \leq \tilde{O}(1)\frac{\sigma^2 d_{\mathsf{X}}^2}{\zeta^2 T}$, which matches (66) up to the log factors. As noted previously, their logarithmic dependencies are sharper than ours. Furthermore, their result can also handle the unstable regime when $\rho \leq 1 + O(1/T)$, which ours cannot. However, we note that Theorem 6.2 also bounds $L^2$ excess risk with logarithmic dependence on $1/(1-\rho)$, which is not an immediate consequence of parameter error bounds. Indeed, a naïve upper bound using the 1-Lipschitz property of the link function yields: $\|\widehat{f} - f_\star\|_{L^2}^2 \leq \|\widehat{A} - A_\star\|_{\mathsf{op}}^2 \frac{1}{T}\sum_{t=0}^{T-1} \mathbf{E}\|X_t\|_2^2 \lesssim \|\widehat{A} - A_\star\|_{\mathsf{op}}^2 \frac{1}{(1-\rho)^2}$. Hence, even if the parameter error only depends logarithmically on $1/(1-\rho)$, it does not immediately translate over to excess risk.

## J.2   Proof of Theorem 6.2

We first turn our attention to controlling the states $\{X_t\}_{t\geq 0}$ in expectation. Note that the Lyapunov assumption in Assumption 6.1 implies that for every $x \in \mathbb{R}^{d_{\mathsf{X}}}$:

$$\|\sigma(A_\star x)\|_{P_\star}^2 \leq \rho\|x\|_{P_\star}^2, \tag{53}$$

and hence the function $x \mapsto \|x\|_{P_\star}^2$ is a Lyapunov function (recall that $P_\star \succcurlyeq I$) which certifies exponential stability to the origin for the deterministic system $x_+ = \sigma(A_\star x)$.

**Proposition J.1.** *Consider the GLM process $\{X_t\}_{t\geq 0}$ from (16). Under Assumption 6.1:*

$$\sup_{t\in\mathbb{N}} \mathbf{E}\|X_t\|_2^4 \leq B_X^4, \;\; B_X \triangleq \frac{12\sqrt{2}\|H\|_{\mathsf{op}}\|P_\star\|_{\mathsf{op}}^{1/2}\sqrt{d_{\mathsf{X}}}}{1-\rho}. \tag{54}$$

*Proof.* For any $a, b \in \mathbb{R}$ and $\varepsilon > 0$, $(a + b)^4 \leq (1 + \varepsilon)^3 a^4 + (1 + 1/\varepsilon)^3 b^4$. Hence for any $\varepsilon > 0$:

$$\mathbf{E}\|X_t\|_{P_\star}^4 = \mathbf{E}\|\sigma(A_\star X_t) + HV_t\|_{P_\star}^4$$
$$\leq (1 + \varepsilon)^3 \mathbf{E}\|\sigma(A_\star X_t)\|_{P_\star}^4 + (1 + 1/\varepsilon)^3 \mathbf{E}\|HV_t\|_{P_\star}^4$$
$$\leq (1 + \varepsilon)^3 \rho^2 \mathbf{E}\|X_t\|_{P_\star}^4 + (1 + 1/\varepsilon)^3 \mathbf{E}\|HV_t\|_{P_\star}^4 \qquad \text{using (53).} \qquad (55)$$

Now, we first assume that $\rho \in [1/2, 1)$. For any $\varepsilon \in [0, 1]$, we have that $(1 + \varepsilon)^3 \leq 1 + 12\varepsilon$. Choosing $\varepsilon = \frac{1 - \rho^2}{24\rho^2}$, we have that $\varepsilon \leq 1$, and therefore continuing from (55):

$$\mathbf{E}\|X_t\|_{P_\star}^4 \leq (1 + 12\varepsilon)\mathbf{E}\|X_t\|_{P_\star}^4 + (1 + 1/\varepsilon)^3 \mathbf{E}\|HV_t\|_{P_\star}^4$$
$$= \frac{1 + \rho^2}{2}\mathbf{E}\|X_t\|_{P_\star}^4 + \frac{24^3}{(1 - \rho^2)^3}\mathbf{E}\|HV_t\|_{P_\star}^4$$
$$\leq \frac{1 + \rho^2}{2}\mathbf{E}\|X_t\|_{P_\star}^4 + \frac{3 \cdot 24^3}{(1 - \rho^2)^3}(\mathbf{E}\|HV_t\|_{P_\star}^2)^2 \qquad \text{using Proposition G.5}$$
$$\leq \frac{1 + \rho^2}{2}\mathbf{E}\|X_t\|_{P_\star}^4 + \frac{3 \cdot 24^3}{(1 - \rho^2)^3}\operatorname{tr}(H^\mathsf{T} P_\star H)^2.$$

Unrolling this recursion yields:

$$\mathbf{E}\|X_t\|_{P_\star}^4 \leq \frac{6 \cdot 24^3}{(1 - \rho^2)^4}\operatorname{tr}(H^\mathsf{T} P_\star H)^2.$$

We now handle the case when $\rho \in [0, 1/2]$. Setting $\varepsilon = 2^{1/3} - 1$ and starting from (55):

$$\mathbf{E}\|X_t\|_{P_\star}^4 \leq \frac{1}{2}\mathbf{E}\|X_t\|_{P_\star}^4 + 125\mathbf{E}\|HV_t\|_{P_\star}^4$$
$$\leq \frac{1}{2}\mathbf{E}\|X_t\|_{P_\star}^4 + 375\operatorname{tr}(H^\mathsf{T} P_\star H)^2.$$

Unrolling this recursion yields:

$$\mathbf{E}\|X_t\|_{P_\star}^4 \leq 750\operatorname{tr}(H^\mathsf{T} P_\star H)^2.$$

The claim now follows by taking the maximum of these two bounds and using the inequalities $\operatorname{tr}(H^\mathsf{T} P_\star H) \leq \|H\|_{\mathsf{op}}^2 \|P_\star\|_{\mathsf{op}} d_\mathsf{X}$ and $1 - \rho^2 \geq 1 - \rho$. ∎

This proof proceeds quite similarly to the linear dynamical systems proof given in Appendix I. We start again by defining the truncated GLM dynamics:

$$\bar{X}_{t+1} = \sigma(A_\star \bar{X}_t) + H\bar{V}_t, \quad \bar{X}_0 = H\bar{V}_0, \quad \bar{V}_t = V_t \mathbf{1}\{\|V_t\|_2 \leq R\}. \qquad (56)$$

We set $R = \sqrt{d_\mathsf{X}} + \sqrt{2(1 + \beta)\log T}$ where $\beta \geq 2$ is a free parameter. Define the event $\mathcal{E}$ as:

$$\mathcal{E} := \left\{\max_{0 \leq t \leq T-1}\|V_t\|_2 \leq R\right\}. \qquad (57)$$

Note that by the setting of $R$, we have $\mathbf{P}(\mathcal{E}^c) \leq 1/T^\beta$ using standard Gaussian concentration results plus a union bound. Furthermore on $\mathcal{E}$, the original GLM process driven by Gaussian noise (16) coincides with the truncated process (56). Let $\hat{f}$ denote the LSE on the original process (56), and let $\bar{f}$ denote the LSE on the truncated process (56). Hence:

$$\mathbf{E}\|\hat{f} - f_\star\|_{L^2}^2 = \mathbf{E}\|\hat{f} - f_\star\|_{L^2}^2 \mathbf{1}\{\mathcal{E}\} + \mathbf{E}\|\hat{f} - f_\star\|_{L^2}^2 \mathbf{1}\{\mathcal{E}^c\}$$
$$\leq \mathbf{E}\|\bar{f} - f_\star\|_{L^2}^2 + \mathbf{E}\|\hat{f} - f_\star\|_{L^2}^2 \mathbf{1}\{\mathcal{E}^c\}. \qquad (58)$$

Let us now control the error term $\mathbf{E}\|\hat{f} - f_\star\|_{L^2}^2 \mathbf{1}\{\mathcal{E}^c\}$. Write $\hat{f}(x) = \sigma(\hat{A}x)$, and put $\hat{\Delta} = \hat{A} - A_\star$. We have:

$$\mathbf{E}\|\hat{f} - f_\star\|_{L^2}^2 \mathbf{1}\{\mathcal{E}^c\} = \frac{1}{T}\sum_{t=0}^{T-1}\mathbf{E}\|\sigma(\hat{A}X_t) - \sigma(A_\star X_t)\|_2^2 \mathbf{1}\{\mathcal{E}^c\} \overset{(a)}{\leq} \frac{1}{T}\sum_{t=0}^{T-1}\mathbf{E}\|\hat{\Delta}X_t\|_2^2 \mathbf{1}\{\mathcal{E}^c\}$$
$$\overset{(b)}{\leq} \frac{4B^2}{T}\sum_{t=0}^{T-1}\mathbf{E}\|X_t\|_2^2 \mathbf{1}\{\mathcal{E}^c\} \overset{(c)}{\leq} \frac{4B^2}{T^{1+\beta/2}}\sum_{t=0}^{T-1}\sqrt{\mathbf{E}\|X_t\|_2^4} \overset{(d)}{\leq} \frac{4B^2 B_X^2}{T^{\beta/2}}. \qquad (59)$$

Here, (a) follows since $\sigma$ is 1-Lipschitz, (b) uses the definition of $\mathscr{F}$ in (17), (c) follows by Cauchy-Schwarz, and (d) uses Proposition J.1.

The remainder of the proof is to bound the LSE error $\mathbf{E}\|\bar{f} - f_\star\|_{L^2}^2$. First, we establish an almost sure bound on $\{\bar{X}_t\}_{t \geq 0}$.

**Proposition J.2.** *Consider the truncated GLM process* (56). *Under Assumption 6.1, the process* $\{\bar{X}_t\}_{t \geq 0}$ *satisfies:*

$$\sup_{t \in \mathbb{N}} \|\bar{X}_t\|_{P_\star} \leq \frac{2\|P_\star\|_{\mathsf{op}}^{1/2}\|H\|_{\mathsf{op}}(\sqrt{d_{\mathsf{X}}} + \sqrt{2(1+\beta)\log T})}{1 - \rho} \triangleq B_{\bar{X}}. \tag{60}$$

*Proof.* By triangle inequality and (53):
$$\|\bar{X}_{t+1}\|_{P_\star} = \|\sigma(A_\star \bar{X}_t) + H\bar{V}_t\|_{P_\star} \leq \|\sigma(A_\star \bar{X}_t)\|_{P_\star} + \|H\bar{V}_t\|_{P_\star}$$
$$\leq \rho^{1/2}\|\bar{X}_t\|_{P_\star} + \|H\bar{V}_t\|_{P_\star} \leq \rho^{1/2}\|\bar{X}_t\|_{P_\star} + \|P_\star^{1/2}H\|_{\mathsf{op}}R.$$

Unrolling this recursion, and using the fact that $\inf_{x \in [0,1]} \frac{1 - \sqrt{x}}{1 - x} = 1/2$ yields the result. ∎

We next establish uniform bounds for the covariance matrices of the truncated process.

**Proposition J.3.** *Suppose $T \geq 4$. Consider the truncated GLM process* (56)*, and let the covariance matrices for the process* $\{\bar{X}_t\}_{t \geq 0}$ *be denoted as* $\bar{\Gamma}_t \triangleq \mathbf{E}[\bar{X}_t \bar{X}_t^\mathsf{T}]$. *Under Assumption 6.1:*

$$\frac{1}{2} HH^\mathsf{T} \preccurlyeq \bar{\Gamma}_t \preccurlyeq B_{\bar{X}}^2 \cdot I.$$

*Proof.* The upper bound is immediate from Proposition J.2, since $\mathbf{E}[\bar{X}_t \bar{X}_t^\mathsf{T}] \preccurlyeq \mathbf{E}[\|\bar{X}_t\|_2^2]I \preccurlyeq B_{\bar{X}}^2 I$. For the lower bound, it is immediate when $t = 0$ using Proposition G.4. On the other hand, for $t \geq 1$, since $\bar{V}_t$ is zero-mean:
$$\mathbf{E}[\bar{X}_t \bar{X}_t^\mathsf{T}] = \mathbf{E}[(\sigma(A_\star \bar{X}_{t-1}) + H\bar{V}_{t-1})(\sigma(A_\star \bar{X}_{t-1}) + H\bar{V}_{t-1})^\mathsf{T}]$$
$$= \mathbf{E}[\sigma(A_\star \bar{X}_{t-1})\sigma(A_\star \bar{X}_{t-1})^\mathsf{T}] + \mathbf{E}[H\bar{V}_{t-1}\bar{V}_{t-1}^\mathsf{T}H^\mathsf{T}] \succcurlyeq \mathbf{E}[H\bar{V}_{t-1}\bar{V}_{t-1}^\mathsf{T}H^\mathsf{T}] \succcurlyeq \frac{1}{2}HH^\mathsf{T}.$$

The last inequality again holds from Proposition G.4. ∎

### J.2.1 Trajectory hypercontractivity for truncated GLM

For our purposes, the link function assumption in Assumption 6.1 ensures the following approximate isometry inequality which holds for all $x \in \mathbb{R}^{d_{\mathsf{X}}}$ and all matrices $A, A' \in \mathbb{R}^{d_{\mathsf{X}} \times d_{\mathsf{X}}}$:

$$\zeta^2 \|Ax - A'x\|_2^2 \leq \|\sigma(Ax) - \sigma(A'x)\|_2^2 \leq \|Ax - A'x\|_2^2. \tag{61}$$

This inequality is needed to establish trajectory hypercontractivity for $\mathscr{F}_\star$.

**Proposition J.4.** *Suppose that $T \geq 4$. Fix any matrix $A \in \mathbb{R}^{d_{\mathsf{X}} \times d_{\mathsf{X}}}$. Under Assumption 6.1, the truncated process* (56) *satisfies:*

$$\frac{1}{T} \sum_{t=0}^{T-1} \mathbf{E}\|\sigma(A\bar{X}_t) - \sigma(A_\star \bar{X}_t)\|_2^4 \leq \frac{4B_{\bar{X}}^4}{\sigma_{\min}(H)^4 \zeta^4} \left( \frac{1}{T} \sum_{t=0}^{T-1} \mathbf{E}\|\sigma(A\bar{X}_t) - \sigma(A_\star \bar{X}_t)\|_2^2 \right)^2. \tag{62}$$

*Hence, the function class $\mathscr{F}_\star$ with $\mathscr{F}$ defined in* (17) *satisfies the* $(C_{\mathsf{GLM}}, 2)$-*trajectory hypercontractivity condition with* $C_{\mathsf{GLM}} = \frac{4B_{\bar{X}}^4}{\sigma_{\min}(H)^4 \zeta^4}$.

*Proof.* Put $\Delta \triangleq A - A_\star$ and $M \triangleq \Delta^\mathsf{T}\Delta$. We have:
$$\mathbf{E}\|\Delta \bar{X}_t\|_2^4 = \mathbf{E}[\bar{X}_t^\mathsf{T} M \bar{X}_t \bar{X}_t^\mathsf{T} M \bar{X}_t]$$
$$\begin{aligned}
&\leq B_{\bar{X}}^2 \operatorname{tr}(M^2 \bar{\Gamma}_t) && \text{using Proposition J.2} \\
&\leq B_{\bar{X}}^2 \|M\|_{\mathsf{op}} \operatorname{tr}(M\bar{\Gamma}_t) && \text{Hölder's inequality} \\
&\leq B_{\bar{X}}^2 \operatorname{tr}(M) \operatorname{tr}(M\bar{\Gamma}_t) && \text{since } M \text{ is positive semidefinite} \\
&\leq B_{\bar{X}}^4 \operatorname{tr}(M)^2 && \text{using Proposition J.3} \\
&\leq \frac{B_{\bar{X}}^4}{\lambda_{\min}(HH^\mathsf{T})^2} \operatorname{tr}(MHH^\mathsf{T})^2.
\end{aligned}$$

On the other hand, by Proposition J.3:

$$\mathbf{E}\|\Delta \bar{X}_t\|_2^2 = \text{tr}(M\bar{\Gamma}_t) \geq \frac{1}{2}\text{tr}(MHH^\mathsf{T}).$$

Combining these bounds yields:

$$\frac{1}{T}\sum_{t=0}^{T-1}\mathbf{E}\|\Delta \bar{X}_t\|_2^4 \leq \frac{B_{\bar{X}}^4}{\lambda_{\min}(HH^\mathsf{T})^2}\text{tr}(MHH^\mathsf{T})^2 \leq \frac{4B_{\bar{X}}^4}{\lambda_{\min}(HH^\mathsf{T})^2}\left(\frac{1}{T}\sum_{t=0}^{T-1}\mathbf{E}\|\Delta \bar{X}_t\|_2^2\right)^2.$$

The claim now follows via the approximate isometry inequality (61). ∎

### J.2.2 Bounding the dependency matrix for truncated GLM

We will use the result in Lemma G.1 to bound the total-variation distance by the 1-Wasserstein distance. This is where the non-degenerate noise assumption in Assumption 6.1 is necessary.

The starting point is the observation that the diagonal Lyapunov function in Assumption 6.1 actually yields *incremental stability* [48] in addition to Lyapunov stability. In particular, let $\{a_i\}$ denote the rows of $A_\star$. For any $x, x' \in \mathbb{R}^{d_\mathsf{X}}$:

$$\begin{aligned}
\|\sigma(A_\star x) - \sigma(A_\star x')\|_{P_\star}^2 &= \sum_{i=1}^{d_\mathsf{X}}(P_\star)_{ii}(\sigma(\langle a_i, x\rangle) - \sigma(\langle a_i, x'\rangle))^2 \\
&\leq \sum_{i=1}^{d_\mathsf{X}}(P_\star)_{ii}(\langle a_i, x\rangle - \langle a_i, x'\rangle)^2 \\
&= (x - x')^\mathsf{T}A_\star^\mathsf{T}P_\star A_\star(x - x') \\
&\leq \rho\|x - x'\|_{P_\star}^2.
\end{aligned} \tag{63}$$

This incremental stability property allows us to control the dependency matrix as follows.

**Proposition J.5.** *Consider the truncated GLM process $\{\bar{X}_t\}_{t\geq 0}$ from (56). Let $\mathsf{P}_{\bar{X}}$ denote the joint distribution of $\{\bar{X}_t\}_{t=0}^{T-1}$. Under Assumption 6.1 and when $B \geq 1$, we have that:*

$$\|\Gamma_{\mathsf{dep}}(\mathsf{P}_{\bar{X}})\|_{\mathsf{op}} \leq \frac{22}{1-\rho}\log\left(\frac{B\sqrt{d_\mathsf{X}}(B_{\bar{X}} + B_X)}{2\sigma_{\min}(H)}\right).$$

*Proof.* Let $\{X_t\}_{t\geq 0}$ denote the original GLM dynamics from (16). Fix indices $t \geq 0$ and $k \geq 1$. We construct a coupling of $(\mathsf{P}_{X_{t+k}}(\cdot \mid X_t = x), \mathsf{P}_{X_{t+k}})$ as follows. Let $\{V_t\}_{t\geq 0}$ be iid $N(0, I)$. Let $\{Z_s\}_{s\geq t}$ be the process such that $Z_t = x$, and follows the GLM dynamics (16) using the noise $\{V_t\}_{t\geq 0}$ (we do not bother defining $Z_{t'}$ for $t' < t$ since we do not need it). Similarly, let $\{Z_s'\}_{s\geq 0}$ be the process following the GLM dynamics (16) using the same noise $\{V_t\}_{t\geq 0}$. Now we have:

$$\begin{aligned}
\mathbf{E}\|Z_{t+k} - Z_{t+k}'\|_{P_\star} &= \mathbf{E}\|\sigma(A_\star Z_{t+k-1}) - \sigma(A_\star Z_{t+k-1}')\|_{P_\star} \\
&\leq \rho^{1/2}\mathbf{E}\|Z_{t+k-1} - Z_{t+k-1}'\|_{P_\star} \qquad \text{using Equation (63)}.
\end{aligned}$$

We now unroll this recursion down to $t$:

$$\mathbf{E}\|Z_{t+k} - Z_{t+k}'\|_{P_\star} \leq \rho^{k/2}\mathbf{E}\|Z_t - Z_t'\|_{P_\star} = \rho^{k/2}\mathbf{E}\|x - Z_t'\|_{P_\star}.$$

Since $P_\star \succeq I$, this shows that:

$$W_1(\mathsf{P}_{X_{t+k}}(\cdot \mid X_t = x), \mathsf{P}_{X_{t+k}}) \leq \rho^{k/2}(\|x\|_{P_\star} + \mathbf{E}\|X_t\|_{P_\star}) \leq \rho^{k/2}(\|x\|_{P_\star} + B_X),$$

where the last inequality follows from Proposition J.1 and Jensen's inequality.

Next, it is easy to see the map $x \mapsto \sigma(A_\star x)$ is $\|A\|_{\mathsf{op}}$-Lipschitz. Furthermore, since $H$ is full rank by Assumption 6.1, then for any $t$ and $k \geq 1$ both $\mathsf{P}_t$ and $\mathsf{P}_{X_{t+k}}(\cdot \mid X_t = x)$ are absolutely continuous w.r.t. the Lebesgue measure in $\mathbb{R}^{d_\mathsf{X}}$. Using Lemma G.1, we have for any $k \geq 2$:

$$\begin{aligned}
\|\mathsf{P}_{X_{t+k}}(\cdot \mid X_t = x) - \mathsf{P}_{X_{t+k}}\|_{\mathsf{TV}} &\leq \frac{\|A_\star\|_{\mathsf{op}}\sqrt{\text{tr}((HH^\mathsf{T})^{-1})}}{2}W_1(\mathsf{P}_{X_{t+k-1}}(\cdot \mid X_t = x), \mathsf{P}_{X_{t+k-1}}) \\
&\leq \frac{\|A_\star\|_{\mathsf{op}}\sqrt{\text{tr}((HH^\mathsf{T})^{-1})}}{2}\rho^{(k-1)/2}(\|x\|_{P_\star} + B_X).
\end{aligned}$$

Using Proposition G.2 to bound $\|\Gamma_{\mathsf{dep}}(\mathsf{P}_{\bar{X}})\|_{\mathsf{op}}$ (which is valid because we constrained $\beta \geq 2$), and Proposition J.2 to bound $x \in \bar{\mathsf{X}}_t$, for any $\ell \geq 1$:

$$\|\Gamma_{\mathsf{dep}}(\mathsf{P}_{\bar{X}})\|_{\mathsf{op}} \leq 3 + \sqrt{2} \sum_{k=1}^{T-1} \max_{t=0,\dots,T-1-k} \operatorname*{ess\,sup}_{x \in \bar{\mathsf{X}}_t} \sqrt{\|\mathsf{P}_{X_{t+k}}(\cdot \mid X_t = x) - \mathsf{P}_{X_{t+k}}\|_{\mathsf{TV}}}$$

$$\leq 3 + \sqrt{2}\ell + \left[\frac{\|A_\star\|_{\mathsf{op}}\sqrt{\operatorname{tr}((HH^\mathsf{T})^{-1})}(B_{\bar{X}} + B_X)}{2}\right]^{1/2} \sum_{k=\ell+1}^{T-1} \rho^{(k-1)/4}$$

$$\overset{(a)}{\leq} 5\ell + \left[\frac{B\sqrt{d_{\mathsf{X}}}(B_{\bar{X}} + B_X)}{2\sigma_{\min}(H)}\right]^{1/2} \frac{\rho^{\ell/4}}{1-\rho^{1/4}}.$$

Above, (a) uses the bounds $\|A_\star\|_{\mathsf{op}} \leq B$ and $\operatorname{tr}(HH^{-1}) \leq d_{\mathsf{X}}/\sigma_{\min}(H)^2$. Now put $\psi \triangleq \frac{B\sqrt{d_{\mathsf{X}}}(B_{\bar{X}} + B_X)}{2\sigma_{\min}(H)}$. We choose $\ell = \left\lceil \frac{\log(\sqrt{\psi})}{1-\rho^{1/4}} \right\rceil$ so that $\rho^{\ell/4} \leq 1/\sqrt{\psi}$. This yields:

$$\|\Gamma_{\mathsf{dep}}(\mathsf{P}_{\bar{X}})\|_{\mathsf{op}} \leq \frac{11\log\psi}{2(1-\rho^{1/4})} \overset{(a)}{\leq} \frac{22\log\psi}{1-\rho} = \frac{22}{1-\rho}\log\left(\frac{B\sqrt{d_{\mathsf{X}}}(B_{\bar{X}} + B_X)}{2\sigma_{\min}(H)}\right).$$

Above, (a) follows from $\inf_{x \in [0,1]} \frac{1-x^{1/4}}{1-x} = 1/4$. ∎

### J.2.3  Finishing the proof of Theorem 6.2

Below, we let $c_i$ be universal positive constants that we do not track precisely.

For any $\varepsilon > 0$ we now construct an $\varepsilon$-covering of $\mathscr{F}_\star \setminus B(r)$, with $\mathscr{F}_\star$ the offset class of $\mathscr{F}$ from (17). Note that we are not covering $\partial B(r)$ since the class $\mathscr{F}_\star$ is not star-shaped. However, an inspection of the proof of Theorem B.2 shows that one can remove the star-shaped assumption by instead covering the set $\mathscr{F}_\star \setminus B(r)$. To this end, we let $\{A_1, \dots, A_N\}$ be a $\delta$-cover of $\mathscr{A} \triangleq \{A \in \mathbb{R}^{d_{\mathsf{X}} \times d_{\mathsf{X}}} \mid \|A\|_F \leq B\}$, for a $\delta$ to be specified. By a volumetric argument we may choose $\{A_1, \dots, A_N\}$ such that $N \leq \left(1 + \frac{2B}{\delta}\right)^{d_{\mathsf{X}}^2}$. Now, any realization of $\{\bar{X}_t\}$ will have norm less than $B_{\bar{X}}$ from (60), where $B_{\bar{X}}$ is bounded by:

$$B_{\bar{X}} \leq \frac{c_0\|P_\star\|_{\mathsf{op}}^{1/2}\|H\|_{\mathsf{op}}(\sqrt{d_{\mathsf{X}}} + \sqrt{(1+\beta)\log T})}{1-\rho}.$$

Now fix any $A \in \mathscr{A}$, and let $A_i$ be an element in the $\delta$-cover satisfying $\|A - A_i\|_F \leq \delta$. We observe that for any $x$ satisfying $\|x\|_2 \leq B_{\bar{X}}$:

$$\|(\sigma(A_ix) - \sigma(A_\star x)) - (\sigma(Ax) - \sigma(A_\star x))\|_2 = \|\sigma(A_ix) - \sigma(Ax)\|_2 \leq \|(A_i - A)x\|_2$$
$$\leq \|A_i - A\|_F \|x\|_2 \leq \delta B_{\bar{X}}.$$

Thus, it suffices to take $\delta = \varepsilon/B_{\bar{X}}$ to construct the $\varepsilon$ cover of $\mathscr{F}_\star$, i.e., $\mathcal{N}_\infty(\mathscr{F}_\star, \varepsilon) \leq \left(1 + \frac{2BB_{\bar{X}}}{\varepsilon}\right)^{d_{\mathsf{X}}^2}$. This then implies [see e.g. 39, Example 4.2.10]:

$$\mathcal{N}_\infty(\mathscr{F}_\star \setminus B(r), \varepsilon) \leq \mathcal{N}_\infty(\mathscr{F}_\star, \varepsilon/2) \leq \left(1 + \frac{4BB_{\bar{X}}}{\varepsilon}\right)^{d_{\mathsf{X}}^2}.$$

Next, by Proposition J.4, $(\mathscr{F}_\star, \mathsf{P}_{\bar{X}})$ is $(C_{\mathsf{GLM}}, 2)$-hypercontractive for all $T \geq 4$, where

$$C_{\mathsf{GLM}} \leq \frac{4B_{\bar{X}}^4}{\sigma_{\min}(H)^4\zeta^4} \leq \frac{c_1\|P_\star\|_{\mathsf{op}}^2\operatorname{cond}(H)^4(d_{\mathsf{X}}^2 + ((1+\beta)\log T)^2)}{\zeta^4(1-\rho)^4}.$$

Furthermore, by Proposition J.5:

$$\|\Gamma_{\mathsf{dep}}(\mathsf{P}_{\bar{X}})\|_{\mathsf{op}}^2 \leq \frac{c_2}{(1-\rho)^2}\log^2\left(\frac{B\sqrt{d_{\mathsf{X}}}(B_{\bar{X}} + B_X)}{2\sigma_{\min}(H)}\right) \triangleq \gamma^2.$$

The class $\mathscr{F}_\star$ is $2BB_{\bar{X}}$-bounded on (56). Invoking Theorem 4.1, for every $r > 0$:

$$\mathbf{E}\|\bar{f} - f_\star\|_{L^2}^2 \leq 8\mathbf{E}\bar{\mathsf{M}}_T(\mathscr{F}_\star) + r^2 + 4B^2B_{\bar{X}}^2\left(1 + \frac{4\sqrt{8}BB_{\bar{X}}}{r}\right)^{d_{\mathsf{X}}^2}\exp\left(\frac{-T}{8C_{\mathsf{GLM}}\gamma^2}\right). \tag{64}$$

Here, the notation $\mathbf{E}\bar{\mathsf{M}}_T(\mathscr{F}_\star)$ is meant to emphasize that the offset complexity is with respect to the truncated process $\mathsf{P}_{\bar{X}}$ and *not* the original process $\mathsf{P}_X$. We now set $r^2 = \|H\|_{\mathsf{op}}^2 d_{\mathsf{X}}^2 / T$, and compute a $T_0$ such that the third term in (64) is also bounded by $\|H\|_{\mathsf{op}}^2 d_{\mathsf{X}}^2 / T$. To do this, it suffices to compute $T_0$ such that for all $T \geq T_0$:

$$T \geq c_3 C_{\mathsf{GLM}} \gamma^2 d_{\mathsf{X}}^2 \log\left(\frac{TBB_{\bar{X}}}{\|H\|_{\mathsf{op}}\sqrt{d_{\mathsf{X}}}}\right).$$

It suffices to take (assuming $\beta$ is at most polylogarithmic in any problem constants):

$$T_0 \geq c_4 \frac{\|P_\star\|_{\mathsf{op}}^2 \mathrm{cond}(H)^4 d_{\mathsf{X}}^4}{\zeta^4 (1-\rho)^6} \mathrm{polylog}\left(B, d_{\mathsf{X}}, \|P_\star\|_{\mathsf{op}}, \mathrm{cond}(H), \frac{1}{\zeta}, \frac{1}{1-\rho}\right). \qquad (65)$$

Again, we do not attempt to compute the exact power of the polylog term, but note it can in principle be done via Du et al. [43, Lemma F.2].

Next, from (59) we have that the error term $\mathbf{E}\|\widehat{f} - f_\star\|_{L^2}^2 \mathbf{1}\{\mathcal{E}^c\} \leq \frac{4B^2 B_{\bar{X}}^2}{T^{\beta/2}}$. Thus if we further constrain $\beta > 2$ and require $T_0 \geq c_5 \left[\frac{B^2\|P_\star\|_{\mathsf{op}}}{(1-\rho)^2}\right]^{\frac{1}{\beta/2-1}}$, then $\mathbf{E}\|\widehat{f} - f_\star\|_{L^2}^2 \mathbf{1}\{\mathcal{E}^c\} \leq \frac{\|H\|_{\mathsf{op}}^2 d_{\mathsf{X}}^2}{T}$. Note that setting $\beta = \max\{3, c_6 \log B\}$ suffices.

To finish the proof, it remains to bound $\mathbf{E}\bar{\mathsf{M}}_T(\mathscr{F}_\star)$. Now, unlike the linear dynamical systems case, there is no closed-form expression for $\mathbf{E}\bar{M}_T(\mathscr{F}_\star)$. Hence, we will bound it via the chaining bound (7). This computation is done in Ziemann et al. [15, Example 3]. Before we can use the result, however, we need to verify that the truncated noise process $\{H\bar{V}_t\}_{t\geq 0}$ is a sub-Gaussian MDS. The MDS part is clear since $\bar{V}_t \perp \bar{V}_{t'}$ for $t \neq t'$, and $\bar{V}_t$ is zero-mean. Furthermore, Proposition G.3 yields that $H\bar{V}_t$ is a $4\|H\|_{\mathsf{op}}^2$-sub-Gaussian random vector. Hence, we have:

$$\mathbf{E}\bar{\mathsf{M}}_T(\mathscr{F}_\star) \leq c_7 \frac{\|H\|_{\mathsf{op}}^2 d_{\mathsf{X}}^2}{T} \log(1 + \|H\|_{\mathsf{op}}\sqrt{d_{\mathsf{X}}} BB_{\bar{X}} T^2).$$

The claim now follows.

## J.3 Further discussion related to Theorem 6.2

Several remarks on Assumption 6.1 are in order. First, the rank condition on $H$ ensures that the noise process $\{HV_t\}_{t\geq 0}$ is non-degenerate. Viewing (16) as a control system mapping $\{V_t\}_{t\geq 0} \mapsto \{X_t\}_{t\geq 0}$, this condition ensures that this system is one-step controllable. Next, the link function assumption is standard in the literature (see e.g. Kowshik et al. [20], Sattar and Oymak [30], Foster et al. [31]). The expansiveness condition $|\sigma(x) - \sigma(y)| \geq \zeta|x - y|$ ensures that the link function is increasing at a uniform rate. For efficient parameter recovery, some extra assumption other than Lipschitzness and monotonicity is needed [20, Theorem 4], and expansiveness yields a sufficient condition. However, for excess risk, it is unclear if any extra requirements are necessary. We leave resolving this issue to future work. Finally, the Lyapunov stability condition is due to Foster et al. [31, Proposition 2], and yields a certificate for global exponential stability (GES) to the origin. It is weaker than requiring that $\|A_\star\|_{\mathsf{op}} < 1$, which amounts to taking $P_\star = I$. The assumption $P_\star \succcurlyeq I$ is without loss of generality by rescaling $P_\star$.

Theorem 6.2 states that after a polynomial burn-in time (which scales quite sub-optimally as $\tilde{O}(d_{\mathsf{X}}^4)$ in the dimension), the excess risk scales as the minimax rate $\|H\|_{\mathsf{op}}^2 d_{\mathsf{X}}^2 / T$ times a logarithmic factor of various problem constants. To the best of our knowledge, this is the sharpest excess risk bound for this problem in the literature and is nearly minimax optimal. As noted previously, the logarithmic factor enters via the chaining inequality (7) when bounding the martingale offset complexity. We leave to future work a more refined analysis that removes this logarithmic dependence, and also improves the polynomial dependence of $T_0$ on $d_{\mathsf{X}}$. The extra $d_{\mathsf{X}}^2$ factor in (19) over the LDS burn-in time (15) comes from our analysis of the trajectory hypercontractivity constant for this problem, and should be removable.

Much like in (52), we can use the link function expansiveness in Assumption 6.1 to convert the excess risk bound (18) to a parameter recovery rate:

$$\mathbf{E}\|\hat{A} - A_\star\|_F^2 \leq \tilde{O}(1)\frac{\|H\|_{\mathsf{op}}^2 d_{\mathsf{X}}^2}{\zeta^2 T \lambda_{\min}(\bar{\Gamma}_T)}, \qquad (66)$$

where again $\bar{\Gamma}_T \triangleq \frac{1}{T} \sum_{t=0}^{T-1} \mathbf{E}[X_t X_t^\mathsf{T}]$ is the average covariance matrix of the GLM process (16). Note that the one-step controllability assumption in Assumption 6.1 ensures that the covariance matrix $\mathbf{E}[X_t X_t^\mathsf{T}] \succcurlyeq HH^\mathsf{T}$ is invertible for every $t \in \mathbb{N}$. A detailed comparison of the excess risk rate (18) and parameter recovery rate (66) with existing bounds in the literature is given in Appendix J.1.

Let us briefly discuss the proof of Theorem 6.2. As in the LDS case, we use the truncation argument described in Appendix B.1 that allows us apply Theorem 4.1 while still using bounds on the dependency matrix coefficients of the original unbounded process (16). However, an additional complication arises compared to the LDS case, as the covariates are not jointly Gaussian due to the presence of the link function. While at this point we could appeal to ergodic theory, we instead develop an alternative approach that still allows us to compute explicit constants. Building on the work of Chae and Walker [42], we use the smoothness of the Gaussian transition kernel to upper bound the TV distance by the 1-Wasserstein distance. This argument is where our analysis crucially relies on the non-degeneracy of $H$ in Assumption 6.1, as the transition kernel corresponding to multiple steps of (16) is no longer Gaussian. The 1-Wasserstein distance is then controlled by using the *incremental stability* [48] properties of the deterministic dynamics $x_+ = \sigma(A_\star x)$. Since this technique only depends on the GLM dynamics through incremental stability, it is of independent interest as it applies much more broadly.