# OpenReview forum: "Learning with little mixing"
_NeurIPS.cc/2022/Conference — NeurIPS 2022 Accept_

### Official Review · Reviewer_U6WV · 2022-07-05

**Rating:** 4
**Confidence:** 2
**Soundness:** 2 fair
**Presentation:** 2 fair
**Contribution:** 2 fair

**Summary:**

The authors investigate the square loss in a realizable time-series framework with martingale difference noise, which is an interesting topic in machine learning with non-i.i.d. data. Their main result is a fast rate excess risk bound which shows that whenever a trajectory hypercontractivity condition holds, the risk of the least-square estimator on dependent data matches the iid rate order-wise after a burn-in time. Moreover, the authors give some examples of when the condition holds. I find the main context easy to follow.

**Questions:**

1. Can the authors provide a more detailed discussion and comparison with the previous related work so that I can make a fair judgement about the contribution of this work?
2. Can the authors elaborate more on the technical new insight of this work?
3. Can the authors conduct some numerical studies on this phenomenon?
4. Line 524: Both sides of this inequality seem to be random variables while the distance, to which the covering number is respect, is expected value, is this a mistake? Or is there any guarantee that this inequality holds with probability?

**Ethics Review Area:**

["I don’t know"]

**Limitations:**

1. This paper lacks a detailed discussion and comparison with the previous work.
2. This paper seemed not to give any new insight on this field.

**Strengths And Weaknesses:**

Strength: 1. This paper is technical. It is clearly written and well organized.
2. The result in this paper is significant.
Weakness: 1. This paper requires a more detailed discussion and comparison with the previous related work.
2. There are some confusing mistake in the proof of the main results.

---

> ### Author Response · Authors · 2022-08-02
> **Response to Reviewer U6WV**
>
> We thank reviewer U6WV for their time and effort to review this submission. We note that reviewer U6WV believes that the paper is clearly written and well-organized and that the result is significant. It is our understanding that reviewer U6WV awards us score of 4 due to the brevity of our related work section and a misunderstanding about the correctness of one of our arguments (line 524). We hope that the following addresses the reviewer's concerns:
>
> **Line 524: Both sides of this inequality seem to be random variables while the distance, to which the covering number is respect, is expected value, is this a mistake? Or is there any guarantee that this inequality holds with probability?**
>
> - We believe that
> the terseness of our original derivation may have caused
> a simple misunderstanding of our argument, and we have expanded the derivation
> in the revision to clarify the point. In particular:
>
> - The relation on Line 524 (which has now shifted to Line 677 in our new revision) holds for every realization of the covariate process $\{X_t\}$,  since the
> cover $\mathscr{F}_r$ is a cover of the set $\partial B(r)$
> in the *supremum norm* over the compact set $\mathsf{X}$ which the covariates evolve on (i.e., the norm is $\sup ||f(x)||_2$ for a function $f$, and where the supremum is taken over $x \in \mathsf{X}$). This is indicated in the statement of Theorem 4.1.
> It is true that the definition of $\partial B(r)$
> involves an expectation, since it is the
> set of $f \in \mathscr{F}_\star$ such that the $L^2$ norm equals $r$. However, note that the set itself is deterministic.
>
> - If there was anything else reviewer U6WV had in mind, please let us know and we will be more than happy to address your concerns.
>
> **Can the authors provide a more detailed discussion and comparison with the previous related work so that I can make a fair judgement about the contribution of this work?**
>
> We have taken major steps to better place our work in the existing literature:
>
> - We have made significant changes to our contribution and related work section. In total, we have included 10 further related references.
>
> - We have included two new sections devoted to learning system identification with parametric classes (Section 6). Therein, we provide precise statements that recover (and in some cases even extend) existing results on (a) linear dynamical systems and (b) generalized linear model dynamics which both individually have received significant attention in recent literature. We also provide further discussion (found in the Appendix I.4 and Appendix J.3, respectively) on the relationship between our work and these existing results.
>
> **Can the authors elaborate more on the technical new insight of this work?**
>
> - We have expanded our contribution section and improved our literature review to clarify this. In essence, our work marks a departure from the more typical blocking technique by providing a method to analyze the lower tail of the empirical excess risk for dependent process at a rather general level. This allows us to escape polynomial dependency on the mixing time which essentially all other works at this level of generality incur. En route to this result we also introduce the trajectory hypercontractivity condition (Definition 4.1)--- a condition we proceed to show recovers most existing cases of the phenomenon  we study (i.e., the mixing time only having higher order affect) and also provides several new ones including RKHSs.
>
> **Can the authors conduct some numerical studies on this phenomenon?**
>
> -  We have conducted numerical experiments that confirm our findings in Appendix A.

---

### Official Review · Reviewer_SNkB · 2022-07-08

**Rating:** 4
**Confidence:** 3
**Soundness:** 2 fair
**Presentation:** 2 fair
**Contribution:** 2 fair

**Summary:**

The paper shows that for mixing systems under an easiness condition, the rate of convergence of the LSE for rather general hypothesis classes has i.i.d. data like performance.

**Questions:**

Is it possible to show that, under regularity conditions, for most loss functions, "learning with little mixing" isn't too much worse compared to i.i.d. data?

**Strengths And Weaknesses:**

The paper proves excess risk bounds of LSE with dependent data, however the results aren't very surprising, and contributions are a bit too incremental for me.

---

> ### Author Response · Authors · 2022-08-02
> **Response to Reviewer SNkB**
>
> We are disappointed to have received such a brief and dismissive review and encourage the reviewer to offer more concrete criticism, which we would happily engage with. In the meanwhile, we do not believe that our findings are incremental. Namely:
>
> - We are not aware of any other results this level of generality that accurately capture the correct rate for ERM applied to time-series data. To the best of our knowledge, other works operating at this level of generality all rely on the blocking technique and incur a corresponding sub-optimal dependency on the mixing time of the covariates---a dependency which directly enters their rate guarantee. Sidestepping this sub-optimality, as we do, is a nontrivial technical contribution and requires careful adaptation of Mendelson's small-ball method to dependent processes.
>
> - A consequence of operating at this level of generality is that we are able to provide the first unified framework which captures many existing examples of when dependent processes exhibit the iid rate as diverse as discrete Markov chains, linear dynamical systems, and now also generalized linear model dynamics. These types of systems have received significant interest at recent venues such as NeurIPS and COLT. Thus, from a theoretical perspective, such a unification behind a single technical machinery is both interesting and important as it clarifies precisely why this particular phenomenon occurs.
>
> - Existing results on this topic (demonstrating the iid rate phenomenon) consist of analyses specialized to particular finite-dimensional parametric classes which roughly correspond to problems solvable by either linear regression or generalized linear regression. While our results also apply in the above-mentioned cases, they stand in contrast to previous work in that they do not rely on any such assumption. In fact, we are first to show that the phenomenon extends to general hypothesis classes including for instance infinite-dimensional RKHSs.
>
> The reviewer also asks whether it is possible to extend the material beyond square loss.  If one wishes to use Samson's inequality (Theorem B.1) to prove lower isometry for the excess risk, then one needs a nonnegative proxy for the excess risk (which one might hope to obtain under certain strong convexity and Lipschitz assumptions on the loss).
> Given such a proxy, the remaining issue is localizing the empirical excess risk. For square loss this can be done without the standard symmetrization-contraction machinery due to the offset basic inequality (cf. Eq. 30). For more general loss functions, more work remains to be
> done, which we leave as future work.

---

### Official Review · Reviewer_AYjz · 2022-07-12

**Rating:** 7
**Confidence:** 4
**Soundness:** 4 excellent
**Presentation:** 3 good
**Contribution:** 4 excellent

**Summary:**

The authors study the problem of learning from dependent data over time, with the aim of obtaining empirical risk minimization bounds that do not depend on the mixing time of the process. They consider a time-series framework with martingale difference noise, and prove a general result: under an assumption they introduce called *trajectory hypercontractivity* (and sublinear growth in the dependency matrix), the risk of the least-squares estimator matches the iid rate after a burn-in time (note the burn-in time can depend on the mixing). This is in contrast to naive bounds where the effective sample size is deflated by a factor of the mixing time.

The proof relies on using the hypercontractivity to control the lower tail of sums involving the dependent random variables.

The authors specialize the result both to non-parametric function classes and those with logarithmic metric entropy. They give several examples where their conditions are satisfied (and which recover or generalize previous results): finite-state Markov chains, bounded function classes for which $L^2,L^{2+\epsilon}$ norms are equivalent, and infinite-dimensional function classes based on subsets of $\ell^2(\mathbb N)$ ellipsoids (e.g., functions of bounded norm in a RKHS).

**Questions:**

* 233: When $\alpha = 1$, it is not possible to remove the dependence on $\|\Gamma_{\text{dep}}(P_X)\|_{\text{op}}^2$ in the lowest order term - Why? Is there a lower bound?
* 304: Why is a lower bound on the marginal probabilities needed? Is this necessary?
* 315: What are natural examples where a $L^{2+\epsilon},L^2$ norm equivalence holds?
* (23): How is this inequality obtained? The assumption is that $\|\phi_j\|_\infty\le Bm^q$ for $j\le m$, but the terms in the sum are $j>m$.

Minor edits

* 211: hypercontracitity -> hypercontractivity
* 544: some constant factors as missing in the following inequalities.
* 549: missing log in following inequality
* 559: missing inequality $T\ge T_1,T_2,...$

**Limitations:**

Certain known results are not covered by the framework, in particular, learning linear dynamical systems that are marginally stable or which have unbounded noise. Additionally, as the max eigenvalue approaches 1, the necessary burn-in time given by the theorem blows up, whereas known results do not have this dependence. (This stems from reliance on rate of growth of the dependency matrix--while the asymptotic rates do not depend on the mixing, the burn-in time does.) The authors discuss this in Section 4.3.

**Strengths And Weaknesses:**

The strength of the paper lies in the fact that it gives a very general result that unites previous results under a general framework, e.g., results on learning linear dynamical systems and finite Markov chains. There is a large degree of quantitative flexibility in the assumption that the authors introduce (trajectory hypercontractivity, which interpolates between boundedness and small-ball behavior). The proofs in the appendix are easy to follow.

However, the main body of the paper is technically dense and not easy to digest; the examples are fairly abstract. It would help the exposition significantly to expand on concrete instantiations of the theorem (moving more techical commentary to the appendix as necessary), for example, writing out the theorem for linear dynamical systems obtained from the general theorem.

There are also some limitations to the theorem (see below).

---

> ### Author Response · Authors · 2022-08-02
> **Response to Reviewer AYjz**
>
> We thank the reviewer for their time and effort to review this submission and their helpful suggestions. We address your suggestions and feedback below:
>
> **It would help the exposition significantly ... writing out the theorem for linear dynamical systems obtained from the general theorem.**
>
> - We have followed your suggestion and instantiated our main result to linear dynamical systems (Theorem 6.1).  Moreover, we also did this for generalized linear model dynamics which have also been recently considered in related literature (we were unable to do this optimally in the original submission).
> Theorem statements can be found in Section 6, with proofs relegated to the appendix.
>
> - In both cases, we recover minimax optimal rates in the stable regime (but with an extraneous logarithm in the GLM case). For these particular instances,
> we also show how to overcome the bounded noise limitation by a coupling argument (in which we couple the original unbounded noise process with a noise-truncated process). An extended discussion of these results (and their relation to previous work) can also be found in the appendix.
>
> **Answers to questions:**
> - [line 233]:  (necessity of $\alpha > 1$) In our work, $\alpha>1$ corresponds to small-ball like behavior (as can be seen by an application of Paley-Zygmund). The issue is that $\alpha =1$ (boundedness) does not really offer any small-ball behavior beyond that which is offered by concentration of measure. If one tries to replicate the covering argument in our lower isometry proof, using only concentration, one is hit by a central-limit theorem (CLT) variance/scale term. For dependent processes, the CLT variance is known (in the worse case) to scale with the mixing time of the corresponding process [1]. This also marks the critical obstacle to removing the mixing term at $\alpha=1$.
>
> - We have unfortunately not yet been able to come up with a lower bound or counter-example in the absence of $\alpha>1$, but certainly agree that this is an interesting question. We hope to be able to shed further light on this in future work.
>
> - [line 304]: (lower bound on marginal probability for RKHS) Here, a lower on the marginal probabilities is used to estimate the $L^4$-norm in terms of the $L^2$ norm, via a simple change of measure
>     argument. Some assumption is likely needed, but we think alternative hypotheses are possible. We think it would be interesting to further study the interplay between dimensionality and mixing-free rates.
>
> - [line 315]: (Examples of equiv. $L^p$ norms) For $\epsilon<2$, the point of the $L^{2+\epsilon}-L^2$ proposition is mostly to show that from a theoretical perspective there is nothing really special with the case $\epsilon=2$ (other than that examples are relatively easy to come by). For a concrete example, we refer to Theorem 7 of [2] which gives such equivalences for polynomials of log-concave random variables.  The assumption has also frequently featured in the literature, see e.g., [3].
>
> - [eq. (23)]: Thanks for noticing! This is a good catch and a mistake from our side. The correct hypothesis should read $\|\phi_n\|_\infty \leq B n^q \: \forall n\in \mathbb{N}$ (as to capture subexponential growth of the eigenfunctions). We have adjusted the statement to accommodate for this (it affects the final statement by no more than a factor of two inside a logarithm).
>
> We also appreciate the pointers to the typos and various
> constant factor miscalculations.
> We have corrected them in the updated manuscript; none of the miscalculations affect the results beyond
> constant factors.
>
> [1] Meyn, Sean P., and Richard L. Tweedie. Markov chains and stochastic stability. Springer Science {\&} Business Media, 2012.
>
> [2] Carbery, Anthony, and James Wright. Distributional and $ L^{q} $ norm inequalities for polynomials over convex bodies in ${\Bbb R}^ n$. Mathematical research letters 8.3 (2001): 233-248.
>
> [3] Mendelson, Shahar. On aggregation for heavy-tailed classes. Probability Theory and Related Fields 168.3 (2017): 641-674.

---

### Official Review · Reviewer_nYrc · 2022-07-15

**Rating:** 7
**Confidence:** 1
**Soundness:** 3 good
**Presentation:** 3 good
**Contribution:** 3 good

**Summary:**

This paper studies square loss in a realizable time-series framework, the main result shows that whenever a trajectory hypercontractivity condition holds, the risk of least squares estimator on dependent data matches the iid rate order-wise after a burn-in time. The paper formulates a phenomenon called learning with little mixing and presents several examples where such phenomenon occurs.

**Questions:**

Can we testify the theory on simple regression problems, empirically?

**Ethics Review Area:**

["I don’t know"]

**Limitations:**

There is no negative societal impact.

**Strengths And Weaknesses:**

This paper gives solid theoretical results on learning with dependent data. It shows on a broad class of examples, the LSE applied to time-series model behaves as if all samples are independent given enough data. Although I am not familiar with the background of this problem, the results look insightful. On the other hand, I'd also be curious to see if the theory can be testified empirically on simple regression problems.

---

> ### Author Response · Authors · 2022-08-02
> **Response to Reviewer nYrc**
>
> Thank you for the positive comments regarding our paper.
> As for numerical validation,
> we have included new simulations detailed in Appendix A.

---

### Author Response · Authors · 2022-08-02
**Overview of changes**

**EDIT:** Fixed some minor typos in our response.

We thank the reviewers for their time, effort and helpful comments. To aid the reviewers we now provide an overview of all major changes made between the current version and original submission. We then proceed to answer each reviewer individually in separate responses.

**Major Changes**:

- As requested by reviewer AYjz, we now explicitly instantiate  our results to linear dynamical systems (Theorem 6.1).
- In contrast to the version we originally submitted, we are now also able to recover (and extend) existing results on generalized linear model dynamical systems. The result is stated in Theorem 6.2.
- In both cases above, we provide comprehensive discussion on Theorems 6.1 and 6.2, and how they relate to previous literature (see Appendix I.4 and Appendix J.3, respectively). We hope that this resolves some of the concerns of reviewer U6WV (related to our literature review).
- As requested by reviewers U6WV and nYrc, we have run a numerical experiment which clearly illustrates the reported phenomenon. More on this below.
- As requested by reviewer U6WV, we have reworked our related work (Section 2) as to better place our results in the context of existing literature.
- To create space and accommodate for the new material asked for by the reviewers, we have moved the proof overview/sketch to Appendix B, and it now immediately precedes the proof itself.

**Numerical evidence:**

We compare the performance of the empirical risk minimizer on the following two datasets:
- The first dataset is simply a single trajectory of length $T$ from the generalized linear model dynamical system described in Section 6.2.
- The second dataset (which we name the *independent baseline*)
    consists of $T$ samples drawn independently from the *marginal*
    distributions of the GLM dynamical system. This dataset breaks the correlations of the
    covariates $X_t$ across time.

We generate our data this way to avoid confounding our claims with issues of
mismatching $L^2$ risks.
Our plots do indeed demonstrate that as $T$ grows, the excess risk of learning
from a trajectory of length $T$ approaches that of learning from
the independent baseline.
A more thorough description can be found in Appendix A.

---

### Author Response · Authors · 2022-08-09
**Further questions**

As the reviewer/author discussion period ends tomorrow, if there are any additional points regarding our response that the reviewers would like clarified, please don't hesitate to reach out with further questions.

---

### Meta-Review · Area_Chair_ve4c · 2022-08-25

**Recommendation:** Accept
**Confidence:** Less certain

**Metareview:**

This paper studies the problem of learning under dependent data. Existing bounds usually work by deflating the effective sample size by a factor that depends on the mixing time. Essentially when the samples are far enough away from each other, depending on the mixing time, they can be treated as independent. This paper introduces a new framework that they call the trajectory hypercontractivity condition, which stipulates that there is sublinear growth in the dependency matrix. This is a flexible perspective, and the paper derives both general results and applies them in interesting settings. There are some weaknesses, e.g. they cannot recover results in the marginally stable case or in settings with unbounded noise. For example, as the maximum eigenvalue approaches one, the burn-in blows up. I think reviewer SNkB's perfunctory review should be ignored. The paper is somewhat borderline, but in my opinion it is technical stronger and more interesting than some of the other borderline papers in my batch. I recommend acceptance.

**Award:**

No

---

### Decision · Program_Chairs · 2022-09-14

Accept